# Metrological concepts applied to Total Alkalinity measurements in seawater: reference materials, inter-laboratory comparison and uncertainty budget

Gaëlle Capitaine[1,2], Samir Alliouane[3], Thierry Cariou[4], Jonathan Fin[5,a], Paola Fisicaro[1], Thibaut Wagener[2]

[1]Laboratoire National de Métrologie et d'Essais (LNE), Paris, 75015, France
[2]Aix Marseille Université, CNRS, IRD, MIO, Marseille, 13288, France
[3]Sorbonne Université, CNRS, Laboratoire d'Océanographie de Villefranche, LOV, Villefranche-sur-Mer, F-06230, France
[4]IRD, UAR191, Instrumentation, Moyens Analytiques, Observatoires en Géophysique et Océanographie (IMAGO), Technopôle de Brest-Iroise, Plouzané, 29280, France
[5]Laboratoire LOCEAN/IPSL, Sorbonne Université-CNRS-IRD-MNHN, Paris, 75005, France
[a]Present address: Institut des Sciences de la Terre, Grenoble, 38058, France

*Correspondence to*: Gaëlle Capitaine (gaelle.capitaine@lne.fr)

**Abstract.** Total alkalinity (TA) measurements in seawater are crucial for characterizing and monitoring the oceanic carbonate system. While international best practices and guidelines exist, the field still lacks widely available traceable reference materials and a well-established uncertainty budget of the measurement method. In this study, we applied key metrological principles—development of reference materials, inter-laboratory comparison and uncertainty quantification—to TA measurements. We developed two reference materials, including an artificial material with a rigorously characterized reference value and an associated uncertainty budget, being traceable to the International System of units (SI). These materials were tested in an inter-laboratory comparison involving five laboratories and demonstrated the applicability of the reference materials developed for quality control. Additionally, we established an uncertainty budget for the TA measurement method using two metrological approaches. The resulting expanded uncertainty was 5 µmol kg$^{-1}$ (with a coverage factor $k = 2$) in TA, approaching the 4 µmol kg$^{-1}$ target set by the Global Ocean Acidification Observing Network for climate monitoring. These findings mark a significant step toward improving the quality and comparability of TA measurements, thereby strengthening long-term ocean carbonate system monitoring.

## 1 Introduction

Total alkalinity of seawater (TA) represents the excess of proton acceptors over proton donors and can be described in a simplified manner as the buffer capacity of seawater. The exact definition of the total alkalinity is the number of moles of hydrogen ion equivalent to the excess of proton acceptors (bases formed from weak acids with a dissociation constant K ≤ 10$^{-4.5}$, at 25°C and zero ionic strength) over proton donors (acids with K > 10$^{-4.5}$, same conditions) in one kilogram of seawater (Dickson, 1981). This definition, which is the one commonly accepted, is represented in terms of ionic chemical model by Eq. 1.

$$TA = [HCO_3{}^-] + 2[CO_3{}^{2-}] + [B(OH)_4{}^-] + [OH^-] + [HPO_4{}^{2-}]$$
$$+2[PO_4{}^{3-}] + [SiO(OH)_3{}^-] + [NH_3] + [HS^-] + [...] - [H^+]_F -$$
$$[HSO_4{}^-] - [HF] - [H_3PO_4] - [...]$$

$$(1)$$

where brackets represent amount contents (mol kg$^{-1}$) and ellipses corresponds to minor species.

Total alkalinity is an essential independently measurable variable contributing to the monitoring of changes in the ocean carbon cycle and ocean acidification, that can be used together with pH$_T$ (on the total scale), Dissolved Inorganic Carbon (DIC) or partial pressure of CO$_2$ (pCO$_2$) to compute all other variables of the ocean carbonate system. Moreover, total alkalinity is a relatively simple variable to monitor thanks to the fact that (1) it is independent from temperature and pressure, unlike pH$_T$ and pCO$_2$, and (2) it isn't affected by atmospheric CO$_2$, unlike DIC, that could come for example from the exposure of the sample to air.

Ensuring the quality of total alkalinity measurement results is of great importance. The Global Ocean Acidification Observing Network and the World Meteorological Organization have fixed a data quality objective corresponding to a standard uncertainty in total alkalinity measurement results of 1 and 2 µmol kg$^{-1}$, respectively (GOA-ON, 2019; WMO et al., 2022). The GOA-ON value was chosen in order to obtain a 1% standard uncertainty in the computation of the carbonate ion amount content variable, enabling to highlight climatic variations in the monitoring of ocean acidification.

Contributing to the objective of achieving comparable TA measurement results, the Scripps Institution of Oceanography currently distributes reference materials constituted of a stabilized natural seawater (Dickson, 2010). These materials are carefully characterized in terms of total alkalinity using the open-cell multi-step potentiometric titration method, whose accuracy has been validated with synthetic solutions constituted of bases such as sodium carbonate, borate or TRIS (Dickson et al., 2003). It is the only laboratory producing reference materials (RMs) for total alkalinity measurements on a regular basis (Acquafredda et al., 2022). However, the RMs distributed aren't fully traceable partly due to the fact that they aren't given with a rigorously assessed uncertainty. Other traceability issues coming from the measurement process should also be carefully investigated. Developing a reference material made in artificial seawater, characterized with a traceable reference method, and with a thoroughly quantified uncertainty that could be distributed together with a natural seawater such as the one from Scripps, might help in assessing more robustly a possible measurement bias. Moreover, the uncertainty budget of the measurement results is required to check the compatibility among total alkalinity values.

The first aim of the work presented in this paper was thus to develop a reference material produced following the international standards appropriate to the production of reference materials (ISO 17034:2016, 2016; ISO 33405:2024, 2024). This material is made of artificial seawater with a total alkalinity reference value attributed from knowledge of the composition. The uncertainty budget associated to the reference value is determined following the Guide to the expression of Uncertainty in Measurement (GUM, 'JCGM 100:2008', 2008) and integrates information about stability and homogeneity of the material.

This developed reference material has been tested in an inter-laboratory comparison conducted with five French laboratories conducting TA measurements with the standardized method, being the multi-step potentiometric titration method. Measurements were also performed on a second material, produced similarly to the one from Scripps, being a stabilized natural seawater from the Mediterranean Sea. The second aim of the work presented was, from the inter-laboratory comparison, to study the applicability in quality control for total alkalinity measurements of the artificial and natural solutions developed as reference materials.

The third objective was to thoroughly establish the uncertainty budget of the standardized measurement method, which is up to date lacking. This paper thus presents for the first time an uncertainty estimation for the open-cell multi-step potentiometric titration method together with Gran's data treatment, established following the GUM (i.e. bottom-up approach). A comparison with the uncertainty budget obtained from the inter-laboratory comparison results (i.e. top-down approach) is also presented.

## 2 TA measurement method

The titration method has always been the measurement method of choice for the determination of seawater total alkalinity (Greenberg et al., 1932).

The procedure consists in a multi-step addition of acid, with algorithmic determination of the equivalence point from potentiometric data titration curve (Dickson, 1981; Edmond, 1970). This method either can be used in an open or closed cell. This multi-step titration is recognized as the best-practice method for measurements in seawater compared to the single-step method (Dickson et al., 2007). Therefore, this is the procedure chosen and thus the one referring to in the rest of this paper.

The standard procedure as well as the data treatment method for the open-cell multi-step titration has been well described in the literature (Dickson et al., 2003, 2007; Okamura et al., 2014; Wolf-Gladrow et al., 2007), and is detailed in Appendix A.

The measurement model of TA obtained from the titration curve using Gran's method (Gran, 1952) is presented below, and will be used for the establishment of the uncertainty budget.

$$TA = \frac{-b}{a} \frac{v_{HCl}}{m_{init}} \tag{2}$$

where $v_{HCl}$ is the acid amount content (mol kg$^{-1}$), $m_{init}$ the mass of sample analysed (g) and coefficients $a$ and $b$ represent, respectively, the slope and the intercept of the linear regression $F1 = a * m_{HCl} + b$.

The Gran function $F1$ is represented by Eq. 3.

$$F1 = (m_{init} + m_{HCl}) \times \exp\left(\frac{E}{\frac{RT}{F}}\right) \tag{3}$$

where $m_{HCl}$ is the mass of acid added (g), $E$ is the potential measured by the glass electrode (V), $R$ the universal gas constant (J mol$^{-1}$ K$^{-1}$), T the temperature of the sample (K), F the Faraday constant (C mol$^{-1}$)

This method gives a first estimation of the total alkalinity. However, errors are introduced when using the Gran's method for seawater analysis due to competing acid-base equilibria in seawater. A method allowing to solve the equivalence point using a nonlinear least-squares (NLLS) regression has thus been developed (Dickson, 1981; Martz, 2005). This method is detailed in Appendix A.

## 3 Development of reference materials

### 3.1 Materials and Methods

Two reference materials have been developed for the quality control of seawater total alkalinity measurement methods: a stabilized natural seawater and an artificial seawater. This section details the methods applied for the preparation, characterization, stability and homogeneity studies, as well as for the uncertainty quantification of the reference TA value assigned to the artificial solution.

### 3.1.1 Preparation

Stabilized natural seawater

The natural seawater was collected by the Mediterranean Institute of Oceanography (MIO) during an oceanographic field trip to the Antares station (42°48 N 6°10 E) of the Mediterranean Ocean Observation System for the Environment (MOOSE; Lefevre, 2010). Deep waters of respectively 2000, 1750, 1500 and 1000 meters depth were collected in two plastic containers and homogenized, for a total of 35 liters of seawater. The containers were stored protected from light at 4°C until filtration. 25 liters of the collected seawater were filtered with a 0.2 µm Sartobran filter using a Masterflex peristaltic pump, and gathered from the two containers to one unique container in Nalgene (polycarbonate). 10 ml of a solution of mercuric chloride at 36 g l$^{-1}$ were added to the seawater in the Nalgene container, corresponding to the usual concentration of 0.02% saturated $HgCl_2$. The container was stirred to ensure homogeneity of the seawater. The natural seawater was then bottled in 42 ground-neck *PYREX* borosilicate 3.3 bottles of 500 ml sealed with greased glass stoppers held on with elastic bands. The bottles have previously been cleaned with diluted acid and detergent in several consecutive cycles, rinsed with distilled water and dried.

Artificial seawater

The composition of the artificial seawater was chosen in order to have a total alkalinity of 2500 µmol kg$^{-1}$, based on gravimetric information, and with salinity and pH values that match those of a natural seawater. To fulfill these criteria, the artificial seawater is composed of sodium bicarbonate ($NaHCO_3$), sodium carbonate ($Na_2CO_3$) and is made in a NaCl matrix. It should be noted that even if called "artificial seawater", the solution presented is made in a simple NaCl matrix, without other common seawater salts. The targeted composition of the artificial solution is given in Table 1.

**Table 1: Targeted composition and information about ionic strength (I) and absolute salinity of the artificial solution for total alkalinity reference material**

|  | Salts coulometric purity assessment (%) | $\nu$ (mol kg$^{-1}$) | $b$ (mol kg$^{-1}$ H$_2$O) |
|---|---|---|---|
| $NaHCO_3$ (*Merck*) | 100.039 | 0.0022746 | 0.0023578 |
| $Na_2CO_3$ (*Merck*) | 99.948 | 0.0001127 | 0.0001168 |
| NaCl (*VWR chemicals*) | 99.945 | 0.6000000 | 0.6219348 |
| I | | 0.625 | |
| Absolute salinity | | 35 | |

Note: where $\nu$ is the amount content, expressed in mol kg$^{-1}$, $b$ is the molality, expressed in mol kg$^{-1}$ H$_2$O and I is the ionic strength. The suppliers of the salts are given in brackets.

The artificial solution was prepared from respective stock solutions of $NaHCO_3$ and $Na_2CO_3$. The purity of the sodium carbonate and sodium bicarbonate salts used were characterized in terms of purity as bases expressed as sodium carbonate by coulometric analysis performed at the National Metrology Institute of Japan (NMIJ), while the purity of the sodium chloride was characterized by coulometric analysis at the Slovack Metrology Institute (SMU) based on chloride content. The sodium carbonate salt was dried at 280°C for 4 hours and cooled down in a desiccator for 1h before use, to remove potential humidity. This procedure wasn't applied to sodium bicarbonate salt due to the decomposition reaction caused by heat. The same pre-treatment of the salts was applied before

characterization at NMIJ and before stock solution preparation at LNE, ensuring that the purity obtained by coulometric analysis is suitable for the solution preparation.

Two batches of artificial seawater solution of respectively 12 and 7 kg were prepared. The first batch (Batch 1) was bottled in equivalent bottles to the one used for the natural seawater (i.e. ground-neck borosilicate *PYREX* 3.3 bottles of 500 ml sealed with greased glass stoppers maintained with elastic bands). While the second batch (Batch 2) is bottled in *DURAN - SCHOTT* borosilicate 3.3 bottles of 500 ml with screw caps containing PTFE coated seals. A cleaning treatment similar to that used for the bottling of natural seawater was applied to the bottles.

The stabilized natural seawater and Batch 1 of the artificial solution were distributed to be studied in the inter-laboratory comparison described in Sect. 4. The Batch 2 of the artificial seawater was analysed at LNE, with the objective of comparing the stability of the material for the two methods of bottling.

### 3.1.2 Characterization

The artificial seawater was characterized in terms of total alkalinity based on gravimetric information. Given the composition of Table 1 and the total alkalinity definition (Eq. 1), the alkalinity introduced in the artificial solution is supposed to only come from carbonate and bicarbonate ions, as described by Eq. 4.

$$TA = 2 \, v_{Na_2CO_3} + v_{NaHCO_3} \tag{4}$$

where $v$ represents amount contents (mol kg$^{-1}$).

However, the impurities contained in the NaCl salt (0.055%) can also contribute to the total alkalinity of the solution. This source of alkalinity is hereafter called background alkalinity (noted $TA_{background}$, expressed in mol kg$^{-1}$).

The final total alkalinity of the artificial solution was thus obtained from Eq. 5.

$$TA = 2 \, v_{Na_2CO_3} + v_{NaHCO_3} + TA_{background} \tag{5}$$

The background alkalinity was quantified based on the preparation of four solutions with the same amount contents of bicarbonate and carbonate ions while varying the NaCl amount content from 0 to 3 mol kg$^{-1}$ (solutions in NaCl matrix of respectively 0, 1, 2 and 3 mol kg$^{-1}$). The total alkalinity of each of the four solutions was determined at LNE from, respectively, the mean of at least three repeatability measurements made using the open-cell multi-step potentiometric titration method, as described in Sect. 2, and using an HCl certified solution from SMU and a Metrohm titration system, as described in Appendix B. The difference of total alkalinity between the measured one and the theoretical value calculated with Eq. 4, was represented as a function of the amount content of NaCl for each of the four solutions. With these data a linear regression was computed passing through the origin. The linear regression was forced to pass through the origin as the background alkalinity coming from NaCl impurities is theoretically zero for a solution without NaCl matrix. The measurement results at zero NaCl mol kg$^{-1}$ are shown in Fig. 1 and support this reasonable assumption. The linear relation obtained allowed the determination of the background alkalinity for the solution studied (i.e. for a NaCl amount content of 0.6 mol kg$^{-1}$, Table 1). Linearity of the measurement results is a rough assumption that is further discussed in Sect. 3.3.2.

Batch 1 of the artificial reference material and the stabilized natural seawater were characterized in terms of practical salinity aboard the *Thalassa* oceanographic vessel during the 2023 PIRATA cruise (Bourles et al., 2023; Llido, 2023), using an *OSIL Portasal 8410A* salinometer. This variable is needed for computing total alkalinity with the NLLS regression.

The natural seawater was also characterized in terms of dissolved nutrients (i.e. silicates, nitrites, phosphates and nitrates) based on colorimetric determination using a *SEAL AutoAnalyzer 3 HR* at the platform of Analysis of Basic Parameters (PAPB) of the MIO. The monitoring of nutrients gives relevant information for stability assessment.

### 3.1.3 Stability and homogeneity studies

Homogeneity

Homogeneity estimations were based on TA measurements carried out at LNE, MIO and at the French National Service for Analysis of Oceanic $CO_2$ Parameters (SNAPO-$CO_2$) following the standardized multi-step potentiometric titration method. The compatibility of the measurements performed by these three institutes have first been established.

The homogeneity assessment integrates two components: (1) the between-bottle homogeneity, taking into account standard deviation between different bottles of a same batch, and (2) within-bottle homogeneity, taking into account standard deviation within one bottle.

Between-bottle homogeneity of the stabilized natural seawater and of Batch 1 of the artificial seawater was computed from standard deviation of single measurements made consecutively on three bottles of the same batch. It was conducted with the closed-cell multi-step potentiometric titration method at the SNAPO-$CO_2$.

The between-bottle homogeneity of the Batch 2 of the artificial seawater was obtained from the standard deviation of the mean TA values of three different bottles, themselves computed from at least three repeatability measurements. These measurements were made at LNE, using the open-cell multi-step potentiometric titration method.

The within-bottle homogeneity was computed, for the stabilized natural seawater and Batch 1 of the artificial seawater, from the square root of the mean of the variances obtained at LNE and MIO, from, respectively, three repeatability measurements made in one bottle. The within-bottle homogeneity of the Batch 2 of the artificial seawater was obtained from the square root of the mean of the variances obtained from repeatability measurements of the same three bottles used for between-bottle homogeneity assessment.

Stability

The stability of the stabilized natural seawater and Batch 1 of the artificial reference material, both bottled in ground-neck bottles, were followed by each participant to the inter-laboratory comparison over one year, with total alkalinity measurements performed every three months. For the results obtained at each deadline, Grubb's and Cochran's tests was applied to remove eventual outliers and the median of the remaining values were taken to establish the stability over time. The stability over time of the Batch 2 of the artificial seawater, bottled in glass bottles with screw caps, was followed at LNE on the same schedule.

The stability was established with a statistical Student test (t test) highlighting whether there is a significant trend in the evolution of the material or not (ISO 33405:2024, 2024). This test is based on the determination of the slope, noted b1, of the regression line of the TA values as a function of time. It computes $t_0$, defined as being the ratio of the slope to the standard deviation of the slope, noted sb1, and compares it to the threshold value $t_\alpha$ in the Student's table with n-2 degrees of freedom at a 95% confidence level (Linsinger et al., 2001).

Stability to transport was estimated as the discrepancy between measurements results obtained at LNE and MIO, LNE being the source laboratory of the artificial reference material and MIO the source of the natural seawater reference material. It is not computed for the second batch of artificial solution in screw cap bottles that was only

tested at LNE. This value represents a first estimate of short-term stability, while a proper evaluation according to ISO 17034 is still pending.

Dissolved nutrients of the stabilized natural seawater and of Batch 1 of the artificial seawater were also analyzed at the end of the stability study to highlight an eventual evolution.

### 3.1.4 Uncertainty estimation of the artificial reference material value

The uncertainty associated to the total alkalinity reference value of the artificial solution was obtained based on the ISO 33405:2024 (2024) and takes into account the uncertainties coming from the preparation and the

225 characterization $u_{charac}$, the homogeneity $u_{hom}$ and the stability $u_{stab}$ (Eq. 6).

$$u_{RM} = \sqrt{u_{charac}^2 + u_{hom}^2 + u_{stability}^2}$$ (6)

Preparation and characterization uncertainty

The uncertainty coming from the preparation and characterization was estimated based on Eq. 5 following the law of uncertainty propagation of the Guide to the expression of Uncertainty in Measurement (JCGM 100:2008, 2008).

The evaluation of the uncertainties of the different terms in Eq. 5 requires several steps of uncertainty determination.

> (1) The uncertainty of the amount contents of the stock solutions of NaHCO$_3$ and Na$_2$CO$_3$ ($v_{stock}$) were determined using uncertainty propagation for Eq. 7.

$$v_{stock} = \frac{m_{salt}*p*1000}{(m_{salt}+m_{H2O})*M_{salt}}$$ (7)

where $m_{salt}$ is the mass of salt of either NaHCO$_3$ or Na$_2$CO$_3$ salts (g), $m_{H2O}$ the mass of water (g), $p$ the purity of the salt and $M_{salt}$ the molar mass of the salt.

The uncertainties on the masses were obtained from the weighing scales calibration, the uncertainty of the purity was known from NMIJ coulometric characterization certificate and uncertainties on the molar masses were taken from IUPAC (Meija et al., 2016).

> (2) The uncertainty of the amount content of NaHCO$_3$ and Na$_2$CO$_3$ in the artificial reference material ($v_{stock_{RM}}$) was determined using the law of uncertainty propagation for Eq. 8.

$$v_{stock_{RM}} = \frac{m_{stock}*v_{stock}}{m_{total}}$$ (8)

where $m_{total}$ is the total mass of the reference material ($m_{stock.NaHCO3} + m_{stock.Na2CO3} + m_{NaCl} + m_{H2O}$), in g. The quantification of the uncertainties of masses and $v_{stock}$ are detailed above.

> (3) The uncertainty associated with the background alkalinity coming from the NaCl matrix also needs to be quantified, as it contributes to the TA value of the reference material ($TA_{background}$, Eq. 5). The amount content of NaCl introduced in the reference material solution and in each of the four solutions at different NaCl amount contents used to determine the background TA was obtained with Eq. 9, whose term's uncertainties quantification is detailed in the steps above.

$$v_{NaCl} = \frac{m_{NaCl}*1000}{m_{total}*M_{NaCl}}$$ (9)

The difference between measured and theoretical total alkalinity $\Delta(TA_{measured} - TA_{theoretical})$ was represented as a function of the amount content of the NaCl of the solutions used to study the background alkalinity, being respectively, 0, 1, 2 and 3 mol kg$^{-1}$ sol. The uncertainty estimate chosen to be attributed to the measured TA, pending a thorough assessment, was 2 µmol kg$^{-1}$. Indeed, this is approximately the precision reported to be

achievable in the literature for TA measurements. Since the same operator, instrument, and procedure were used to establish the relationship between $\Delta(TA_{measured} - TA_{theoretical})$ and $v_{NaCl}$, these parameters contribute to systematic uncertainty sources. They cancelled when establishing a trend and do not contribute to the uncertainty of the observed slope. The uncertainty of the slope is thus expected to be relatively low. The uncertainty of the theoretical total alkalinity was obtained using the law of uncertainty propagation in Eq. 4, whose term's uncertainties quantification are detailed in step (2) above.

The slope $b_b$ giving the evolution of $\Delta(TA_{measured} - TA_{theoretical})$ as a function of $v_{NaCl}$ was obtained by linear regression passing through the origin.

The uncertainty associated with this slope was obtained using the LNE-RegPoly software. LNE-RegPoly estimates a polynomial of degree k as $y = b_a + b_b x + b_c x^2 + ... + b_k x^k$ using n pairs of points (xi, yi), taking into account the uncertainties associated with these points. It then propagates the uncertainties from the points to the coefficients of the polynomial. A second uncertainty component was added to this uncertainty to take into account the fact that the regression is forced to pass through the origin. Indeed, the residuals were thus slightly bigger. To do so, the standard deviations of slopes (i) with regular linear regression and (ii) forced to pass through the origin, were computed from knowledge of the residuals of the regressions. The difference of the standard deviations of regressions (i) and (ii) was added as an uncertainty component of the slope $b_b$.

A second approach based on weighted orthogonal distance regression was applied with help of statisticians to compute the uncertainty of the slope $b_b$ (Boggs et al., 1992). This approach yielded a slightly lower uncertainty. The first approach, described above, was adopted as it is the more conservative estimate of the uncertainty of the slope.

Knowing the uncertainties of, respectively, the slope $b_b$ and the amount content of NaCl in the reference material solution ($v_{NaCl_{RM}}$), the uncertainty on the background total alkalinity is obtained by propagation in Eq. 10.

$$TA_{background} = b_b * v_{NaCl_{RM}} \tag{10}$$

(4) The final step is to propagate the uncertainties quantified in steps (2) and (3) in the Eq. 5, giving the total alkalinity of the reference material.

Homogeneity uncertainty

The within and between bottles homogeneities were assessed from the homogeneity study described in Sect. 3.1.3. As the SNAPO-CO$_2$ cannot perform several measurements per bottles, needed to perform one-way ANOVA analysis, homogeneities uncertainties were computed from measurements made at LNE and MIO for the batch 1 and at LNE only for the batch 2 of the artificial reference material.

This study highlighted that the robustness of the determination of the homogeneity is highly dependent on the variability of the measurement method. It was chosen to neglect the within-bottle homogeneity component, supposed to be negligible, in the homogeneity uncertainty quantification. Indeed, uncertainty resulting from within-bottle homogeneity can usually be neglected for liquid reference materials. The uncertainty relative to homogeneity was obtained, for the evaluation of the between-unit term, from Eq. 11 (ISO 33405:2024, 2024). It is computed for each batch respectively.

$$u_{hom} = \sqrt{\frac{M_{between} - M_{within}}{n_0}} \tag{11}$$

where $M_{between}$ and $M_{within}$ are, respectively, between- and within-bottle mean square term from analysis of variance (ANOVA) and $n_0$ the number of replicates per bottle (i.e. 3).

   Stability uncertainty

The uncertainty on the stability is obtained from Eq. 12 if no significant trend is established by the t tests described in Sect. 3.1.3, and by Eq. 13 if a significant trend is established, where the added term corresponds to the estimated degradation of the material. These equations are introduced below.

$$u_{stab} = s_{b1} \cdot t_m \tag{12}$$

$$u_{stab\prime} = \sqrt{(\frac{b1 \cdot \frac{t_m}{2}}{\sqrt{3}})^2 + (s_{b1} \cdot t_m)^2} \tag{13}$$

where $b_1$ corresponds to the slope of the evolution over time, $s_{b1}$ corresponds to the slope standard deviation and $t_m$ to the time ($t_m$ = 3 months).

The assessment of stability to transport showed no significant discrepancy between the source and recipient laboratory, this source of uncertainty was thus neglected in the uncertainty budget.

**3.2 Results**

**3.2.1 Characterization**

Two reference materials have been produced, an artificial seawater (2 batches) and a stabilized natural seawater. Table 2 presents the characteristics of these two reference materials, established following the methods described in Sect. 3.1.2.

**Table 2: Characteristics of the produced reference materials for total alkalinity measurements**

| | **Artificial solution** | | **Stabilized natural seawater** | |
|---|---|---|---|---|
| | **Batch 1 (Ground neck bottles)** | **Batch 2 (Screw caps bottles)** | Practical salinity[b] | 38.533 |
| Absolute salinity[a] | 35.189 | 35.184 | Silicates | 12.37 |
| Ionic strength | 0.623 | 0.623 | Nitrites | 0.02 |
| $TA_{background}$ (µmol kg$^{-1}$) | 3.53 | 3.53 | Dissolved nutrients (µmol l$^{-1}$)     Phosphate | 0.40 |
| Reference Total Alkalinity value (µmol kg$^{-1}$) | 2503.64 | 2503.78 | Nitrates & nitrites | 9.08 |

[a] The absolute salinity was calculated based on the composition of the solution (g of dissolved salts per kg of solution).

[b] The practical salinity was measured with a salinometer and is based on a conductivity ratio.

Reference values of total alkalinity for the two batches of artificial reference materials are computed from Eq. 5, giving values of respectively 2503.64 and 2503.78 µmol kg$^{-1}$. The close similarity of these values is evidence of the good reproducibility of the preparation of the batches. The background alkalinity has been quantified, following the method described in Sect. 3.1.2, to be 3.53 µmol kg$^{-1}$ for both batches, and is included in the reference

values given above, meaning $TA_{background}$, coming from the NaCl matrix, is a contribution to the TA value of the reference material (Eq. 5). Figure 1 represents the results of the quantification of the background alkalinity.

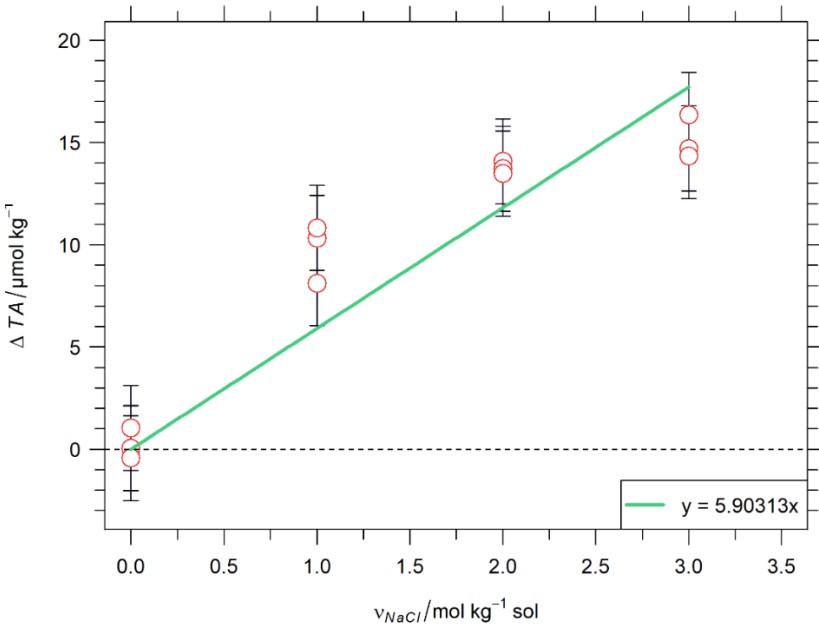

**Figure 1: Quantification of background alkalinity originating from the NaCl matrix for the artificial reference material. $\Delta TA$ represents the difference of total alkalinity between the measured and the theoretical value calculated with Eq. 4. Red circles represent single TA potentiometric measurements with error bars representing their standard uncertainty, and green straight line represents the slope obtained from linear regression passing through the origin.**

**3.2.2 Homogeneity, stability studies and uncertainty quantification**

Homogeneity and stability studies results

Table 3 presents the results of the homogeneity and stability assessments as described in the Sect. 3.1.3.

**Table 3: Results obtained from the homogeneity and stability assessments of the reference materials developed.**

|  | Artificial solution | | Stabilized natural seawater |
|---|---|---|---|
|  | Batch 1 (Ground neck bottles) | Batch 2 (Screw caps bottles) |  |
| Homogeneity - standard deviation |  |  |  |
| Between-bottle ($\mu$mol kg$^{-1}$) | 1.0820 | 0.9915 | 1.6100 |
| Within-bottle ($\mu$mol kg$^{-1}$) | 1.2190 | 1.3576 | 1.8418 |
| Stability over time |  |  |  |
| Slope (b1) ($\mu$mol kg$^{-1}$ month$^{-1}$) | 0.8065 | 0.9325 | 0.2219 |
| Slope standard deviation (Sb1) ($\mu$mol kg$^{-1}$ month$^{-1}$) | 0.2735 | 0.1341 | 0.4366 |
| $t_0$ (b1 / Sb1) | 2.9485 | 6.9528 | 0.5081 |
| $t_\alpha$ (Student n-2) | 4.3027 | 3.1824 | 3.1824 |
| Stability to transport – standard deviation |  |  |  |
| ($\mu$mol kg$^{-1}$) | 1.2052 | / | 1.1781 |

The between and within bottles standard deviations are in the range 1.0 – 1.8 µmol kg$^{-1}$, and seem to be slightly greater for the natural seawater than for the artificial solutions.

Assessments of stability over time show a significant trend (t0 > tα) only for Batch 2 of the artificial reference material. Its stability has been studied up to fourteen months after the preparation, however, the significant trend, and thus instability of the material, was already established after eleven months (Figure 2).

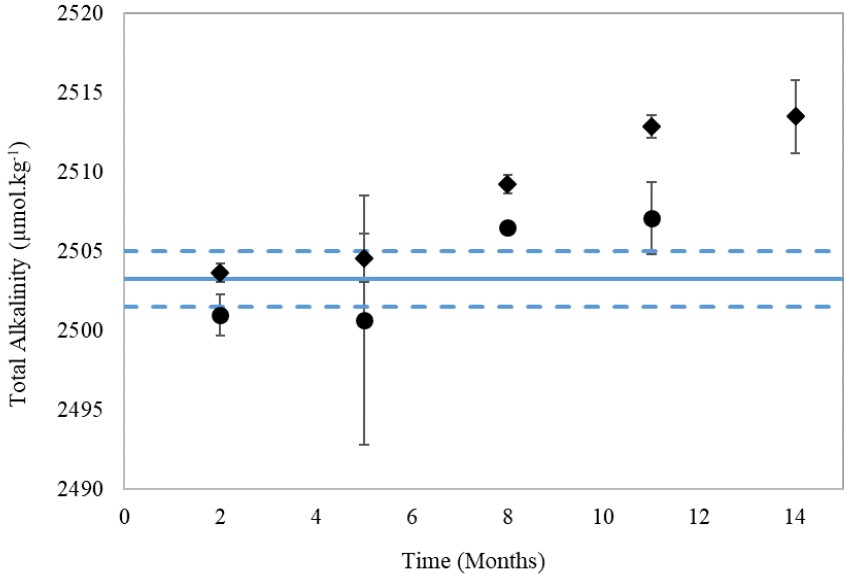

**Figure 2: Stability study of the artificial solution – Batch 1 (●) and Batch 2 (♦). ● symbols represent the median TA value obtained by the participants from replicates made in a bottles of Batch 1, with error bars indicating the corresponding standard uncertainty of the median. ♦ symbols represent the mean TA value obtained at LNE from replicates in a single bottle of Batch 2, with error bars indicating the corresponding standard deviation. The solid line**
**represents the assigned reference value of Batch 2, and the dotted lines indicate its expanded uncertainty ($k = 2$, level of confidence of approximately 95%).**

The stability study of the stabilized natural seawater and of the 1$^{st}$ batch of artificial seawater doesn't show a significant trend, which seems to indicate a better stability. The stability study on the Batch 1 of the artificial solution had to be interrupted due to lack of remaining bottles to pursue the study, it was conducted up to eleven
345     months. The stability study of the natural solution was conducted up to fourteen months after bottling. The detail of the alkalinity values used to establish the stability of the materials are given in Appendix C1.

The stability to transport is negligible (discrepancies are in the level of within and between bottles homogeneities reported) and is thus not taken into account in the final uncertainty budget. It is expected that the second batch of artificial solution behaves similarly to transportation as the other materials. Thus, the uncertainty of the second
batch is also computed, even if stability to transport study wasn't performed.

Uncertainty budget of the artificial material reference values

Table 4 presents the uncertainty budget attributed to the reference values of the artificial reference materials as detailed in Sect. 3.1.4.

**Table 4: Uncertainties involved in Eq. 6 for the assessment of total alkalinity reference value's uncertainty of the artificial reference material**

| | Batch 1 (Ground neck bottles) | Batch 2 (Screw caps bottles) |
|---|---|---|
| Characterization and preparation | | |
| $v_{NaHCO_3}$ ($\mu$mol kg$^{-1}$) | $5.11 \times 10^{-1}$ | $8.06 \times 10^{-1}$ |
| $v_{Na_2CO_3}$ ($\mu$mol kg$^{-1}$) | $5.53 \times 10^{-2}$ | $7.77 \times 10^{-2}$ |
| Background TA ($\mu$mol kg$^{-1}$) | $3.09 \times 10^{-1}$ | $3.09 \times 10^{-1}$ |
| Combined u | $6.00 \times 10^{-1}$ | $8.67 \times 10^{-1}$ |
| Homogeneity | $4.81 \times 10^{-1}$ | $6.07 \times 10^{-1}$ |
| Stability | $8.21 \times 10^{-1}$ | $9.02 \times 10^{-1}$ |
| Total ($u_{RM}$) | | |
| u ($k$=1) | 1.12 | 1.39 |
| U ($k$=2) | 2.25 | 2.78 |

The standard uncertainties of the terms in equations 5 are assessed following the method given in Sect. 3.1.4 and are given in Table 4, corresponding to the "characterization and preparation" uncertainty. The difference of uncertainty coming from the preparation and characterization between the two batches is explained by the fact that the volume of the Batch 1 is higher, allowing to reduce the relative uncertainty of gravimetric preparation.

The uncertainty of stability of the Batch 1 and Batch 2 of the artificial reference material are computed from Eq. 12 and 13, respectively, using the t-test information given in Table 3 and for a material shelf life of 3 months (chosen as the ILC was conducted within 3 months after the preparation of the solutions).

The standard uncertainty attributed to the TA reference values of Batch 1 and Batch 2 of the artificial reference material are of 1.12 and 1.39 µmol kg$^{-1}$, respectively, and were computed from Eq. 6. This gives a global expanded uncertainty budget (i.e. with a coverage factor, $k$, of 2, and a level of confidence of approximately 95%) of respectively 2.25 and 2.78 µmol kg$^{-1}$.

## 3.3 Discussion

### 3.3.1 Composition of the artificial solution

The artificial solution developed for use as a reference material for total alkalinity measurements differs slightly from synthetic solution described in the literature for accuracy checking. Previous studies by Dickson et al. (2003) used solutions composed of either sodium carbonate, TRIS, or borax in an NaCl matrix. In this study, a decision was made to use a combination of sodium carbonate and sodium bicarbonate for two main reasons: (1) to mimic the alkalinity source found in natural seawater, which primarily arises from carbonate and bicarbonate ions, and (2) to achieve a pH level representative of seawater, ensuring that the potential measured by the glass electrode during titration is similar to that of a natural sample.

For natural seawater, knowledge of salinity is required to determine the TA value through NLLS regression, which accounts for the competing acid–base equilibria present in seawater. In the artificial solution, the addition of NaCl to the solution background helps maintain an ionic strength similar to that of natural seawater aiming to mimic any

potential dilution effect caused by the addition of HCl; although this effect is most likely negligible (Okamura et al., 2014). However, it introduces background alkalinity, which complicates the determination of the total alkalinity reference value of the material. Moreover, the zero level of protons defined to measure TA in seawater is based on seawater acid-base chemical equilibria (Schulz et al., 2023). The impact of using a reference material with a simplified matrix must be further investigated, even though the good results obtained in the ILC seem to show that the artificial RM could be adequate.

For the artificial solution, the practical salinity was also measured with a salinometer, highlighting a difference of +2.05 compared to the absolute salinity obtained from knowledge of the composition. The significant discrepancy between both comes from the difference in definition. Absolute salinity is defined as the mass fraction of dissolved material in solution while the practical salinity is defined as the ratio of the conductivity of the solution on the conductivity of a standard KCl solution. For a natural seawater, the discrepancy between both is in the order of 0.2 (Pawlowicz, 2013). The composition of the artificial seawater being composed in high majority of NaCl, may explain the higher discrepancy between practical and absolute salinity observed. The definition of absolute salinity may not be adequate for the artificial solution, where it is probably more relevant to compare ionic strengths, which should be of 0.7 mol kg$^{-1}$ for a salinity 35. For the natural solution, only the practical salinity is given due to a lack of knowledge of its exact composition.

Developing an artificial reference material also served as a step forward in eliminating the use of mercuric compounds, which are currently employed to inhibit the growth of microorganisms in natural seawater solutions. This is however tied to the stability of the RM.

### 3.3.2 Reference value determination

The total alkalinity reference value is determined through the gravimetric preparation of the solution, which is composed of salts previously characterized in terms of base amount content via coulometric analysis. Both the gravimetry and coulometry methods provide SI traceable results. Since the background alkalinity ($TA_{background}$) is carefully quantified, along with its associated uncertainty, the reference TA value assigned to the material may also be considered traceable to the SI units. It however necessitates that NaCl is the only significant source of background alkalinity and the purity of NaHCO$_3$ is correctly assessed.

The determination of background alkalinity resulting from NaCl impurities involves the gravimetric preparation of artificial solutions with varying NaCl concentrations, coupled with TA potentiometric measurements. This approach establishes a relationship between NaCl concentration and background alkalinity, along with its uncertainty (as described in Eq. 10). The linear regression was forced to pass through the origin as the background alkalinity coming from NaCl impurities is zero for a solution without NaCl matrix, this is confirmed by the measurements made with $v_{NaCl} = 0$ mol kg$^{-1}$ (Fig. 1). From all measurements presented in Fig. 1, the intercept would be 1.96 µmol kg$^{-1}$ if it was not forced to zero. However, as both NaHCO$_3$ and Na$_2$CO$_3$ salts were characterized by coulometry at NMIJ in term of base amount contents, no background alkalinity should exist in these salts.

Initially, an uncertainty of 2 µmol kg$^{-1}$ was chosen for the TA potentiometric results to quantify $TA_{background}$, which closely matches the final estimated uncertainty obtained through the bottom-up approach (2.63 µmol kg$^{-1}$). Attributing this final uncertainty to the TA measurements for determining $TA_{background}$ would only marginally increase the uncertainty of the reference value of the material by 1 to 2%, which is negligible. Additionally,

considering the presence of systematic errors between measurements (such as the same operator, device, method, acid titrant, etc.), 2 µmol kg$^{-1}$ can be deemed a realistic uncertainty for this purpose.

However, the linear behaviour can be questioned when looking at Fig. 1. The determination of $TA_{background}$ will necessitate further rigorous investigation, although the method presented here is deemed a first estimate.

To simplify the determination of the TA reference value, it may be considered to purify the NaCl before use, as suggested by Dickson et al. (2003), who estimated that the resulting $TA_{background}$ could be negligible. Alternatively, removing completely the NaCl background may be a final solution but it should first be investigated whether the difference in salinity with natural samples would introduce other mechanisms that could hinder the artificial solution's applicability for quality control.

### 3.3.3 Other considerations for the artificial RM

Having a reference material such as the developed artificial solution, with a for a SI traceable reference value provided alongside a comprehensive uncertainty budget, unlike the natural seawater reference material, offers several advantages:

      (1) It enables the validation of experimental protocols and the Gran's determination of the TA value.

(2) It facilitates the control of device accuracy

      (3) It assists in qualifying new operators

      (4) It allows for the quantification of acid titrant amount content for laboratories that do not have access to coulometric methods.

Another benefit of having an artificial reference material is the ability to provide reference materials for a wide

range of total alkalinity values. This is important for end-users to verify that there isn't a bias in the measurement method across the studied range of alkalinity.

However, the different chemical composition compared to natural seawater prevents from using the nonlinear least-squares regression method, which is yet widely applied to natural seawater samples to correct the value considering the acid-base system in the solution. Therefore, it is highly recommended to distribute a second

material being a stabilized seawater, as the one from Scripps or the one developed in this study (i.e. a stabilized natural seawater), to ensure the comparability of TA measurements on natural seawater samples.

Having a natural seawater reference material that is easy to collect during open ocean oceanographic cruises also offers the availability of a reference material that can be a bit cheaper and in bigger volume than the artificial one, which has to be produced in the lab from high purity compounds. The artificial material could serve as a reference

material for validating the measurement method of oceanographic institutes prior to assigning a reference value to the natural seawater. This natural seawater reference material could also be sent to reference laboratories (e.g., National Metrology Institutes) for characterization.

### 3.3.4 Homogeneity and stability of the materials

The homogeneity of the material was assessed through potentiometric total alkalinity measurements. However,

due to the method's limited repeatability, it was challenging to detect significant within- and between-bottle inhomogeneity (Table 3). This was confirmed by the statistical test described in ISO 33405:2024 (Section 7.5.1 of the ISO), which states that the repeatability standard deviation of the homogeneity study procedure should be less than one third of the target standard uncertainty of the TA measurement result for the procedure to be considered

suitable. In this case, the criterion was slightly exceeded (i.e. $M_{within}$ is of approximately 1.5 µmol kg$^{-1}$) , but the results can nevertheless be regarded as a preliminary estimate of the material's homogeneity. Furthermore, it is anticipated that the homogenization step during material preparation ensures sufficient homogeneity for the intended use. Only the estimation of between-bottle homogeneity was included in the uncertainty budget of the TA reference value of the artificial materials, accounting for 6 to 7% of the final budgets.

The stability studies presented in Table 3 revealed a significant trend in Batch 2 of the artificial solution bottled with screw caps borosilicate bottles, indicating instability. Conversely, Batch 1 of the artificial solution and the stabilized natural seawater, both bottled in ground neck borosilicate bottles, did not exhibit significant instability based on the t-test results (Table 3). This suggests potential better stability with the latter bottles. However, it should be noted that the standard deviation of corresponding slopes is not negligible compared to the slopes themselves. Further stability studies with longer durations could provide clearer insights into the stability of Batch 1 of the artificial solution and the stabilized natural seawater.

Dissolved inorganic carbon (DIC) analyses conducted at SNAPO-CO$_2$ on the natural seawater solution after 2, 8, and 11 months since bottling indicated a mean DIC value of 2385 µmol kg$^{-1}$ with no significant evolution over time, meaning the carbon content was most probably stable. These results are presented in Appendix C2. Consequently, any instability of the RM would result from sources other than carbon.

Nutrients analysis performed at MIO revealed an increase in silicate content from 12.1 to 23.2 µmol kg$^{-1}$ for the stabilized natural seawater in the ground neck bottles, representing an increase of about 11 µmol kg$^{-1}$ between months 2 and 14 post-bottling (Appendix C3). Additionally, nutrient analysis on Batch 2 of the artificial seawater indicated a silicate content of 26.5 µmol kg$^{-1}$ 14 months after preparation, despite an initial supposed content of zero. This suggests a release of silicate ions from the borosilicate glass containers, as previously suggested by Mos et al. (2021). The release appears to be more significant from glass bottles with screw caps, which may align with the findings that Batch 2 of the artificial solution lack stability. This may be due to the difference in glass supplier, although all bottles were borosilicate 3.3. However, the fact that silicates are also released in glass bottles with ground necks suggests that Batch 1 of the artificial solution and stabilized natural seawater may also lack stability. The increase in alkalinity measured by potentiometric measurements for both solutions is different than the amount of silicates released, indicating potential secondary processes influencing alkalinity (e.g. biological activity, pollution, other ion exchange processes with the glass). Improving the stability of the developed materials likely necessitates using different bottling methods, such as employing glass with specific treatments to prevent silicate release. The reference materials distributed by Scripps Institution of Oceanography are known to be more stable, possibly due to differences in glass bottle suppliers. It could also be worthwhile to test storing solutions in polypropylene bottles, as investigations by Mos et al. (2021) suggested better stability, although this may compromise eventual stable DIC values simultaneously.

**4 Inter-laboratory comparison**

**4.1 Materials and Methods**

**4.1.1 Protocol of the inter-laboratory comparison**

The inter-laboratory comparison (ILC) was conducted with five French laboratories conducting seawater total alkalinity measurements with the standardized measurement method, being the multi-step potentiometric measurement method (Dickson et al., 2007 - SOP 3a & 3b).

Figure3 gathers information about the participants to the ILC, including affiliation and measurement methods.

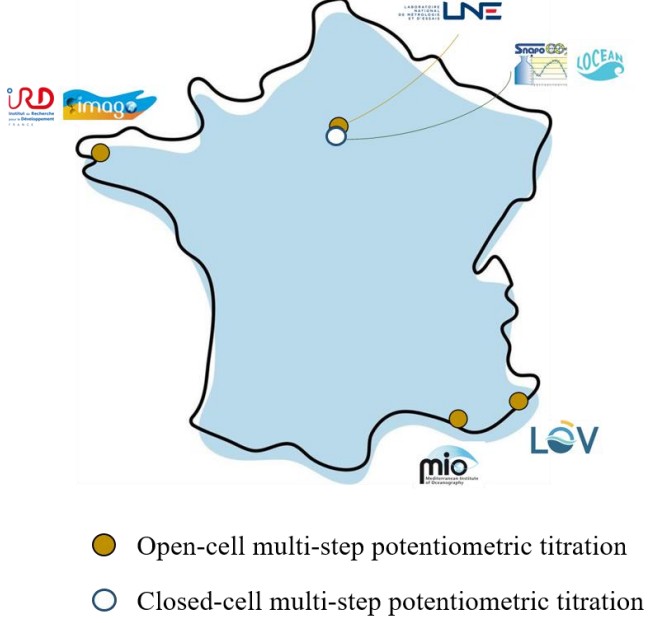

⬤  Open-cell multi-step potentiometric titration

◯  Closed-cell multi-step potentiometric titration

**Figure 3: Information concerning the participants to the inter-laboratory comparison**

The total alkalinity measurements were performed by each participant on the two reference materials developed: the stabilized natural seawater, and the artificial seawater of known TA. The preparation of the artificial seawater and the bottling for the natural seawater were made approximately at the same time. The measurements were

performed two months after the preparation of these materials, and all the participants completed their measurements within a two weeks period. Each participants performed three repeatability measurements, within one bottle for LNE, MIO, IMAGO and LOV, and using three bottles for SNAPO-CO$_2$ as the closed-cell measurement method requires a larger volume of sample (around 500 ml).

The amount content of the hydrochloric acid used by MIO, SNAPO-CO$_2$, IMAGO and LOV was characterized

based on the analyses of a total alkalinity reference material purchased from Scripps Institution of Oceanography. LNE's hydrochloric acid was a Standard Reference Material purchased from SMU (Slovack metrology institute, $v = 1$ mol kg$^{-1}$) diluted gravimetrically.

As the artificial reference material didn't contain sulphate and fluoride ions, the data treatment applied was Gran's regression method. However, for the natural seawater samples, a correction computed from the nonlinear least-

squares regression method was applied to take into account for the matrix.

In addition, the SNAPO-CO$_2$ used the titration data from the closed-cell multi-step potentiometric titration to compute dissolved inorganic carbon values for the stabilized natural seawater reference material.

### 4.1.2 Treatment of the inter-laboratory comparison results

The median of the set of means (from repeatability measurements made by each participant) was calculated for

two different materials—natural seawater and the artificial RM (Batch 1). The uncertainty associated to the median was calculated with the method given in Muller (2000).

The Cochran's and Grubbs' statistical tests were used to identify possible isolated values or outliers due to, respectively, intra-laboratories variances and discrepancy to the mean observation. These values were removed from the ILC results treatment used to quantify trueness and precision of the method. The bias between the mean

obtained on the artificial seawater and the reference value obtained with Eq. 5, gives information on the trueness of the measurement method. The precision was computed from the combination of intra- and inter-laboratory variances obtained for the two materials.

As all participants realised measurements on the two different solutions developed, the ILC results were presented under a Youden plot (Youden, 1959; ISO 13528, 2022,). The methodology of the Youden plot is as follows:

- The results of each participant is represented on a graph by a single point $(X_l, Y_l)$, representing its mean TA value obtained on the natural seawater in the x-axis $(X_l)$ and its mean TA value of the artificial seawater in the y-axis $(Y_l)$.
- The medians obtained on the two material are drawn as the centroid (X,Y). Vertical and horizontal lines are drawn from that centroid, representing x-axis and y-axis medians (i.e. natural seawater and artificial

seawater median values, respectively).
- A 45° line passing through the centroid (X,Y) is then drawn.
- A 95% confidence circle is finally represented from the centroid with a radius, $r$, calculated with Eq. 14.

$$r = \sqrt{\frac{\sum_i^p ((X_l - Y_l) - \overline{(X_l - Y_l)})^2}{2*(p-1)}} * 2.448 \tag{14}$$

where $\overline{(X_l - Y_l)}$ represents the mean difference $(X_l - Y_l)$ from the five participants, p is the number of participants

and 2.448 is the factor allowing to obtain a circle with a 95% confidence level (Youden, 1959).

Having a data point $(X_l, Y_l)$ outside of the confidence circle is considered as having biased results. The shortest distance of $(X_l, Y_l)$ to the 45° line is proportional to random errors for the concerned laboratory, while the distance between that point on the 45° line to the centroid is proportional to systematic errors (Martín et al., 2017).

### 4.2 Results

Figure 4 represents, following the method detailed in Sect. 4.1.2, the Youden plot obtained from the results of the inter-laboratory comparison conducted with five laboratories (Fig. 3) for total alkalinity measurements on the two reference materials. The total alkalinity values obtained by each participant are presented in Appendix C1, corresponding to the measurements made at the first deadline.

The median value obtained on the stabilized natural seawater is of 2581.43 ± 2.19 µmol kg$^{-1}$ ($k$=2). The median

obtained on the artificial solution is of 2501.61 ± 2.78 µmol kg$^{-1}$ ($k$=2).

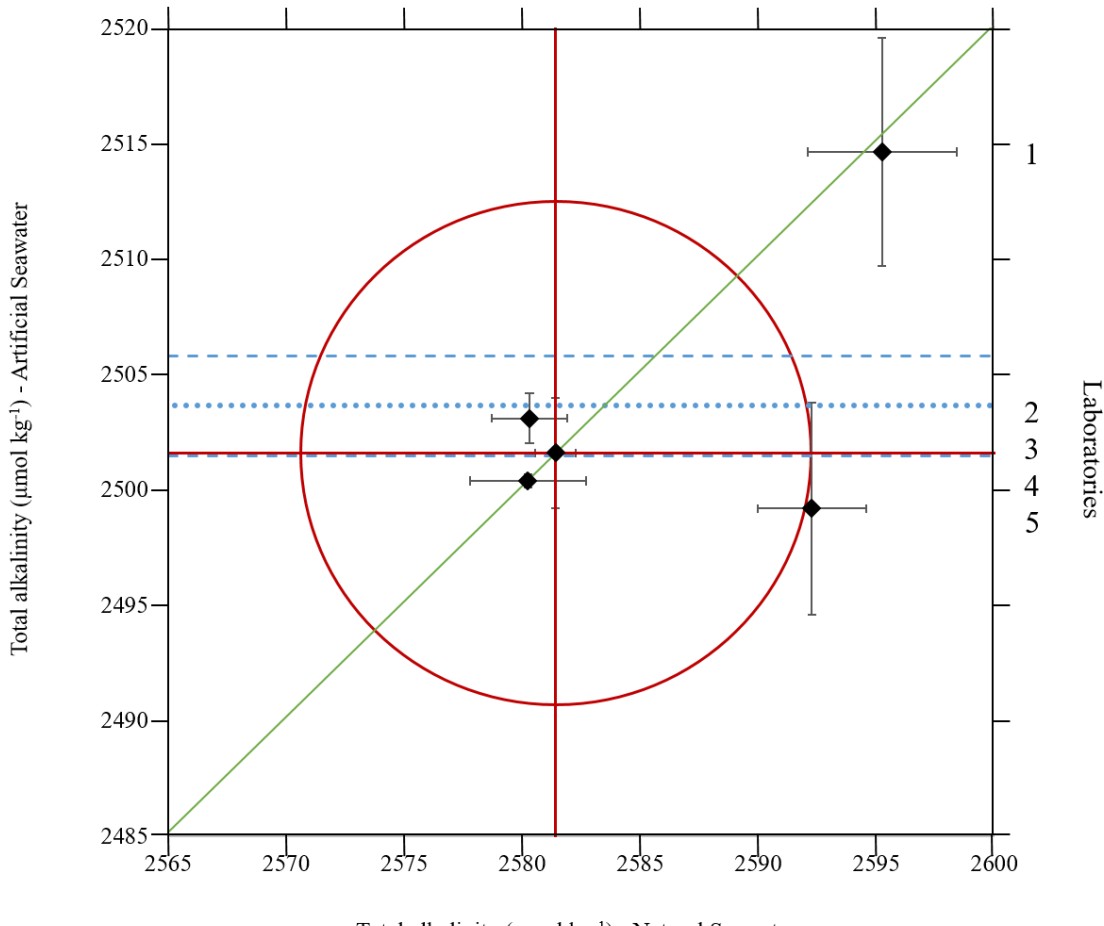

**Figure 4: Youden plot of the inter-laboratory comparison conducted on the Stabilized natural seawater (X-axis) and the Artificial seawater, Batch 1 (Y-axis). Diamonds correspond to the data point ($X_l$, $Y_l$) of each participant, green straight line corresponds to the 45° line graphical marker, the two straight red horizontal and vertical lines represent the median of total alkalinity results obtained from the five laboratories on the artificial solution and the stabilized natural seawater, respectively, and the red circle represents the 95% confidence level circle. The blue dotted line represents the artificial reference material reference value together with its expanded uncertainty in blue dashed lines.**

The Youden plot shows that three laboratories out of five obtained really close results that are centred around the medians of the two samples (laboratories, 2, 3 and 4). The standard deviation (represented by error bars) obtained by the laboratory 5 on the natural seawater shows that this laboratory obtained results that could be considered compatible according to the 95% confidence circle. Moreover, its error bars obtained on the artificial reference material are compatible with the TA reference value. The laboratory 1, on the other hand, is considered as having biased results according to the 95% confidence circle drawn on the plot. Its proximity with the 45° line clearly indicates that the source of error is systematic (i.e. similar bias to the medians for the two samples).

The value of the laboratory 1 obtained on the artificial seawater was also isolated by Grubb's test. Laboratory 1 reported a problem in the acid injection system at the time of the ILC (see Sect. 4.3). This laboratory was thus not taken into account in the computation of the precision and bias of the measurement method.

The bias between the reference value and the mean value obtained by the four participants selected for the artificial solution is of -2.56 ± 2.86 µmol kg$^{-1}$ ($k$=2). The precision of the method, given by the computation of $s_L$ and $s_r$, being, respectively, inter and intra laboratory standard deviations, is of 1.99 µmol kg$^{-1}$ on the artificial seawater. This precision reaches level of precision reported in the literature (Millero et al., 1998; Bockmon and Dickson,

2015). However, the precision on the natural seawater is of 5.85 µmol kg$^{-1}$, which mostly arises from the laboratory 5 random error highlighted by the Youden plot in Fig. 4.

**4.3 Discussion**

The results of the inter-laboratory comparison indicate an acceptable agreement among four out of five participants, even though the random error highlighted for laboratory 5 seems to affect the estimated precision of the method. Also, laboratory 1 exhibits a systematic error in Fig. 4, evidenced by an equivalent bias compared to the median for both analysed samples: 13.64 µmol kg$^{-1}$ for the artificial solution and 13.82 µmol kg$^{-1}$ for the stabilized natural seawater. Laboratory 1 later reported that the acid injection system was not functioning properly 580    during the ILC. A leak at the microvalve, leading to inconsistent acid delivery volumes, could explain the observed bias. To address this issue, laboratory 1 subsequently removed the microvalve and replaced it with a microcapillary, which seemed to improve both the precision and accuracy of the measurements.

More broadly, the results of this inter-comparison suggest that:

(1) The TA reference value assigned to the artificial material based on knowledge of its composition aligns
well with TA values measured using the standardized potentiometric titration method. The bias between the two, being of -2.56 ± 2.86 µmol kg$^{-1}$ ($k$=2), falls within the expanded uncertainties of the bias itself.

(2) In the case of a laboratory showing a systematic bias, the artificial material with a well-characterized reference value might be employed to correct potentiometric measurements by calculating the trueness bias, however this needs further careful investigation.

(3) The closed-cell measurement method, being employed by one of the laboratories among laboratories 2, 3, and 4 in Fig. 4, yields results that are compatible with those obtained using the open-cell measurement method.

These seem to demonstrate that both developed materials could be considered suitable for use in quality controls of the multi-step potentiometric titration method for total alkalinity measurements in seawater.

**5 Uncertainty estimation of the total alkalinity measurement results**

**5.1 Materials and Methods**

**5.1.1 Top-down approach**

The uncertainty of the total alkalinity measurement results was established using the top-down approach (i.e. using experimental data from the inter-laboratory comparison exercise), following the ISO standard 21748 "Guidance
for the use of repeatability, reproducibility and trueness estimates in measurement uncertainty evaluation" (2017). This document gives guidance for the evaluation of measurement uncertainties using data obtained from studies conducted in accordance with NF ISO 5725-2: basic method for the determination of repeatability and reproducibility of a standard measurement method. The uncertainty budget was calculated using Eqs. 15, 16 and 17, presented in the ISO standard 21748, giving the general expression for combined standard uncertainty.

$u^2(y) = s_L^2 + s_r^2 + u^2(\hat{\delta})$                                                        (15)

where $u(y)$ is the estimated measurement result uncertainty, $s_L$ the inter-laboratory standard deviation (i.e. the standard deviation of the mean values obtained by each laboratory), $s_r$ the intra-laboratory standard deviation

divided by the square root of the mean number of replicates (Eq. 16), and $u(\hat{\delta})$ the uncertainty associated to the estimated bias of the measurement method (Eq. 17).

$$610 \qquad s_r^2 = \left(\frac{s_s}{\sqrt{n}}\right)^2 \qquad\qquad\qquad (16)$$

where $s_s$ is the standard deviation of the replicates standard deviations obtained by each laboratory and n is the number of replicates in each laboratory (n=3).

$$u^2(\delta) = \frac{(s_L^2 + s_r^2) - \left(1 - \frac{1}{n}\right)s_r^2}{p} + u^2{}_{RM} \qquad\qquad (17)$$

where p is the number of laboratories and $u_{RM}$ the standard uncertainty of the certified reference value (obtained
with Eq. 6).

As the uncertainty estimates require information about trueness of the method, only the measurement results carried out on the artificial seawater, having a characterized reference value, can be used. Thus, the final uncertainty budget can be attributed to the results obtained with the multi-step potentiometric titration method using the Gran's data treatment. Uncertainties coming from the nonlinear least-squares regression data treatment
are not included in the budget.

**5.1.2 Bottom-up approach**

The bottom-up approach for the establishment of the uncertainty budget of the total alkalinity measurement results is hereafter detailed for the open-cell multi-step potentiometric titration method applied at LNE and Gran's data treatment, following the Guide to Uncertainty in Measurement (JCGM 100:2008, 2008). Details of the
instrumentation for TA measurements performed at LNE are given in Appendix B.

The bottom-up approach involves several steps: the establishment of the measurement model, the uncertainty quantification of the input variables, the identification of possible covariances, the propagation of the uncertainty through the established model and the final expression of the results.

Establishment of the measurement model
The measurement model was established from the measurement and Gran's data treatment methods presented in Sect. 2, represented by Eqs. 2 and 3.

Identification and uncertainty quantification of input variables

Table 6 gathers all the input variables identified in the measurement model presented above. It also presents the sub-sources of uncertainty influencing these input variables and the method allowing their uncertainty
quantification. The sub-sources identification together with their uncertainty quantification methods are based on the procedure conducted at LNE and instrumentation available there, as described in Sect. 2 and Appendix B.

Uncertainty propagation and expression of the results

The final uncertainty on TA results was obtained using the law of uncertainty propagation in Eq. 2. The use of the software LNE-RegPoly for the quantification of the uncertainties of coefficients $a$ and $b$ coming from the linear
regression of F1 in function of $m_{HCl}$ allowed demonstrating that these two terms are highly correlated. The factor of correlation between $a$ and $b$, quantified by the software, was integrated in a correlation matrix introduced in the process of uncertainty propagation using partial derivatives.

The final uncertainty is expressed as expanded uncertainty using a coverage factor, $k$, of 2, corresponding to a confidence level of approximately 95%.

## 5.2 Results

Table 5 presents the uncertainty budget of the total alkalinity measurement results obtained from the inter-laboratory comparison, i.e. following the top-down approach, as described in Sect. 5.1.1. The uncertainty budget corresponds to the multi-step potentiometric measurement method and Gran's data treatment, as it was computed from measurements made on the artificial solution. As laboratory 1 was isolated, the uncertainty budget is obtained from results of the four remaining laboratories. The uncertainties attributed to inter and intra laboratory variation are, respectively, 1.67 and 1.09 µmol kg$^{-1}$. The uncertainty attributed to the bias is of 1.43 µmol kg$^{-1}$. The global standard uncertainty budget is thus of 2.45 µmol kg$^{-1}$; giving an expanded uncertainty of 4.90 µmol kg$^{-1}$.

**Table 5: Uncertainty budget of total alkalinity measurement results computed with the top-down approach. $s_L$ and $s_r$ represent respectively inter and intra laboratory variation, $u(\hat{\delta})$ the uncertainty of the bias and $u(y)$ the global standard uncertainty.**

| | Standard uncertainty estimation |
|---|---|
| Uncertainty sources | u ($k$=1) µmol kg$^{-1}$ |
| $s_L$ | 1.67 |
| $s_r$ | 1.09 |
| $u(\hat{\delta})$ | 1.43 |
| $u(y)$ | 2.45 |

Table 6 presents the uncertainty quantification of all input variables involved in the measurement model of the multi-step open-cell potentiometric titration procedure with Gran's data treatment. The uncertainty quantification is detailed in Sect. 5.1.2. Details on the uncertainty propagation, obtained with in-house LNE software WINCERT, is given in Appendix D. The overall total alkalinity uncertainty budget gives a standard uncertainty of 2.63 µmol kg$^{-1}$, thus an expanded uncertainty (i.e. with a coverage factor, $k$, of 2, and a confidence level of approximately 95%) of 5.26 µmol kg$^{-1}$.

**Table 6: Quantification of the uncertainty sources involved in the Total Alkalinity measurement method and Gran's data treatment following the bottom-up approach.**

| Input variables | Definition | Sub-sources of uncertainty | Quantification method | Standard uncertainty ($k=1$) | |
|---|---|---|---|---|---|
| | | | | Sub-sources | Combined u |
| $m_{HCl}$ (g) | Mass of HCl delivered during the titration | -HCl density : Densimeter and temperature of the acid accuracies (g/cm3 and °C, respectively) | -Densimeter specification | $2.00 \times 10^{-5}$ | $\mathbf{2.89 \times 10^{-3}}$ |
| | | | -Temperature probe calibration certificate | $1.25 \times 10^{-1}$ | |
| | | -Volume delivered: burette accuracy (ml) | -Tolerance/√3 | $2.89 \times 10^{-3}$ | |
| $m_{init}$ (g) | Mass of sample analysed | | -Weighing scale calibration certificate | | $\mathbf{1.70 \times 10^{-3}}$ |
| $E$ (V) | Potential measured by the glass electrode | -Tolerance of the electrode | -Tolerance/√3 | $1.15 \times 10^{-1}$ | $\mathbf{1.16 \times 10^{-1}}$ |
| | | -Repeatability | -standard deviation from experimental data | $1.11 \times 10^{-2}$ | |
| $T$ (°C) | Temperature of the sample during the titration | -Resolution | -Probe specification | $1.00 \times 10^{-1}$ | $\mathbf{1.00 \times 10^{-1}}$ |
| | | -Repeatability | - standard deviation from experimental data | $2.95 \times 10^{-3}$ | |
| | | -Trueness | -Calibration with a certified temperature probe | $5.66 \times 10^{-3}$ | |
| $R$ (J mol$^{-1}$ K$^{-1}$) | Universal gas constant | | (Pratt, 2014) | | $\mathbf{1.50 \times 10^{-5}}$ |
| $F$ (C mol$^{-1)}$ | Faraday constant | | (Pratt, 2014) | | $\mathbf{8.30 \times 10^{-3}}$ |
| $v_{HCl}$ (mol kg$^{-1}$) | Amount content of the acid titrant | -Stock solution (HCl 1 mol kg$^{-1}$) amount content | -SMU certificate | $6.00 \times 10^{-5}$ | $\mathbf{1.08 \times 10^{-5}}$ |
| | | -Gravimetric dilution | -Calibration certificate | $5.09 \times 10^{-3}$ | |
| $a$ and $b$ | Gran's regression coefficients | -$m_{HCl}$ | -As for $m_{HCl}$ above | | $\mathbf{2.89 \times 10^{-3}}$ |
| | | -F1 | -Law of uncertainty propagation in Eq. 3 | | $\mathbf{= 0.0055 * F1 - 76.39097}$ |
| | | -Gran's regression method | -LNE-RegPoly | a | $\mathbf{5.33 \times 10^{3}}$ |
| | | | | b | $\mathbf{1.59 \times 10^{4}}$ |
| | | | | corr(a,b) | **-0.99889** |
| **Total uncertainty budget of total alkalinity (µmol kg$^{-1}$)** | **u ($k=1$)** | | | | **2.63** |
| | **U ($k=2$)** | | | | **5.26** |

The two uncertainty quantification approaches, i.e. top-down and bottom-up approaches, gave the same level of uncertainty for the total alkalinity measurement results obtained with the standardized measurement method and the Gran's data treatment, being an expanded uncertainty of 5 µmol kg$^{-1}$ ($k$=2, confidence level of approximately 95%).

### 5.3 Discussion

The bottom-up and top-down approaches applied for uncertainty quantification of total alkalinity measurement results obtained from the standardized method and Gran's data treatment yielded really close results, with standard uncertainties of 2.63 and 2.45 µmol kg$^{-1}$, respectively. This level of uncertainty is coherent regarding the precision of the method reported in the literature, typically ranging between 2 and 4 µmol kg$^{-1}$ (Millero et al., 1998; Bockmon and Dickson, 2015). Moreover, it is close to the data quality objective required by the GOA-ON for monitoring ocean acidification, indicating promising prospects for achieving good data quality for TA results.

#### 5.3.1 Top-down approach

The top-down approach appears to provide a realistic evaluation of the uncertainty of TA measurement results. However, conducting an inter-laboratory comparison with more participants could lead to a more robust uncertainty budget. According to the top-down uncertainty budget (Table 5), reducing uncertainty would necessitate mitigating the contributions of inter-laboratory deviation and bias uncertainty. This could be achieved through better harmonization of measurement procedures and reducing uncertainty in the reference value of the material, which requires improving material stability.

#### 5.3.2 Bottom-up approach

The bottom-up approach, which relies on a detailed identification and quantification of sources of uncertainty in the measurement process, helps identify the main contributions to the overall budget.

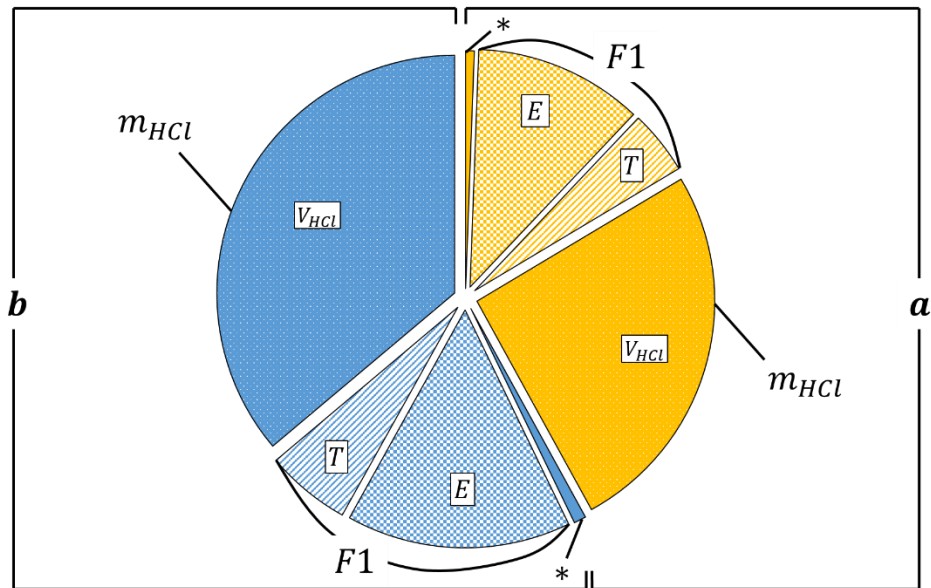

**Figure 5: Main sources of uncertainty contributing to the overall budget of total alkalinity measurement results obtained with the open-cell titration measurement method together with Gran's data treatment. With uncertainty**

**sources coming from $a$ and $b$ represented in yellow and blue, respectively right and left hand sides of the figure. The symbol \* corresponds to the residuals of Gran's regression.**

In Table 6, the main sources identified are the coefficients $a$ and $b$ of Gran's regression, contributing 42% and 58%, respectively, to the uncertainty budget. These components are computed from F1 (Eq. 3) and from the weight of HCl added during titration ( $m_{HCl}$ ). The main sources of uncertainty influencing F1 are the measured potential ($\approx 72.5\%$) and temperature ($\approx 27.5\%$). The main source of uncertainty influencing $m_{HCl}$ is the volume of acid delivered by the burette (nearly 100%). The importance of this parameter on the measurement result was well illustrated by the issue encountered by laboratory 1 during the ILC (Fig. 4, Sect. 4.2). The weight of all of these sources in the overall budget are presented in Fig. 5, remaining sources are highly negligible, and do thus not appear on the chart. Reducing uncertainties in these three components can help diminish the overall budget. However, this is heavily reliant on device resolution and tolerance, and thus depends on the choice of the device and manufacturer. The similarity of the sources of uncertainty influencing $a$ and $b$ in Fig. 5 well illustrates the high correlation between both (corr(a,b) = -0.999).

### 5.3.3 Other considerations

The uncertainty budget detailed in this paper does not include uncertainty arising from the nonlinear least-squares (NLLS) regression typically applied to natural seawater samples. This remains to be quantified and may slightly increase the uncertainty budget. Quantifying this uncertainty will entail considering several aspects and input variables from the NLLS equation, including:

(1) The uncertainty of practical salinity measurements

(2) Possible discrepancies between total fluoride and total sulphate amount contents computed from salinity and the actual composition of natural seawaters worldwide

(3) The uncertainty of dissociation constants of fluoride and sulphate ions

The Monte Carlo approach, as described in GUM Supplement 1 (JCGM 101:2008, 2008), might be pertinent for computing the uncertainty of the NLLS regression as it enables uncertainty computation from the distribution of the regression.

### 6 Metrological traceability

Metrological traceability is defined as the "property of a measurement whereby the result can be related to a reference through a documented unbroken chain of calibrations, each contributing to the measurement uncertainty" (JCGM 200:2012, 2012). The absence of an uncertainty budget associated to the measurement results, and to the TA value of the reference materials currently distributed by the Scripps Institution of Oceanography, prevents proper traceability of the measurement results. Other traceability issues coming from the measurement process should also be carefully investigated. By developing an artificial reference material with a reference value accompanied by a complete uncertainty budget, as well as by providing an initial estimation of the uncertainty in TA measurement results, this study advances the establishment of traceability. To fully establish traceability, it will be necessary to quantify the background alkalinity in the artificial reference material in a more robust manner, and to evaluate the uncertainty associated with the NLLS regression. An enhanced traceability route, based on the two reference materials developed, is presented in Capitaine et al. (in preparation).

## 7 Conclusion

This study explores the application of various metrological tools to measurements of total alkalinity (TA) of seawater using the standardized multi-step potentiometric titration method.

Two batches of an artificial certified reference material with reference TA values of, respectively, $2503.6 \pm 2.3$ µmol kg$^{-1}$ and $2503.8 \pm 2.8$ µmol kg$^{-1}$ ($k$=2) for a shelf-life of three months have been produced, alongside a second reference material comprising stabilized natural seawater. These materials underwent homogeneity and stability studies to comply with ISO standard 17034 "General requirements for the competence of reference material producers" (2016). While the homogeneity study requires more precise measurements to obtain significant information, stability was evaluated to be unsatisfactory due to an increase in TA over time. This might be partly attributed to the release of silicates from the glass container but needs further investigation. Using a different type of bottling is suggested to enhance stability.

An inter-laboratory comparison involving five laboratories indicated that both reference materials could be suitable for quality control of the standardized total alkalinity measurement method.

The uncertainty of the multi-step potentiometric measurement method with Gran's data treatment was quantified through both bottom-up and top-down approaches, yielding expanded uncertainties of 5.26 and 4.90 µmol kg$^{-1}$, respectively ($k$=2, confidence level of approximately 95%). Although slightly higher than required by the GOA-ON ($< 4$ µmol kg$^{-1}$) for monitoring ocean acidification, these results are very encouraging for achieving good data quality of TA measurement results. The bottom-up approach helps identify key sources of uncertainty to prioritize for improvement. Quantification of the nonlinear least-squares regression will be necessary to establish the overall uncertainty budget of seawater TA measurement results.

The results presented in this paper represent progress towards ensuring compatibility and accuracy of seawater total alkalinity measurement results.

**Appendix A**

 **Standard procedure as well as data treatment method for the open-cell multi-step titration for the determination of seawater total alkalinity.**

A known amount of sample, measured by gravimetry, is placed in an open-cell thermostated at 25°C. The sample is titrated by an HCl solution of known amount content and density. A glass electrode allows the monitoring of the potential during the titration. The glass electrode is first calibrated in total pH scale (noted $pH_T$) using a TRIS buffer.

The titration is then carried out in two stages. The first stage consists in adding enough HCl to reach a $pH_T$ situated just beyond the endpoint (between $pH_T$ 4 and 3.5). At this $pH_T$, the predominant weak bases, $HCO_3^-$ and $CO_3^{2-}$, are converted to $CO_2$. This $CO_2$ is removed by agitation and by bubbling of air through the solution for around 6 minutes. A further addition of HCl, in a series of small increments, allows reaching a $pH_T$ of about 3. At this $pH_T$, all proton acceptors are consumed. The data given by the titration (i.e. measured potential, temperature and volume

of HCl added) during the second stage is used to compute the total alkalinity. Data are taken only for this range of $pH_T$ as it is low enough to neglect residual bicarbonate ions and high enough so that the Nernst equation still holds true (Dickson et al., 2007).

An initial estimate of the total alkalinity is obtained from the titration curve using Gran's method (Gran, 1952). This is a highly effective method for the determination of the equivalence point in potentiometric titrations.

At each point of the titration, the amount content of hydrogen ions $v_H$ (mol kg$^{-1}$) can be described by Eq. A1.

$$v_H = \frac{m_{HCl} \, v_{HCl} - m_{init} \, TA}{m_{HCl} + m_{init}} \tag{A1}$$

where $m_{HCl}$ is the mass of acid added (g), $v_{HCl}$ the acid amount content (mol kg$^{-1}$), and $m_{init}$ the mass of sample analysed (g).

In the range of $pH_T$ corresponding to the second stage of the titration, the following equation is valid (Eq. A2):

$$\frac{m_{HCl} \, v_{HCl} - m_{init} \, TA}{m_{HCl} + m_{init}} = [H^+] + [HSO_4^-] + [HF] \approx [H^+]_T \tag{A2}$$

The glass electrode has been calibrated in total pH:

$$[H^+]_T = \exp\left(\frac{E - E^\circ}{\frac{RT}{F}}\right) = cst \, \exp\left(\frac{E}{\frac{RT}{F}}\right) \tag{A3}$$

where $E$ is the potential measured by the glass electrode (V), $E^\circ$ its reference potential (V), $R$ the universal gas constant (J mol$^{-1}$ K$^{-1}$), T the temperature of the sample (K), F the Faraday constant (C mol$^{-1}$) and *cst* and undefined

constant.

From Eq. A2 and A3, the Eq. A4 is computed.

$$(m_{init} + m_{HCl}) \times \exp\left(\frac{E}{\frac{RT}{F}}\right) = \frac{- TA \, m_{init} + m_{HCl} \, v_{HCl}}{cst} \tag{A4}$$

The left-hand side of this equation defines the Gran function F1 (Eq. A5).

$$F1 = (m_{init} + m_{HCl}) \times \exp\left(\frac{E}{\frac{RT}{F}}\right) \tag{A5}$$

F1 is plotted as a function of the amount of HCl added for each point of the second stage of the titration, and TA

is thus obtained with Eq. A6.

$$TA = \frac{-b}{a} \frac{v_{HCl}}{m_{init}} \tag{A6}$$

where coefficients $a$ and $b$ represent, respectively, the slope and the intercept of the linear regression $F1 = a * m_{HCl} + b$.

This method gives a first estimation of the total alkalinity. However, errors are introduced when using the Gran's method for seawater analysis due to competing acid-base equilibria in seawater. A method allowing to solve the equivalence point by curve fitting has thus been developed (Dickson, 1981; Martz, 2005). This method consists in an iterative process where the standard potential of the glass electrode (E°) is calculated from the estimation of TA obtained by the Gran's method. A nonlinear least-squares (NLLS) regression is then used to refine the values of E° and TA. The refinement in E° first allows the calculation of the factor $f = [H^+]_T/[H^{+'}]_T$, where $[H^{+'}]_T$ is obtained from the refinement in E° and Eq. A3. $f$ is then itself used to determine a new value of TA using Eq. A7.

$$TA + \frac{S_T}{1 + \frac{K_S Z}{f [H^{+'}]_T}} + \frac{F_T}{1 + \frac{K_F}{f [H^{+'}]_T}} + \frac{m_{init} + m_{HCl}}{m_{init}} \frac{f [H^{+'}]_T}{Z} - \frac{m_{HCl} v_{HCl}}{m_{init}} = 0 \qquad (A7)$$

where $S_T$ is the total sulphate amount content (mol kg$^{-1}$), $F_T$ the total fluoride ion amount content (mol kg$^{-1}$), $K_S$ the dissociation constant of $[HSO_4^-]$, $K_F$ the dissociation constant of hydrogen fluoride, and $Z = 1 + \frac{S_T}{K_S}$.

The nonlinear least-squares regression consists in computing how much the left-hand side of Eq. A7 differs from zero. The residuals are squared and the sum of squares is minimized by adjusting $f$ and TA using an algorithm. By applying this method, the errors in $K$s are negligible (Dickson et al., 2007).

**Instrumentation for total alkalinity measurements performed at LNE**

The measurement of total alkalinity at LNE was made using the titration system and the 888 Titrando electroburette from *Metrohm*, associated with the 801 Stirrer agitation system. A thermostated glass cell with a capacity of 50-150 ml was used. The volume of the samples analysed was of 100 ml. This cell was connected to a *LAUDA* Eco Gold bath to control the temperature of the cell. The setpoint was fixed in order to obtain a temperature of 25 ± 0.2°C in the cell. The temperature was maintained stable, i.e. the temperature acquired during the whole titration

had a standard deviation within 0.05°C. The potential measurement was carried out with a *Metrohm Ecotrode Plus* glass electrode (ref: 6.0262.100) and the temperature with the *Metrohm* temperature probe (ref: 6.1110.100). The data acquisition software used was Tiamo 2.4.

The airflow for $CO_2$ degassing was obtained from a compressed air tank connected to an inlet system in the cell. Fig. B1 illustrates the description of the setup.

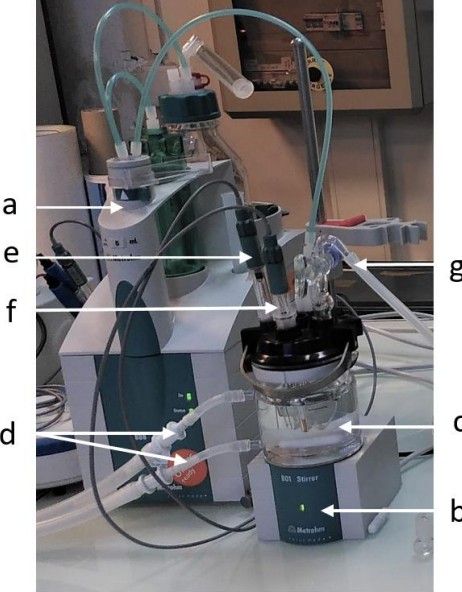

**Figure B1: Description of LNE's total alkalinity measurement setup. With a) Burette and titration system, b) Magnetic stirrer, c) Thermostated cell, d) Tubing connected to the water bath, e) Glass electrode, f) Temperature probe, and g) Air inlet for degassing.**

The hydrochloric acid used was a Standard Reference Material (SRM) at 1 mol kg$^{-1}$ prepared and characterized by

coulometry at the Slovenský Metrologický Ústav (Slovak Institute of Metrology, SMU). It was diluted by mass to 0.1 mol kg$^{-1}$ before its use as acid titrant; it wasn't prepared in an NaCl matrix. It is often recommended that the HCl solution is prepared in an NaCl matrix in order to keep the ionic strength, and thus the activity coefficients, constant during the titration. However, Okamura et al. (2014) has indicated that using HCl solution containing no NaCl has a negligible effect on the TA results (about -0.2 µmol kg$^{-1}$). The density of the titrant solution was

measured with a DMA 4500M *Anton Paar* densimeter.

The mass of the analyzed sample was weighed on a calibrated 2 kg balance with a resolution of 0.1 mg.

The data treatment was performed as described in Appendix A, with an R routine written based on the function "alkalinity" of the package seacarb (Gattuso et al., 2021).

The glass electrode was calibrated with a TRIS buffer prepared and characterized with the Harned cell

measurement method at LNE. Its linearity over a range of pH was checked with NBS buffers of pH 4, 7 and 12.

The electroburette and the temperature probe are calibrated once a year.

The accuracy of the method was controlled either with a reference material purchased from Scripps or with a synthetic solution prepared gravimetrically at LNE.

**Appendix C**

**Table C1: Total Alkalinity measurement results of all participants to the inter-laboratory comparison (µmol kg$^{-1}$), including the stability monitoring over time. The "Mean" corresponds to the mean value of replicates, with standard deviation reported as "SD".**

| Laboratory | 1 | | 2 | | 3 | | 4 | | 5 | |
|---|---|---|---|---|---|---|---|---|---|---|
| Time after solution preparation (months) | Mean | SD | Mean | SD | Mean | SD | Mean | SD | Mean | SD |
| Artificial reference material - Batch 1   2 | 2514.6 | 4.9 | 2503.1 | 1.1 | 2501.6 | 2.4 | 2500.4 | 0.3 | 2499.2 | 4.6 |
| 5 | 2516.8 | 0.7 | | | 2502.1 | 1.6 | 2499.2 | 1.8 | 2487.5 | 0.5 |
| 8 | 2506.5 | 0.7 | | | 2506.5 | 0.6 | 2510.5 | 1.2 | 2522.5 | 3.4 |
| 11 | 2507.1 | 1.9 | | | 2510.2 | 1.9 | 2505.4 | 1.3 | 2502.8 | 5.8 |

| Laboratory | LNE | |
|---|---|---|
| Time after solution preparation (months) | Mean | SD |
| Artificial reference material - Batch 2   2 | 2503.7 | 0.6 |
| 5 | 2504.6 | 1.5 |
| 8 | 2509.3 | 0.6 |
| 11 | 2512.9 | 0.7 |
| 14 | 2513.5 | 2.3 |

| Laboratory | 1 | | 2 | | 3 | | 4 | | 5 | |
|---|---|---|---|---|---|---|---|---|---|---|
| Time after solution bottling (months) | Mean | SD | Mean | SD | Mean | SD | Mean | SD | Mean | SD |
| Natural Reference Material   2 | 2595.3 | 3.2 | 2580.3 | 1.6 | 2581.4 | 0.9 | 2580.2 | 2.5 | 2592.3 | 2.3 |
| 5 | 2593.3 | 0.4 | | | 2583.5 | 0.9 | 2582.0 | 1.4 | 2573.5 | 1.4 |
| 8 | 2592.5 | 0.6 | 2583.7 | 1.6 | 2581.2 | 0.4 | 2588.1 | 0.6 | 2603.4 | 5.9 |
| 11 | 2590.5 | 0.6 | 2590.3 | 1.0 | 2589.7 | 1.4 | 2587.0 | 0.7 | 2573.5 | 0.3 |
| 14 | 2580.3 | 1.8 | | | 2582.0 | 1.8 | 2585.1 | 2.2 | 2577.4 | 0.8 |

**Table C2: Dissolved Inorganic Carbon measurement results performed at the SNAPO-CO$_2$ (µmol kg$^{-1}$), including the stability monitoring over time. The "Mean" corresponds to the mean value of replicates, with standard deviation reported as "SD".**

| Laboratory | SNAPO-CO2 | |
|---|---|---|
| Time after solution bottling (months) | Mean | SD |
| Natural Reference Material   2 | 2384.9 | 1.2 |
| 8 | 2384.4 | 2.2 |
| 11 | 2385.5 | 1.1 |

**Table C3: Nutrients measurement results performed at the MIO (µmol l$^{-1}$), including the stability monitoring over time. The "Mean" corresponds to the mean value of replicates, with standard deviation reported as "SD".**

| Laboratory | MIO | | | | | | | |
|---|---|---|---|---|---|---|---|---|
| | silicates | | nitrites | | phosphates | | nitrates+nitrites | |
| Time after solution bottling (months) | Mean | SD | Mean | SD | Mean | SD | Mean | SD |
| 2 | 12.4 | 0.1 | 0.02 | 0 | 0.40 | 0.01 | 9.08 | 0.03 |
| 11 | 21.9 | - | 0.00 | - | 0.75 | - | 9.32 | - |
| 14 | 23.8 | 0.0 | 0.01 | 0 | 0.45 | 0.01 | 8.79 | 0.09 |

(Natural Reference Material)

| Laboratory | MIO | | | |
|---|---|---|---|---|
| | silicates | nitrites | phosphates | nitrates + nitrites |
| Time after solution preparation (months) | Mean | Mean | Mean | Mean |
| 14 | 27.1 | 0.03 | 0.03 | 0.4 |

(Artificial Reference Material - Batch 2)

**Appendix D**

**Details on the uncertainty propagation process for the quantification of the uncertainty sources involved in the Total Alkalinity measurement method and Gran's data treatment following the bottom-up approach**

**Table D1: F1 uncertainty**

<table>
<tr><td colspan="5" align="center">F1</td></tr>
</table>

**Fichier :**

**Titre de l'étude :**  F1

**Fichier Modèle :**

**Commentaires :**

**Définition des variables :**

| Code | Description | Valeur | Méthode | Incertitude absolue |
|---|---|---|---|---|
| m_HCl | m_HCl | 3.078869534E+000 g | Personnalisée | 2.890000E-003 g |
| m_init | m_init | 1.0048000E+002 g | Personnalisée | 1.70E-003 g |
| Potentiel | E | 2.199620972E+002 mV | Personnalisée | 1.160000E-001 mV |
| R | R | 8.31447215E+000 J/mol.K | Personnalisée | 1.500E-005 J/mol.K |
| T | T | 2.982023838E+002 K | Personnalisée | 1.000000E-001 K |
| F | F | 9.648533992E+004 C/mol | Personnalisée | 8.30E-003 C/mol |

**Incertitudes sur les fonctions :**   Méthode Numérique
Incertitudes calculées sans les corrélations des variables

| Code | Nom | Expression | Valeur | Incertitude élargie U |
|---|---|---|---|---|
| F1 | F1 | (m_HCl+m_init)*EXP(((Potentiel/1000)*F)/(T*R)) | 5.40330214E+005 | 5.78111471E+003  (k = 2) |

**Bilan par composantes :**
F1

| Variable | Sensibilité C(Xi) | C(Xi).u(Xi) | Poids |
|---|---|---|---|
| m_HCl | 5.21761E+003 | 1.50789E+001 | 0.00% |
| m_init | 5.21761E+003 | 8.86994E+000 | 0.00% |
| Potentiel | 2.10269E+004 | 2.43913E+003 | 71.20% |
| F | 4.79359E+001 | 3.97868E-001 | 0.00% |
| T | -1.55100E+004 | -1.55100E+003 | 28.79% |
| R | -5.56273E+005 | -8.34409E+000 | 0.00% |

**Table D2:** $v_{HCl}$ **uncertainty**

<table>
<tr><td colspan="5" align="center">**Solution mère HCl**</td></tr>
</table>

**Fichier :**

**Titre de l'étude :**         Solution mère HCl

**Fichier Modèle :**

**Commentaires :**

**Définition des variables :**

| Code | Description | Valeur | Méthode | Incertitude absolue |
|------|-------------|--------|---------|---------------------|
| Stock_HCl | m_stock | 1.00103E+000 mol/kg | Personnalisée | 6E-005 mol/kg |
| m2 | mHCl | 4.9511100000E+001 g | Personnalisée | 5.091169E-003 g |
| m3 | mH2O | 4.61393900000E+002 g | Personnalisée | 5.091169E-003 g |
| M_H | masse molaire H | 1.007940E+000 g/mol | Personnalisée | 3E-006 g/mol |
| M_Cl | masse molaire Cl | 3.5453E+001 g/mol | Personnalisée | 1E-003 g/mol |

**Incertitudes sur les fonctions :**         Méthode Numérique
Incertitudes calculées sans les corrélations des variables

| Code | Nom | Expression | Valeur | Incertitude élargie U |
|------|-----|------------|--------|------------------------|
| v_HCl | | m2*Stock_HCl/(m2+m3) | 9.70084388E-002 | 2.15311E-005  (k = 2) |
| b_HCl | | v_HCl/((1000-(M_H+M_Cl) *v_HCl)/1000) | 9.73527774E-002 | 2.16843E-005  (k = 2) |

**Bilan par composantes :**
**v_HCl**

| Variable | Sensibilité C(Xi) | C(Xi).u(Xi) | Poids |
|----------|-------------------|-------------|-------|
| m2 | 1.769E-003 | 9E-006 | 70.02% |
| Stock_HCl | 9.6909E-002 | 6E-006 | 29.17% |
| m3 | -1.90E-004 | -1E-006 | 0.81% |

**b_HCl**

| Variable | Sensibilité C(Xi) | C(Xi).u(Xi) | Poids |
|----------|-------------------|-------------|-------|
| m2 | 1.782E-003 | 9E-006 | 70.02% |
| Stock_HCl | 9.7598E-002 | 6E-006 | 29.17% |
| m3 | -1.91E-004 | -1E-006 | 0.81% |
| M_H | 9E-006 | 3E-011 | 0.00% |
| M_Cl | 9E-006 | 9E-009 | 0.00% |

**Corrélation des fonctions :**

| F1 | F2 | r(F1,F2) |
|----|----|----------|
| v_HCl | b_HCl | 1 |

**Table D3: TA uncertainty**

---

**AT GRAN**

---

**Fichier :**

**Titre de l'étude :**   AT GRAN

**Fichier Modèle :**

**Commentaires :**

**Définition des variables :**

| Code | Description | Valeur | Méthode | Incertitude absolue |
|------|-------------|--------|---------|---------------------|
| C | C_HCl | 1.00056000E-001 mol/kg | Personnalisée | 1.080E-005 mol/kg |
| a | a | 9.72147379000E+005 | Personnalisée | 5.331889406E+003 |
| b | b | -2.45283952100E+006 | Personnalisée | 1.591406073E+004 |
| m1 | m_init | 1.0048000E+002 g | Personnalisée | 1.70E-003 g |

**Incertitudes sur les fonctions :**   Méthode Numérique
Incertitudes calculées avec les corrélations des variables

| Code | Nom | Expression | Valeur | Incertitude élargie U |
|------|-----|------------|--------|-----------------------|
| m_HCl | m_HCl_GRAN | (-b)/a | 2.523114884E+000 | 5.2573105E-003 (k = 2) |
| AT | AT | (m_HCl*C/m1)*1000000 | 2.512467982E+003 | 5.263834951E+000 (k = 2) |

**Bilan par composantes :**
m_HCl

| Variable | Sensibilité C(Xi) | C(Xi).u(Xi) | Poids |
|----------|-------------------|-------------|-------|
| b | -1E-006 | -1.6370E-002 | 58.32% |
| a | -3E-006 | -1.3839E-002 | 41.68% |

AT

| Variable | Sensibilité C(Xi) | C(Xi).u(Xi) | Poids |
|----------|-------------------|-------------|-------|
| b | -1.024E-003 | -1.63009E+001 | 58.31% |
| a | -2.585E-003 | -1.37804E+001 | 41.67% |
| C | 2.51106E+004 | 2.71195E-001 | 0.02% |
| m1 | -2.50047E+001 | -4.2508E-002 | 0.00% |

**Corrélations des variables :**

| X1 | X2 | r(X1,X2) |
|----|----|----------|
| C | a | 0.0 |
| C | b | 0.0 |
| C | m1 | 0.0 |
| a | b | -0.99889 |
| a | m1 | 0.0 |
| b | m1 | 0.0 |

**Corrélation des fonctions :**

| F1 | F2 | r(F1,F2) |
|----|----|----------|
| m_HCl | AT | 0.995 |

**Data availability**

The data are presented in the main manuscript or in Appendix.

**Authors' contribution**

G.C.: Conceptualization, Formal analysis, Investigation, Methodology, Validation, Visualization, Writing – original draft preparation; S.A., T.C. & J.F.: Investigation, Methodology, Writing – review and editing; P.F.: Funding acquisition, Supervision, Writing – review and editing; T.W.: Conceptualization, Investigation, Methodology, Supervision, Writing – review and editing

**Competing interests**

The authors declare that they have no conflict of interest.

**Acknowledgments**

The authors acknowledge Steeve Comeau and Frederic Gazeau for the constructive discussions on the methodology of the inter-laboratory comparison; LNE's statisticians from the Data Science and Uncertainty Department for their support on computing uncertainty budgets; MIO's scientists from the platform of Analysis of Basic Parameters for the dissolved nutrients analysis; the MOOSE network for the collection of deep natural seawaters, as well as the Scientific Consortium of Expertise for marine pH/CO2 of ODATIS. We also acknowledge the editors and reviewers for their valuable inputs to the paper.

**Financial support**

Gaëlle Capitaine was supported by a CIFRE scholarship provided by ANRT (Association Nationale de la Recherche et de la Technologie).

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
