# Peer review of "Metrological concepts applied to Total Alkalinity measurements in seawater: reference materials, inter-laboratory comparison and uncertainty budget"

_EGUsphere, 2025_

## Referee Comment (RC1)

Review of OS manuscript egusphere-2025-3588

Title: Metrological concepts applied to Total Alkalinity measurements in seawater: reference materials, inter-laboratory comparison and uncertainty budget

General comments

The authors have characterized a reference material (RM) prepared from artificial seawater by means of gravimetric measurements and quantified its measurement uncertainty using a bottom-up approach in accordance with the GUM, taking into account the homogeneity and stability of the RM. To validate the results, a comparison measurement was carried out. In addition, another RM based on natural seawater was investigated. In this case, a top-down approach was applied to quantify the measurement uncertainty based on the results of a second comparison. Here as well, homogeneity and stability were assessed. This RM has been characterization using a common TA measurement procedure. The RMs and the corresponding measurement results and their interpretation provide a first metrological evaluation of the measurement uncertainty of TA measurements, which is currently lacking. The manuscript is therefore addressing a question of scientific interest and is relevant for reliable measurements of seawater TA. Therefore, publication of the paper is recommended/accepted. Nevertheless, a revision is necessary.

Before publication, the paper must address the following issues:

1. The traceability of the TA values and their uncertainties - which must be the core element of any RM, particularly if the authors, as they claim, worked in accordance with ISO 17034 – are inconsistent to some extent and are not sufficiently discussed. As the authors have noted themselves, TA measurements using the conventional method according to Dickson's Guide have no proper traceability. A detailed analysis of its traceability is currently lacking. Nevertheless, the authors use this method to characterize TA of the natural seawater TA. Moreover, even though the gravimetric approach to characterize the artificial RM is traceable to the SI, the quantification of impurities in NaCl is conducted with the common method, which introduces an inconsistency into the characterization. Additionally - as the authors themselves note - the artificial seawater differs chemically from natural seawater. This also affects the regression method used to determine TA and thus its associated measurement uncertainty. Consequently, measurements of seawater TA that are referred to the artificial RM may still contain significant biases, even though the comparison measurements show reasonably good agreement.
Beyond these more fundamental issues, the overall concept of traceability that the authors seem to have in mind is not presented clearly. Traceability can fundamentally be established only to one metrological reference—either to the artificial or to the natural seawater RM. However, two very different RMs are introduced: the artificial RM, characterized gravimetrically, and the natural seawater RM, measured using the standard method according to Dickson's SOP 3a/b. The first is proposed to establish SI

traceability while the latter is proposed to serve as an additional reference for quality control. However, what if this kind of quality control provides a deviating result, which is the RM to believe? In turn, if both RM provide compatible results, why is there a need for second kind of RM?

The manuscript is somewhat vague with respect to those traceability issues. The authors are not expected to solve these difficult, general problems of traceability in this paper, which would likely be beyond the scope of the study that mainly aims to evaluate the RMs, which has sufficient value in its own. Nevertheless, they must discuss and contextualize both RMs in light of these traceability challenges and clearly define their respective limitations.

2. The structure of the paper should be reconsidered. The typical format—theory, results, discussion—makes this paper rather confusing, since each of the three sections successively addresses several different topics. There are two RMs, two characterizations, homogeneity and stability studies, and two comparison measurements. For each, the theoretical background is first explained, then all results are presented together, and finally everything is revisited for discussion. As a result, the reader easily loses track of which parts belong together and is constantly forced to flip back and forth between sections.

This is not a mandatory requirement to be addressed for publication. However, to the reviewer's opinion readability would be improved if the authors restructure the paper by addressing the artificial seawater RM first—covering its preparation, characterization, stability, and homogeneity, including the relevant calculation principles and results, and concluding with the discussion. The same should then be done for the natural seawater RM. Finally, the comparison measurements should be presented, allowing both RMs to be contrasted, and a proposal for traceability should be discussed in more detail, also from a practical perspective.

Additionally, the paper is rather long, considering that it essentially evaluates RMs using well-established procedures. The authors should consider shortening it to some extent.

3. The reviewer recommends using an LLM-based AI to improve the language. In parts, the paper is difficult to read due to linguistic weaknesses, which the reviewer did not further correct.

Specific technical comments

| line | comment |
|------|---------|
| 45 | How do uncertainty limits illustrate climatic variations in TA? Please rephrase for more clarity. |
| 54 | "Not fully traceable" is a strong statement that requires discussion. If characterized HCl was used for the titration, why is the measurement not traceable to the SI? This should be discussed — if not in the introduction, then elsewhere in the manuscript. |

| | |
|---|---|
| 56,57 | The metrological terminology is somewhat imprecise. Comparability of results is achieved through traceability to the same metrological reference, not through uncertainty. Measurement uncertainty defines the limits within which differences between measurement results — or their equivalence — become meaningful (see also VIM: compatibility). Only deviations exceeding the measurement uncertainty can be regarded as significant.
Similarly, the term "uncertainty of a measurement method" is incorrect — a method itself has no uncertainty; only a measured value has one.
The authors should also verify whether the word "trueness" in line 56 expresses what they intend to say. According to the VIM, *trueness* refers to "the closeness of agreement between the average of an infinite number of replicate measured quantity values and a reference quantity value." I am not sure, if this meant. |
| 59 | ISO Guide 35 has been replaced by ISO 33405. A paper related to metrological science should not refer to outdated standards. |
| 76 | Section 2.1: The purpose of this brief summary of Dickson's SOP 3b is not clear. Usually, reference to Dickson's Guide would be suffice, all the more, the paper is already quite long. If there is a reason for the repetition, it should mentioned. I assume the formulas are mentioned because it is relevant for the uncertainty calculations in subsequent sections? |
| 164 | The measurement result at zero NaCl mol/kg sol is shown … |
| 164 | Figure 2 is mentioned in the main text before Figure 1. Figures should be cited in the order of their appearance. |
| 165 | Replace "theory" with "reasonable assumption." |
| 164 | Even with goodwill, Fig. 2 does not support the assumption of a linear relationship passing through the origin. It rather shows a square root like behavior, which is difficult to explain. Alternatively, the $\Delta$TA values at 1, 2 and 3 mol/kg NaCl solution content indeed suggest that there is a linear relationship - which one can expect in dependence of NaCl content – but with an offset at zero NaCl content. Which raises the question, why the measured $\Delta$TA value is zero at zero NaCl content? I suspect, that the reason for this discrepancy can be found in the different metrological references involved in the gravimetric and measured TA values. See also comments related to lines 236 and 522.

Anyhow, the linear extrapolation might be used as a rough estimate for the background alkalinity. However, the authors must comment on the difficulties I have mentioned. |
| 168, 170 | The purpose of measuring practical salinity and dissolved nutrients should be stated. |
| 205 | A reference to ISO 33405 would be more appropriate here. |

| 206 | The authors claim to evaluate the proposed RMs in accordance with ISO 17034. If so, they must fulfill the experimental requirements for short-term stability testing. Using a single, undefined transport of the RM does not meet these requirements. Since this uncertainty contribution can presumably not be readily quantified, I recommend refraining from claiming that this value has been determined. Instead, it should be stated that the value represents a first estimate, while a proper evaluation according to ISO 17034 is still pending. |
|---|---|
| 214 | "ISO Guide 35" — see comment on line 59. |
| 236 et seq | It is unclear whether the calculation of the bias introduced by NaCl impurities is used solely to correct the reference value or to quantify its contribution to the uncertainty. This must be explicitly stated to avoid confusion. In any case, the approach appears to lead to a circular argument regarding traceability. The authors aim to correct the bias and/or assign a corresponding uncertainty to the RM. To do so, they measure TA and subtract this value from the one obtained via gravimetric measurements. However, in order to measure the TA with proper traceability, they would need a characterized RM traceable to the same metrological reference as the artificial RM — which is not available except for the proposed one. If, as assumed, the authors used Dickson's SOP to measure TA, then the traceability of the bias/uncertainty is subject to the same limitations inherent to that SOP (as mentioned in the introduction). Thus, traceability of the assigned TA value of the artificial RM, or its uncertainty, respectively, is questionable. Fundamentally, the bias/uncertainty must be quantified independently of the RM it is intended to characterize and with respect to the same metrological reference. |
| 245 | "Systematic uncertainty sources, such as those arising from the device, the operator, or the procedure, are cancelled here." This statement is not self-evident. Which uncertainties cancel out, and how are they correlated? The authors should explain this important aspect in more detail. |
| 271 | "It was chosen to neglect the within-bottle homogeneity." Again, the authors claim compliance with ISO Guide 35 (ISO 33405, respectively); however, their homogeneity analysis appears superficial to some extent. A one-way ANOVA must be applied to account for both within-unit and between-unit homogeneity. One might decide to disregard within-unit uncertainty for the reasons mentioned by the authors. In that case, only between-bottle homogeneity should be calculated according to the (corrected) Eq. 11. Otherwise, a proper one-way ANOVA analysis is expected for the homogeneity values given in Table 4. The authors must also evaluate the repeatability standard deviation of the homogeneity with respect to the target uncertainty (see Section 7.5.1 of ISO 33405). |
| 274 | Equation 11 is incorrect. In the simplest approach, assuming within-unit homogeneity, |

| | |
|---|---|
| | $u_{\text{hom}} = M_{\text{between}}/n_0$,
where $M_{\text{between}}$ is the mean square of the TA results of the units and $n_0$ is the number of measurements per unit (assuming they are equal for each unit). See, for example, Section 7.7.3 and Annex B1 of ISO 33405. |
| 281, 282 | The equations are mutually inconsistent. $u_{\text{stab}}$ cannot comply with both Eq. 12 and Eq. 13. I would recommend referring to Eq. 11 in Section 8.7.3 and Annex B3 of ISO 33405 instead. |
| 296 | Improve clarity: Do the authors mean that the artificial RM and the stabilized natural RM were prepared at approximately the same time, or one after the other? |
| 301 | Replace "calibrated" with "characterized." |
| 390 | "… and is included in the reference values given above." The meaning is unclear. The authors should be more precise: do the TA values in Table 3 really include the TA contribution from NaCl impurities, meaning the bias has not been corrected, or do they mean it has been considered, meaning the values in Table 3 have been corrected for this bias? See also the comment on line 236. |
| 404 | Table 4: The authors should add the units more precisely, as not all quantities are given in μmol/kg. |
| 404/406 | Table 4: It was mentioned that within-unit homogeneity was neglected; nevertheless, corresponding values are shown. Moreover, the results suggest that within-unit variability is even larger than between-bottle homogeneity, which seems unlikely. This supports the recommendation that measurement repeatability should be assessed in relation to the evaluation of homogeneity, stability and target uncertainty (also see comment on line 271). |
| 409 | "Its stability has been studied …" I recommend adding a figure to illustrate the stability results. |
| 458 | Results should not be excluded solely for statistical reasons, especially when only a small number of participants is involved. Have potential causes of deviation related to the measurement itself been investigated? |
| 475 | Table 6: Using Eq. 16, the values for $s_L$ and $s_r$ given in Table 6, $n = 3$, $p = 4$, and $u(\mu) = 1.08$, I calculate $u(\Delta) = 1.40$ μmol/kg. The authors should verify the values, or clarify which numbers were used in their calculation. |
| 483 | Tables 6 and 7 are nearly identical. I recommend combining them. |
| 510 | Since natural seawater consists of around 90 % NaCl, it is unlikely that the difference in composition between natural and artificial seawater could account for a tenfold discrepancy between the expected and observed differences in practical and absolute salinity. In any case, it is unclear why this matters. The authors should clarify the relationship between TA and salinity to make the relevance of this discrepancy for the study apparent. |

| | |
|---|---|
| 522 | I doubt the validity of this method for the reasons already mentioned in the comment on line 236 et seq. It would be more appropriate to quantify the impurities affecting TA using independent measurement methods. One cannot use the same instrument or procedure intended to be calibrated with the RM to determine the bias of that RM — this constitutes circular reasoning. It becomes impossible to distinguish whether the bias originates from the RM itself (e.g., NaCl impurities) or from the instrument or measurement procedure. For instance, if the bias depends on the ionic strength of the RM, demonstrating that the $\Delta$TA at zero NaCl is zero does not resolve the issue. |
| 551 | Why only "linear": There could be other types of dependencies. |
| 553 | "However, it does not allow for the accuracy verification of TA values obtained using the nonlinear least-squares regression method …" This statement is not incorrect, but it is misleading, as it implies that the evaluation method itself is the cause of the problem. In fact, if an RM with an assigned value is available, the evaluation method is not critical — any method-related bias can be compensated by the known value of the RM. The real issue lies in the different chemical composition of natural versus artificial seawater. This difference necessitates using an evaluation method (NLLS) that differs from the one used to assign the value to the artificial seawater RM (Gran's method). |
| 555 | If a natural seawater RM is needed anyway to measure natural seawater, what is the benefit of using the artificial seawater RM? |
| 558 | "Having a natural seawater reference material that is easy to collect during open-ocean oceanographic cruises …" I find it difficult to see how this proposal could be implemented in practice, or what its benefit would be. Which institution would characterize such an *in situ* RM prepared by the operator during a cruise? And if that were feasible, why would the user rely on any other RM? If the operator is capable of characterizing an RM, they could directly apply the same method to measure their samples. |
| 563 | "… method's limited precision": Where has this been discussed? As mentioned above, this evaluation should indeed be addressed (following the guidance in ISO 33405. |
| 575 | The purpose of the DIC measurements is not stated. I assume they were intended to demonstrate that the carbon content did not change over time. Consequently, any instability of the RM must result from sources other than carbon, such as silicates. The authors should not leave it to the reader to infer the reasons for including specific results in the investigation. |
| 586 | "… lack stability": A good observation that appropriately addresses the scope of the paper. |
| 587 | "… indicating potential secondary processes influencing alkalinity." Such as? It is indeed a peculiar finding that the TA results do not reflect the increase in |

| | |
|---|---|
| | silicate. Identifying secondary processes in natural seawater may be difficult because of its complex composition. However, the composition of the artificial seawater is known—except perhaps for the NaCl impurities—so, an evaluation of the discrepancy should at least be feasible for the artificial RM. |
| 599 | The potential failure should also be mentioned in the results section (see comment on line 458). |

---

## Referee Comment (RC2)

**Overall assessment:**

This study reports on the preparation and evaluation of two different types of reference materials for seawater total alkalinity measurements – (1) a novel artificial material in a NaCl medium with a well-characterized composition traceable to the SI and (2) a stabilized natural seawater material. The authors assessed the homogeneity and stability of the materials and quantified the overall uncertainty through a top-down approach from an interlaboratory study with 5 participants and a bottom-up approach. The work addresses the need for a traceable reference material for total alkalinity measurements and provides a detailed uncertainty budget for the most commonly used method for total alkalinity determination in seawater. The results are informative to reference material producers and advance understanding of the quality of seawater total alkalinity measurements. Although this manuscript is worthy of publication, it requires revisions to improve presentation of statistics and other results as well as clarity of language.

**General comments:**

**Uncertainty estimates for the natural seawater reference material**

The authors do not provide uncertainty estimates for the natural seawater reference material which does not have a well-characterized reference value traceable to the SI. Although information about the trueness of the reference value is required to estimate its uncertainty, it seems that this shouldn't prevent the authors from developing a partial uncertainty budget for the open-cell titration method on natural seawater samples (similar to the uncertainty budget they presented in **Table 7**). In section 4.4, the authors discuss additional contributions that would need to be considered in an uncertainty budget for the titration of natural seawater samples. Why was this not done? Presenting an uncertainty budget for natural seawater samples would be informative to understanding the likely overall uncertainty and its most important contributions in measurements of real samples.

**Expanded uncertainty:**

The authors should include information about the degrees of freedom when presenting estimates of expanded uncertainties, so the level of confidence associated with the chosen coverage factor can be determined.

**Tables:**

The authors should carefully reconsider the organization of their tables. Some tables are cluttered with too much information and should be split into separate tables (or moved to the supplementary information), while others contain redundant information and should be consolidated. Some suggestions are offered in the detailed comments.

The formatting of the table also makes it hard to read, as the table is cut off at the end of the page. Although the formatting will be edited in the final manuscript, for the benefit of the reviewer, the authors should ensure that each table fits onto a single page.

**Equations:**

Many equations lack proper introduction and explanation. The authors should also carefully check the manuscript and make sure proper subscripts and superscripts are used for the various equation terms.

**Detailed comments:**

**Lines 109-113:** The description of the process is unclear. It sounds like a total of 35 liters of seawater was collected from various depths and filled into two containers. From those two containers, 25 liters of seawater were drawn and filled into a single container to produce a single batch of seawater. Was the seawater homogenized before filling into the two containers and/or again after filling into the 25 liter container?

**Line 117:** "Artificial seawater" is a misnomer because the background medium is sodium chloride only without the other major seawater salts.

**Table 1:**

Are the percent purity values listed in the table from the manufacturer or from the assay results at NMIJ and SMU? If they are assay values, they should be listed in a separate column or a separate table rather than in parentheses after the manufacturer name.

What are the likely impurities in the salts, and were the impurities assessed?

Consider changing the phrasing in the caption from "artificial solution for total alkalinity reference material" to "artificial total alkalinity reference material."

Two batches of artificial RM were produced, yet Table 1 only lists amount contents and molalities for one batch. Which batch do these values refer to? Also consider combining **Table 1** with **Table 3**. The background alkalinity of the NaCl should also be explicitly listed in the consolidated table.

**Line 122:** The use of pHT may not be appropriate in a NaCl medium as it does not contain sulfate. The temperature and dissociation constants used to calculate pH should also be given. The text indicates that the pH was estimated roughly based on a Bjerrum plot. The authors should provide a more precisely calculated value (especially if listing the pH in **Table 1**) or exclude the information about pH altogether, as it isn't strictly necessary to report for this reference material.

**Line 159:** "Material described in Appendix B" – Is this referring to the HCl standardized at SMU? Please state explicitly to avoid confusion and reference Appendix B for the details.

**Lines 163-164 and Lines 523-524:**

**Lines 523-524** indicates the possibility of background alkalinity in the other salts such as NaHCO3 and Na2CO3. Was this assessed?

What would be the intercept in **Fig. 2** if the linear regression was not forced to zero? Might the choice to force the regression to zero discard information about the background alkalinity from the NaHCO3 and Na2CO3?

Line 190, 193 – "Means of standard deviation" – If pooling standard deviations with uniform sample sizes, it should be calculated as the square root of the mean of the variances.

**Line 201-204:** Change phrasing to "ratio of the slope to the standard deviation of the slope." Also consider changing the notation so that the Student's t-value is not confused with t for time.  $\alpha$  and 0 should be in subscripts. This comment also applies to **Table 4.**

**Equation 8:** Aren't the salts added as stock solutions? In this case, **Eq. 8** should have  $m_{\text{stock}}$  instead of  $m_{\text{salt}}$ .  $m_{\text{total}}$  should be the sum of the stock solutions plus additional water rather than the sum of the salts and water.

**Line 261:** "To maximize the uncertainty of the slope" suggests that the goal was to have a larger uncertainty. I think what was meant was that the first approach with the larger uncertainty estimate was selected as the more conservative estimate of the uncertainty of the slope.

**Equation 11:** The equation for the homogeneity uncertainty does not make sense to me. It appears to be a standard deviation of the mean, but if so, it should be  $s/\sqrt{n}$ . However, this would not make sense either as the between-bottle variability was estimated differently for the different batches—some batches using the standard deviation of single measurements from different bottles and another batch using the standard deviation of the bottle means from repeatability measurements. It also does not make sense why the within-bottle homogeneity was neglected in the overall homogeneity uncertainty, as the within-bottle homogeneity was explicitly estimated and listed in **Table 4.**

The observed between bottle variance should be a sum of the within bottle variance and the homogeneity variance. If the between bottle variance is calculated as the standard deviation of the means from repeatability measurements in different bottles, then

$$s_{obs,bet-btl} = \sqrt{u_{\text{hom}}^2 + \frac{s_{repeatability}^2}{n}}$$

where n is the number of repeatability measurements within a single bottle.

As the other reviewer noted, the within and between bottle homogeneity components can be evaluated with ANOVA according to ISO Guide 35. It would be beneficial for many readers who do not have access to the ISO documents to derive these equations at a high level.

Eq. 12 and 13: The equations for the stability uncertainty require more explanation for the reader. Two equations are used. In the case of no significant trend, the stability uncertainty only has one contribution from the uncertainty of the slope  $b_1$ , while for cases with significant trends, the stability uncertainty has an additional rectangular distribution component.

Also, what value is used for time *t*? And as noted before, this notation can be confused with the Student's *t* value.

Line 311-313: The phrasing in this sentence is confusing. The median of the set of means (from repeatability measurements made by each participant) was calculated for two different materials—natural seawater and the artificial RM (Batch 1).

Line 325: I recommend replacing "samples" with "materials" to be clear that it was two different materials being analyzed and not two bottles.

**Equation 14:** How is the mean  $\overline{(X_l - Y_l)}$  calculated? Is it the mean difference from the 5 participants? Please clarify in the text. Why does the mean have a different subscript l instead of i, and what does it indicate?

**Equation 15:** There seems to be missing text that should precede this equation. The text following the equation states that  $s_r$  is the intra-laboratory standard deviation divided by the square root of the mean number of replicates – this should be explicitly written in the equation.

**Equations 15 and 16:** Both of these equations need proper introduction for the reader. They are based on the equations in ISO 21748. As some readers may not have access to the ISO documents, it would be helpful to provide an explanation of these equations and how the inter and intra-laboratory standard deviations are calculated.

The term  $u(\hat{\mu})$  needs further explanation than simply "standard deviation of the certified reference value." It is the uncertainty of the reference value which includes contributions from the characterization of the salts, the homogeneity, and stability (**Equation 6**). The notation should be revised so that it is consistent with **Equation 6**.

**Table 4:** The organization of this table is very confusing. Some entries are standard uncertainties with units of μmol kg-1, while others are not. The caption indicates all numerical values have units of μmol kg-1, but this is not true for parameters such as the slope, slope standard deviation, and Student *t* values. The stability uncertainty (from **Equation 12** and **13**) are not listed in this table. At the very least, the authors should clearly indicate in the table which parameters are standard uncertainties and include the appropriate units for each parameter. A better approach, I think, is to limit the table to only one type of information (e.g., the standard uncertainties associated with homogeneity, stability over time, and stability to transport). Additional details on the stability evaluation (e.g., the slope, *t*-tests, etc.) can be described in a separate table. The

authors could also consider combining the information on the homogeneity and stability uncertainties from **Table 4** and **Table 5**, although the natural seawater reference material does not have a certified value and an associated uncertainty.

- **Table 5:** Although **Equation 11** will need to be revised, I will point out that the values listed for the homogeneity uncertainty do not agree with **Equation 11** if using the between bottle standard deviations in Table 4 as s and N = 3. The authors should check their calculations in the tables.
- **Line 461:**  $S_R$  is the reproducibility standard deviation. This term should be introduced and defined much earlier in Section 2.4.1, where it is used in **Equation 16.**
- **Line 464:** Replace "precision" with "reproducibility standard deviation" to be clear what quantity is being reported. The term "precision" should be reserved for qualitative descriptions. It may also be informative to report the reproducibility standard deviation excluding Laboratory 1, as their measurements were discovered to have a systematic error due to malfunctioning of the titrant delivery on their measurement system.
- **Table 6:** The numeric values should not be left in E+00 notation form The units also need to be specified for the standard uncertainties.
- Section 3.3: This section deserves more discussion of the results rather than just a description of the data contained in **Table 6** and **Table 7**. Consider splitting this section into two—one discussing the top-down uncertainty estimates and the other discussing the bottom-up estimates.
- **Table 7:** The formatting of this table needs much reworking to improve readability. The last column lists the individual standard uncertainties for the sub-sources of uncertainty and a combined standard uncertainty for the input parameter all in the same column. These should be separate columns. Other information is also needed such as the sensitivity coefficients used in the uncertainty propagation, and the degrees of freedom for each uncertainty contribution. Consider including a more condensed table of the uncertainty budget in the main manuscript and a more detailed version in the supplementary information.
- **Lines 578-593:** This section discussed leaching of silicate from the borosilicate glass bottles as a potential cause for instability in some batches of reference material. It would be beneficial for other reference material producers and for future investigations to provide more details on the specifications of the borosilicate glass used (such as the manufacturer and coefficient of linear expansion of the glass), as well as any cleaning procedures performed before bottling. Were the bottles cleaned in any way? **Line 140** states that Schott borosilicate bottles were used for Batch 2 of the artificial reference material. What about the other batches?
- **Fig. 4:** This figure is rather confusing and not very informative. Although it highlights some major sources uncertainty such as the measured potential and the volume of acid delivered, it doesn't include all the sources of uncertainty in **Table 7** and their magnitudes. A bar graph providing a visual summary of **Table 7** may be a better choice.

**Equation A2:**

The total hydrogen ion concentration  $[H^+]_T$  on the total pH scale includes free hydrogen ions and bisulfate ions only (Dickson, 1993).  $[H^+] + [HSO_4^-] + [HF]$  is the total hydrogen ion concentration on the *seawater* pH scale. The notation should be revised in this equation.

---

## Author Comment (AC1)

**Response to reviewers – RC1**

OS manuscript egusphere-2025-3588

Title: Metrological concepts applied to Total Alkalinity measurements in seawater: reference materials, inter-laboratory comparison and uncertainty budget

Auhors: Gaëlle Capitaine, Samir Alliouane, Thierry Cariou, Jonathan Fin, Paola Fisicaro, Thibaut Wagener

**Dear reviewer,**

We would like to greatly thank you for taking the time to review our manuscript, and giving us the opportunity to improve it thanks to your valuable comments and suggestions. We have carefully considered all your comments. You will find below how we addressed them in the revised manuscript.

The line numbers correspond to the reviewed manuscript with changes marked.

| Line              | Comment                                                                                                                                                                                                                                                                                                                                                                                                                                                                                                                                                                                                                                                                                                                                                                                                                                                                                                                                                                                                                                                                                                                                                                                                                                             | Response                                                                                                                                                                                                                                                                                                                                                                                                                                                                                                                                                                                                                                                                                                                                                                 |
|-------------------|-----------------------------------------------------------------------------------------------------------------------------------------------------------------------------------------------------------------------------------------------------------------------------------------------------------------------------------------------------------------------------------------------------------------------------------------------------------------------------------------------------------------------------------------------------------------------------------------------------------------------------------------------------------------------------------------------------------------------------------------------------------------------------------------------------------------------------------------------------------------------------------------------------------------------------------------------------------------------------------------------------------------------------------------------------------------------------------------------------------------------------------------------------------------------------------------------------------------------------------------------------|--------------------------------------------------------------------------------------------------------------------------------------------------------------------------------------------------------------------------------------------------------------------------------------------------------------------------------------------------------------------------------------------------------------------------------------------------------------------------------------------------------------------------------------------------------------------------------------------------------------------------------------------------------------------------------------------------------------------------------------------------------------------------|
| General comment 1 | The traceability of the TA values and their uncertainties - which must be the core element of any RM, particularly if the authors, as they claim, worked in accordance with ISO 17034 – are inconsistent to some extent and are not sufficiently discussed. As the authors have noted themselves, TA measurements using the conventional method according to Dickson's Guide have no proper traceability. A detailed analysis of its traceability is currently lacking. Nevertheless, the authors use this method to characterize TA of the natural seawater TA. Moreover, even though the gravimetric approach to characterize the artificial RM is traceable to the SI, the quantification of impurities in NaCl is conducted with the common method, which introduces an inconsistency into the characterization. Additionally - as the authors themselves note - the artificial seawater differs chemically from natural seawater. This also affects the regression method used to determine TA and thus its associated measurement uncertainty. Consequently, measurements of seawater TA that are referred to the artificial RM may still contain significant biases, even though the comparison measurements show reasonably good agreement. | traceability" has been added at the end of the discussion section: "Metrological traceability is defined as the "property of a measurement whereby the result can be related to a reference through a documented unbroken chain of calibrations, each contributing to the measurement uncertainty" (JCGM 200:2012, 2012). The absence of an uncertainty budget associated to the measurement results and to the TA value of the reference materials currently distributed by the Scripps Institution of Oceanography prevents proper traceability of the measurement results. By developing an artificial reference material with a reference value accompanied by a complete uncertainty budget, as well as by providing an initial estimation of the uncertainty in TA |

Beyond these more fundamental issues, the overall concept of traceability that the authors seem to have in mind is not presented clearly. Traceability can fundamentally be established only to one metrological reference—either to the artificial or to the natural seawater RM. However, two very different RMs are introduced: the artificial RM. characterized gravimetrically, and the natural seawater RM, measured using the standard method according to Dickson's SOP 3a/b. The first is proposed to establish SI traceability while the latter is proposed to serve as an additional reference for quality control. However, what if this kind of quality control provides a deviating result, which is the RM to believe? In turn, if both RM provide compatible results, why is there a need for second kind of RM?

The manuscript is somewhat vague with respect to those traceability issues. The authors are not expected to solve these difficult, general problems of traceability in this paper, which would likely be beyond the scope of the study that mainly aims to evaluate the RMs, which has sufficient value in its own. Nevertheless, they must discuss and contextualize both RMs in light of these traceability challenges and clearly define their respective limitations.

traceability. To fully establish traceability, it will be necessary to quantify the background alkalinity in the artificial reference material in a more robust manner, and to evaluate the uncertainty associated with the NLLS regression. A proposed traceability route, based on the two reference materials developed, is presented in Capitaine et al. (in preparation)." We agree this is a highly important subject to discuss. As this is not the core of this paper, we planned to publish a specific opinion paper on the proposal of an enhanced route of traceability for TA measurement results: Capitaine, G., Fisicaro, P., and Wagener, T.: Towards improved metrological traceability of seawater Total Alkalinity measurements: advancing assessment of Ocean Alkalinity Enhancement, In preparation." See also the added comment line 460.

**General comment 2**

The structure of the paper should be reconsidered. The typical format theory, results, discussion—makes this paper rather confusing, since each of the three sections successively addresses several different topics. There are two RMs. two characterizations, homogeneity stability studies, and two comparison measurements. For each, the theoretical background is first explained, then all results are presented together,

Taking your comment into consideration we chose to modify the structure of the paper with three main sections —

Development of reference materials — Inter-laboratory comparison — Uncertainty budget; each of them containing relevant materials & methods, results and discussion subsections.

We did not separated the artificial RM from the natural one to avoid

finally everything is revisited for repetitions, mainly for the methods discussion. As a result, the reader easily of homogeneity and stability loses track of which parts belong studies. We have also added a small together and is constantly forced to flip back and forth between sections. section on traceability (see This is not a mandatory requirement to comment above). addressed for publication. However, to the reviewer's opinion readability would be improved if the authors restructure the paper by addressing the artificial seawater RM first—covering its preparation, characterization. stability, and homogeneity, including the relevant calculation principles and results, and concluding with the discussion. The same should then be done for the natural seawater RM. Finally, the comparison measurements should be presented, allowing both RMs to be contrasted, and a proposal for traceability should be discussed in more detail, also from a practical perspective. Additionally, the paper is rather long, considering that it essentially evaluates RMs well-established using procedures. The authors should consider shortening it to some extent. General The language has been improved The reviewer recommends using an comment 3 LLM-based AI to improve the in parts were it was difficult to language. In parts, the paper is read but without using an AI based difficult to read due to linguistic tool. weaknesses, which the reviewer did not further correct. 47 How do uncertainty limits illustrate The value has been rephrased as climatic variations in TA? Please "These values were chosen in order to obtain a 1% standard rephrase for more clarity. uncertainty in the computation of the carbonate ion amount content variable, enabling to highlight climatic variations in the monitoring of ocean acidification."

| 56           | (Nat fully top ===1=1=2) in ===+=====     | The leak of we contain to 111                                    |
|--------------|-------------------------------------------|------------------------------------------------------------------|
| 56           | "Not fully traceable" is a strong         | The lack of uncertainty budget attributed to the reference value |
|              | statement that requires discussion. If    | prevent from the establishment of the                            |
|              | characterized HCl was used for the        | traceability. The sentence has been                              |
|              | titration, why is the measurement not     | changed accordingly.                                             |
|              | traceable to the SI? This should be       | changed accordingly.                                             |
|              | discussed — if not in the introduction,   |                                                                  |
|              | then elsewhere in the manuscript.         |                                                                  |
| 59-61        | The metrological terminology is           | According to your comments, the                                  |
|              | somewhat imprecise. Comparability of      | sentence has been rephrased as                                   |
|              | results is achieved through traceability  | "Moreover, the uncertainty budget                                |
|              | to the same metrological reference, not   | of the measurement method results                                |
|              | through                                   | is required to check the                                         |
|              | uncertainty. Measurement uncertainty      | compatibility of total alkalinity                                |
|              | defines the limits within which           | values."                                                         |
|              | differences between measurement           | Moreover, "improving the trueness                                |
|              | results — or their equivalence —          | of the results" has been replaced by                             |
|              | become meaningful (see also VIM:          | "assessing more robustly an                                      |
|              | compatibility). Only deviations           | eventual measurement bias".                                      |
|              | exceeding the measurement                 |                                                                  |
|              | uncertainty can be regarded as            |                                                                  |
|              | significant.                              |                                                                  |
|              | Similarly, the term "uncertainty of a     |                                                                  |
|              | measurement method" is incorrect — a      |                                                                  |
|              | method itself has no uncertainty; only a  |                                                                  |
|              | measured value has one.                   |                                                                  |
|              | The authors should also verify            |                                                                  |
|              | whether the word "trueness" in line       |                                                                  |
|              |                                           |                                                                  |
|              | 56 expresses what they intend to say.     |                                                                  |
|              | According to the VIM, trueness            |                                                                  |
|              | refers to "the closeness of agreement     |                                                                  |
|              | between the average of an infinite        |                                                                  |
|              | number of replicate measured              |                                                                  |
|              | quantity values and a reference           |                                                                  |
|              | quantity value." I am not sure, if this   |                                                                  |
|              | meant.                                    |                                                                  |
| 63           | ISO Guide 35 has been replaced by ISO     | The manuscript has been changed                                  |
|              | 33405. A paper related to metrological    | accordingly.                                                     |
|              | science should not refer to outdated      |                                                                  |
|              | standards.                                |                                                                  |
| Sect 2, 1.91 | Section 2.1: The purpose of this brief    | As different methods exists for the                              |
| ,            | summary of Dickson's SOP 3b is not        | measurement of seawater TA (open                                 |
|              | clear. Usually, reference to Dickson's    | or closed cells, multi-step or single                            |
|              | Guide would be suffice, all the more,     | addition of acid), the brief                                     |
|              | the paper is already quite long. If there | description allows to clearly state                              |
|              |                                           | -                                                                |
|              | is a reason for the repetition, it should | on which method the paper focuses                                |
|              | mentioned. I                              | on.                                                              |
|              | assume the formulas are                   | Indeed, the description of the                                   |

|           | mentioned because it is relevant              | measurement model is needed for                                    |
|-----------|-----------------------------------------------|--------------------------------------------------------------------|
|           | for the uncertainty calculations in           | the uncertainty determination. This                                |
|           | subsequent sections?                          | has been specified in the text.                                    |
|           |                                               |                                                                    |
|           |                                               |                                                                    |
| 172       | The measurement result at zero NaCl           | The manuscript has been changed                                    |
| 172       | mol/kg sol is shown                           | accordingly.                                                       |
| 173       | Figure 2 is mentioned in the main text        | Former Figure 2 now appears as                                     |
|           | before Figure 1. Figures should be cited      | Figure 1 due to the restructuration of                             |
|           | in the order of their appearance.             | the paper. The numbers of the                                      |
|           |                                               | figures have been replaced                                         |
|           |                                               | accordingly.                                                       |
| 173       | Replace "theory" with "reasonable             | The manuscript has been changed                                    |
|           | assumption."                                  | accordingly.                                                       |
| 175 & 433 | Even with goodwill, Fig. 2 does not           | The fact that both the gravimetric                                 |
|           | support the assumption of a linear            | and potentiometric approaches yield                                |
|           | relationship passing through the origin.      | ΔTA=0 μmol kg-1 at zero NaCl                                       |
|           | It rather shows a square root like            | content supports the internal consistency of the measurements.     |
|           | behavior, which is difficult to explain.      | We acknowledge, however, that the                                  |
|           | Alternatively, the $\Delta TA$ values at 1, 2 | data presented in Figure 2 do not                                  |
|           | and 3 mol/kg NaCl solution content            | perfectly support a linear                                         |
|           | indeed suggest that there is a linear         | relationship passing through the                                   |
|           | relationship - which one can expect in        | origin, and we agree that this aspect                              |
|           | dependence of NaCl content – but              | warrants improvement. This                                         |
|           | with an offset at zero NaCl content.          | limitation has been clarified and further discussed in the revised |
|           | Which raises the question, why the            | manuscript (Sect. 3.1.2 and 3.3.2)                                 |
|           | measured ΔTA value is zero at zero            | manuscript (Sect. 3.1.2 and 3.3.2)                                 |
|           | NaCl content? I suspect, that the             |                                                                    |
|           | reason for this discrepancy can be            |                                                                    |
|           | found in the different metrological           |                                                                    |
|           | references involved in the gravimetric        |                                                                    |
|           | and measured TA values. See also              |                                                                    |
|           | comments related to lines 236 and             |                                                                    |
|           | 522.                                          |                                                                    |
|           | A 1 11 11 11 11 11 11 11 11 11 11 11 11       |                                                                    |
|           | Anyhow, the linear extrapolation might        |                                                                    |
|           | be used as a rough estimate for the           |                                                                    |
|           | background alkalinity. However, the           |                                                                    |
|           | authors must comment on the                   |                                                                    |
| 170 % 192 | difficulties I have mentioned.                | Dreatical calinity is neaded for                                   |
| 179 & 183 | The purpose of measuring practical            | Practical salinity is needed for                                   |
|           | salinity and dissolved nutrients should       | computing total alkalinity with the                                |
|           | be stated.                                    | NLLS regression and the monitoring                                 |
|           |                                               | of nutrients gives relevant                                        |
|           |                                               | information for stability assessment.                              |
| 21.4      | A 6                                           | This has been added to the text.                                   |
| 214       | A reference to ISO 33405 would be             | The manuscript has been changed                                    |
|           | more appropriate here.                        | accordingly.                                                       |

| 223 | The authors claim to evaluate the                                       | The manuscript has been changed                                                                                                                                                                                                                                                                                                                                                                                                                                                                                                                                                                                                                                                                                                                                                                                                                                                                                                                                                                                                                                                                                                                                                                                                                                                                                                                                                                                                                                                                                                                                                                                                                                                                                                                                                                                                                                                                                                                                                                                                                                                                                                |
|-----|-------------------------------------------------------------------------|--------------------------------------------------------------------------------------------------------------------------------------------------------------------------------------------------------------------------------------------------------------------------------------------------------------------------------------------------------------------------------------------------------------------------------------------------------------------------------------------------------------------------------------------------------------------------------------------------------------------------------------------------------------------------------------------------------------------------------------------------------------------------------------------------------------------------------------------------------------------------------------------------------------------------------------------------------------------------------------------------------------------------------------------------------------------------------------------------------------------------------------------------------------------------------------------------------------------------------------------------------------------------------------------------------------------------------------------------------------------------------------------------------------------------------------------------------------------------------------------------------------------------------------------------------------------------------------------------------------------------------------------------------------------------------------------------------------------------------------------------------------------------------------------------------------------------------------------------------------------------------------------------------------------------------------------------------------------------------------------------------------------------------------------------------------------------------------------------------------------------------|
|     | proposed RMs in accordance with                                         | accordingly.                                                                                                                                                                                                                                                                                                                                                                                                                                                                                                                                                                                                                                                                                                                                                                                                                                                                                                                                                                                                                                                                                                                                                                                                                                                                                                                                                                                                                                                                                                                                                                                                                                                                                                                                                                                                                                                                                                                                                                                                                                                                                                                   |
|     | ISO 17034. If so, they must fulfill                                     |                                                                                                                                                                                                                                                                                                                                                                                                                                                                                                                                                                                                                                                                                                                                                                                                                                                                                                                                                                                                                                                                                                                                                                                                                                                                                                                                                                                                                                                                                                                                                                                                                                                                                                                                                                                                                                                                                                                                                                                                                                                                                                                                |
|     | the experimental requirements for                                       |                                                                                                                                                                                                                                                                                                                                                                                                                                                                                                                                                                                                                                                                                                                                                                                                                                                                                                                                                                                                                                                                                                                                                                                                                                                                                                                                                                                                                                                                                                                                                                                                                                                                                                                                                                                                                                                                                                                                                                                                                                                                                                                                |
|     | short-term stability testing. Using a                                   |                                                                                                                                                                                                                                                                                                                                                                                                                                                                                                                                                                                                                                                                                                                                                                                                                                                                                                                                                                                                                                                                                                                                                                                                                                                                                                                                                                                                                                                                                                                                                                                                                                                                                                                                                                                                                                                                                                                                                                                                                                                                                                                                |
|     | single, undefined transport of the                                      |                                                                                                                                                                                                                                                                                                                                                                                                                                                                                                                                                                                                                                                                                                                                                                                                                                                                                                                                                                                                                                                                                                                                                                                                                                                                                                                                                                                                                                                                                                                                                                                                                                                                                                                                                                                                                                                                                                                                                                                                                                                                                                                                |
|     | RM does not meet these                                                  |                                                                                                                                                                                                                                                                                                                                                                                                                                                                                                                                                                                                                                                                                                                                                                                                                                                                                                                                                                                                                                                                                                                                                                                                                                                                                                                                                                                                                                                                                                                                                                                                                                                                                                                                                                                                                                                                                                                                                                                                                                                                                                                                |
|     | requirements. Since this uncertainty                                    |                                                                                                                                                                                                                                                                                                                                                                                                                                                                                                                                                                                                                                                                                                                                                                                                                                                                                                                                                                                                                                                                                                                                                                                                                                                                                                                                                                                                                                                                                                                                                                                                                                                                                                                                                                                                                                                                                                                                                                                                                                                                                                                                |
|     | contribution can presumably not be                                      |                                                                                                                                                                                                                                                                                                                                                                                                                                                                                                                                                                                                                                                                                                                                                                                                                                                                                                                                                                                                                                                                                                                                                                                                                                                                                                                                                                                                                                                                                                                                                                                                                                                                                                                                                                                                                                                                                                                                                                                                                                                                                                                                |
|     | readily quantified, I recommend                                         |                                                                                                                                                                                                                                                                                                                                                                                                                                                                                                                                                                                                                                                                                                                                                                                                                                                                                                                                                                                                                                                                                                                                                                                                                                                                                                                                                                                                                                                                                                                                                                                                                                                                                                                                                                                                                                                                                                                                                                                                                                                                                                                                |
|     | refraining from claiming that this value                                |                                                                                                                                                                                                                                                                                                                                                                                                                                                                                                                                                                                                                                                                                                                                                                                                                                                                                                                                                                                                                                                                                                                                                                                                                                                                                                                                                                                                                                                                                                                                                                                                                                                                                                                                                                                                                                                                                                                                                                                                                                                                                                                                |
|     | has been                                                                |                                                                                                                                                                                                                                                                                                                                                                                                                                                                                                                                                                                                                                                                                                                                                                                                                                                                                                                                                                                                                                                                                                                                                                                                                                                                                                                                                                                                                                                                                                                                                                                                                                                                                                                                                                                                                                                                                                                                                                                                                                                                                                                                |
|     | determined. Instead, it should be stated                                |                                                                                                                                                                                                                                                                                                                                                                                                                                                                                                                                                                                                                                                                                                                                                                                                                                                                                                                                                                                                                                                                                                                                                                                                                                                                                                                                                                                                                                                                                                                                                                                                                                                                                                                                                                                                                                                                                                                                                                                                                                                                                                                                |
|     | that the value represents a first                                       |                                                                                                                                                                                                                                                                                                                                                                                                                                                                                                                                                                                                                                                                                                                                                                                                                                                                                                                                                                                                                                                                                                                                                                                                                                                                                                                                                                                                                                                                                                                                                                                                                                                                                                                                                                                                                                                                                                                                                                                                                                                                                                                                |
|     | estimate, while a proper evaluation                                     |                                                                                                                                                                                                                                                                                                                                                                                                                                                                                                                                                                                                                                                                                                                                                                                                                                                                                                                                                                                                                                                                                                                                                                                                                                                                                                                                                                                                                                                                                                                                                                                                                                                                                                                                                                                                                                                                                                                                                                                                                                                                                                                                |
|     | according to ISO 17034 is still                                         |                                                                                                                                                                                                                                                                                                                                                                                                                                                                                                                                                                                                                                                                                                                                                                                                                                                                                                                                                                                                                                                                                                                                                                                                                                                                                                                                                                                                                                                                                                                                                                                                                                                                                                                                                                                                                                                                                                                                                                                                                                                                                                                                |
|     | pending.                                                                |                                                                                                                                                                                                                                                                                                                                                                                                                                                                                                                                                                                                                                                                                                                                                                                                                                                                                                                                                                                                                                                                                                                                                                                                                                                                                                                                                                                                                                                                                                                                                                                                                                                                                                                                                                                                                                                                                                                                                                                                                                                                                                                                |
| 228 | "ISO Guide 35" — see comment on                                         | The manuscript has been changed                                                                                                                                                                                                                                                                                                                                                                                                                                                                                                                                                                                                                                                                                                                                                                                                                                                                                                                                                                                                                                                                                                                                                                                                                                                                                                                                                                                                                                                                                                                                                                                                                                                                                                                                                                                                                                                                                                                                                                                                                                                                                                |
|     | line 59.                                                                | accordingly.                                                                                                                                                                                                                                                                                                                                                                                                                                                                                                                                                                                                                                                                                                                                                                                                                                                                                                                                                                                                                                                                                                                                                                                                                                                                                                                                                                                                                                                                                                                                                                                                                                                                                                                                                                                                                                                                                                                                                                                                                                                                                                                   |
| 251 | It is unclear whether the calculation of                                | The calculation of the bias                                                                                                                                                                                                                                                                                                                                                                                                                                                                                                                                                                                                                                                                                                                                                                                                                                                                                                                                                                                                                                                                                                                                                                                                                                                                                                                                                                                                                                                                                                                                                                                                                                                                                                                                                                                                                                                                                                                                                                                                                                                                                                    |
|     | the bias introduced by NaCl impurities                                  | introduced by NaCl impurities is                                                                                                                                                                                                                                                                                                                                                                                                                                                                                                                                                                                                                                                                                                                                                                                                                                                                                                                                                                                                                                                                                                                                                                                                                                                                                                                                                                                                                                                                                                                                                                                                                                                                                                                                                                                                                                                                                                                                                                                                                                                                                               |
|     | is used solely to correct the reference                                 | intended to quantify the contribution                                                                                                                                                                                                                                                                                                                                                                                                                                                                                                                                                                                                                                                                                                                                                                                                                                                                                                                                                                                                                                                                                                                                                                                                                                                                                                                                                                                                                                                                                                                                                                                                                                                                                                                                                                                                                                                                                                                                                                                                                                                                                          |
|     | value or to quantify its contribution to                                | of the background alkalinity to the                                                                                                                                                                                                                                                                                                                                                                                                                                                                                                                                                                                                                                                                                                                                                                                                                                                                                                                                                                                                                                                                                                                                                                                                                                                                                                                                                                                                                                                                                                                                                                                                                                                                                                                                                                                                                                                                                                                                                                                                                                                                                            |
|     | the                                                                     | total TA value of the reference                                                                                                                                                                                                                                                                                                                                                                                                                                                                                                                                                                                                                                                                                                                                                                                                                                                                                                                                                                                                                                                                                                                                                                                                                                                                                                                                                                                                                                                                                                                                                                                                                                                                                                                                                                                                                                                                                                                                                                                                                                                                                                |
|     | uncertainty. This must be explicitly                                    | material. This was made clearer in                                                                                                                                                                                                                                                                                                                                                                                                                                                                                                                                                                                                                                                                                                                                                                                                                                                                                                                                                                                                                                                                                                                                                                                                                                                                                                                                                                                                                                                                                                                                                                                                                                                                                                                                                                                                                                                                                                                                                                                                                                                                                             |
|     | stated to avoid confusion. In any                                       | the text. The uncertainty associated                                                                                                                                                                                                                                                                                                                                                                                                                                                                                                                                                                                                                                                                                                                                                                                                                                                                                                                                                                                                                                                                                                                                                                                                                                                                                                                                                                                                                                                                                                                                                                                                                                                                                                                                                                                                                                                                                                                                                                                                                                                                                           |
|     | case, the approach appears to lead to                                   | with this term is incorporated into                                                                                                                                                                                                                                                                                                                                                                                                                                                                                                                                                                                                                                                                                                                                                                                                                                                                                                                                                                                                                                                                                                                                                                                                                                                                                                                                                                                                                                                                                                                                                                                                                                                                                                                                                                                                                                                                                                                                                                                                                                                                                            |
|     | a circular argument regarding                                           | the overall uncertainty budget of the                                                                                                                                                                                                                                                                                                                                                                                                                                                                                                                                                                                                                                                                                                                                                                                                                                                                                                                                                                                                                                                                                                                                                                                                                                                                                                                                                                                                                                                                                                                                                                                                                                                                                                                                                                                                                                                                                                                                                                                                                                                                                          |
|     | traceability. The authors aim to                                        | assigned reference value.                                                                                                                                                                                                                                                                                                                                                                                                                                                                                                                                                                                                                                                                                                                                                                                                                                                                                                                                                                                                                                                                                                                                                                                                                                                                                                                                                                                                                                                                                                                                                                                                                                                                                                                                                                                                                                                                                                                                                                                                                                                                                                      |
|     | correct the bias and/or assign a                                        | As indicated in Equation (5),                                                                                                                                                                                                                                                                                                                                                                                                                                                                                                                                                                                                                                                                                                                                                                                                                                                                                                                                                                                                                                                                                                                                                                                                                                                                                                                                                                                                                                                                                                                                                                                                                                                                                                                                                                                                                                                                                                                                                                                                                                                                                                  |
|     | corresponding uncertainty to the                                        | $TA_{background}$ is added to the                                                                                                                                                                                                                                                                                                                                                                                                                                                                                                                                                                                                                                                                                                                                                                                                                                                                                                                                                                                                                                                                                                                                                                                                                                                                                                                                                                                                                                                                                                                                                                                                                                                                                                                                                                                                                                                                                                                                                                                                                                                                                              |
|     | RM. To do so, they measure TA and                                       | gravimetrically derived value, not                                                                                                                                                                                                                                                                                                                                                                                                                                                                                                                                                                                                                                                                                                                                                                                                                                                                                                                                                                                                                                                                                                                                                                                                                                                                                                                                                                                                                                                                                                                                                                                                                                                                                                                                                                                                                                                                                                                                                                                                                                                                                             |
|     | subtract this value from the one                                        | subtracted. Because the uncertainty                                                                                                                                                                                                                                                                                                                                                                                                                                                                                                                                                                                                                                                                                                                                                                                                                                                                                                                                                                                                                                                                                                                                                                                                                                                                                                                                                                                                                                                                                                                                                                                                                                                                                                                                                                                                                                                                                                                                                                                                                                                                                            |
|     | obtained via gravimetric                                                | of $TA_{background}$ has been explicitly                                                                                                                                                                                                                                                                                                                                                                                                                                                                                                                                                                                                                                                                                                                                                                                                                                                                                                                                                                                                                                                                                                                                                                                                                                                                                                                                                                                                                                                                                                                                                                                                                                                                                                                                                                                                                                                                                                                                                                                                                                                                                       |
|     | measurements. However, in order to                                      | quantified, its inclusion in the                                                                                                                                                                                                                                                                                                                                                                                                                                                                                                                                                                                                                                                                                                                                                                                                                                                                                                                                                                                                                                                                                                                                                                                                                                                                                                                                                                                                                                                                                                                                                                                                                                                                                                                                                                                                                                                                                                                                                                                                                                                                                               |
|     | measure the TA with proper                                              | reference value does not compromise                                                                                                                                                                                                                                                                                                                                                                                                                                                                                                                                                                                                                                                                                                                                                                                                                                                                                                                                                                                                                                                                                                                                                                                                                                                                                                                                                                                                                                                                                                                                                                                                                                                                                                                                                                                                                                                                                                                                                                                                                                                                                            |
|     | traceability, they would need a                                         | the reliability of the assignment.                                                                                                                                                                                                                                                                                                                                                                                                                                                                                                                                                                                                                                                                                                                                                                                                                                                                                                                                                                                                                                                                                                                                                                                                                                                                                                                                                                                                                                                                                                                                                                                                                                                                                                                                                                                                                                                                                                                                                                                                                                                                                             |
|     | characterized RM traceable to the same                                  | Regarding the question of                                                                                                                                                                                                                                                                                                                                                                                                                                                                                                                                                                                                                                                                                                                                                                                                                                                                                                                                                                                                                                                                                                                                                                                                                                                                                                                                                                                                                                                                                                                                                                                                                                                                                                                                                                                                                                                                                                                                                                                                                                                                                                      |
|     | metrological reference as the artificial                                | traceability, we believe this issue                                                                                                                                                                                                                                                                                                                                                                                                                                                                                                                                                                                                                                                                                                                                                                                                                                                                                                                                                                                                                                                                                                                                                                                                                                                                                                                                                                                                                                                                                                                                                                                                                                                                                                                                                                                                                                                                                                                                                                                                                                                                                            |
|     | RM — which is not available except                                      | does not apply in the way suggested.                                                                                                                                                                                                                                                                                                                                                                                                                                                                                                                                                                                                                                                                                                                                                                                                                                                                                                                                                                                                                                                                                                                                                                                                                                                                                                                                                                                                                                                                                                                                                                                                                                                                                                                                                                                                                                                                                                                                                                                                                                                                                           |
|     | for the proposed one. If, as assumed, the authors used Dickson's SOP to | The limitation mentioned in the                                                                                                                                                                                                                                                                                                                                                                                                                                                                                                                                                                                                                                                                                                                                                                                                                                                                                                                                                                                                                                                                                                                                                                                                                                                                                                                                                                                                                                                                                                                                                                                                                                                                                                                                                                                                                                                                                                                                                                                                                                                                                                |
|     | measure TA, then the traceability of the                                |                                                                                                                                                                                                                                                                                                                                                                                                                                                                                                                                                                                                                                                                                                                                                                                                                                                                                                                                                                                                                                                                                                                                                                                                                                                                                                                                                                                                                                                                                                                                                                                                                                                                                                                                                                                                                                                                                                                                                                                                                                                                                                                                |
|     | bias/uncertainty is subject to the same                                 | currently available reference                                                                                                                                                                                                                                                                                                                                                                                                                                                                                                                                                                                                                                                                                                                                                                                                                                                                                                                                                                                                                                                                                                                                                                                                                                                                                                                                                                                                                                                                                                                                                                                                                                                                                                                                                                                                                                                                                                                                                                                                                                                                                                  |
|     | limitations inherent to that SOP (as                                    | materials are not fully traceable                                                                                                                                                                                                                                                                                                                                                                                                                                                                                                                                                                                                                                                                                                                                                                                                                                                                                                                                                                                                                                                                                                                                                                                                                                                                                                                                                                                                                                                                                                                                                                                                                                                                                                                                                                                                                                                                                                                                                                                                                                                                                              |
|     | mentioned in the introduction). Thus,                                   | because they lack a rigorously                                                                                                                                                                                                                                                                                                                                                                                                                                                                                                                                                                                                                                                                                                                                                                                                                                                                                                                                                                                                                                                                                                                                                                                                                                                                                                                                                                                                                                                                                                                                                                                                                                                                                                                                                                                                                                                                                                                                                                                                                                                                                                 |
|     | traceability of the assigned TA value of                                | assessed uncertainty budget.                                                                                                                                                                                                                                                                                                                                                                                                                                                                                                                                                                                                                                                                                                                                                                                                                                                                                                                                                                                                                                                                                                                                                                                                                                                                                                                                                                                                                                                                                                                                                                                                                                                                                                                                                                                                                                                                                                                                                                                                                                                                                                   |
|     | the artificial RM, or its uncertainty,                                  | and state of the s |

respectively, is questionable.

|                      | Fundamentally, the bias/uncertainty must be quantified independently of the RM it is intended to characterize and with respect to the same metrological reference.                                                                                                                                                                                                                                                                                                                                                                                                                                                                                                                                                                                                                            |                                                                                                                                                                                                                                                                                                                                                                                                                                                                                                                                                                                                                                                                                                                                                                                                                                          |
|----------------------|-----------------------------------------------------------------------------------------------------------------------------------------------------------------------------------------------------------------------------------------------------------------------------------------------------------------------------------------------------------------------------------------------------------------------------------------------------------------------------------------------------------------------------------------------------------------------------------------------------------------------------------------------------------------------------------------------------------------------------------------------------------------------------------------------|------------------------------------------------------------------------------------------------------------------------------------------------------------------------------------------------------------------------------------------------------------------------------------------------------------------------------------------------------------------------------------------------------------------------------------------------------------------------------------------------------------------------------------------------------------------------------------------------------------------------------------------------------------------------------------------------------------------------------------------------------------------------------------------------------------------------------------------|
| 261-264              | "Systematic uncertainty sources, such as those arising from the device, the operator, or the procedure, are cancelled here." This statement is not self-evident. Which uncertainties cancel out, and how are they correlated? The authors should explain this important aspect in more detail.                                                                                                                                                                                                                                                                                                                                                                                                                                                                                                | The sentence "Since the same operator, instrument, and procedure were used to establish the relationship between $\Delta(TA_{measured} - TA_{theoretical})$ and $\nu_{NaCl}$ , these parameters contribute to systematic uncertainty sources. They are cancelled when establishing a trend and do not contribute to the uncertainty of the observed slope." Has been added for clarity.                                                                                                                                                                                                                                                                                                                                                                                                                                                  |
| 288-302 &
466-470 | "It was chosen to neglect the within-bottle homogeneity." Again, the authors claim compliance with ISO Guide 35 (ISO 33405, respectively); however, their homogeneity analysis appears superficial to some extent. A one-way ANOVA must be applied to account for both within-unit and between-unit homogeneity.  One might decide to disregard within-unit uncertainty for the reasons mentioned by the authors. In that case, only between-bottle homogeneity should be calculated according to the (corrected) Eq. 11. Otherwise, a proper one-way ANOVA analysis is expected for the homogeneity values given in Table 4. The authors must also evaluate the repeatability standard deviation of the homogeneity with respect to the target uncertainty (see Section 7.5.1 of ISO 33405). | A one-way ANOVA has been performed on the results obtained from the homogeneity testing. The ANOVA results were then used to calculate the between-bottle homogeneity uncertainty based on the corrected equation 11 (see comment below). The corresponding values of u hom have been corrected accordingly in the manuscript.  ISO 33405:2024 (Section 7.5.1 of the ISO), states that the repeatability standard deviation of the homogeneity study procedure should be less than one third of the target standard uncertainty of the TA measurement result for the procedure to be considered suitable. In our case, the criterion was slightly exceeded, but the results can nevertheless be regarded as a preliminary estimate of the material's homogeneity. This has been added in the discussion section 3.4.4. |
| 297                  | Equation 11 is incorrect. In the simplest approach, assuming within-unit homogeneity, $u_{\text{hom}} = M_{\text{between}}/n_0$ , where $M_{\text{between}}$ is the mean square of the TA results of the units and $n_0$ is the number of measurements per unit (assuming they are equal for each unit). See, for example, Section 7.7.3 and                                                                                                                                                                                                                                                                                                                                                                                                                                                  | The equation has been corrected based on the ISO 33405.                                                                                                                                                                                                                                                                                                                                                                                                                                                                                                                                                                                                                                                                                                                                                                                  |

|                      | Annex B1 of ISO 33405.                                                                                                                                                                                                                                                                                                                                                                                                                                 |                                                                                                                                                                                                                                                                                                                                                                                                                                                                     |
|----------------------|--------------------------------------------------------------------------------------------------------------------------------------------------------------------------------------------------------------------------------------------------------------------------------------------------------------------------------------------------------------------------------------------------------------------------------------------------------|---------------------------------------------------------------------------------------------------------------------------------------------------------------------------------------------------------------------------------------------------------------------------------------------------------------------------------------------------------------------------------------------------------------------------------------------------------------------|
| 305-311              | The equations are mutually inconsistent. $u_{\text{stab}}$ cannot comply with both Eq. 12 and Eq. 13. I would recommend referring to Eq. 11 in Section 8.7.3 and Annex B3 of ISO 33405 instead.                                                                                                                                                                                                                                                        | For consisteny, the equation 13 used for unstable materials is now noted $u_{stab'}$ .                                                                                                                                                                                                                                                                                                                                                                              |
| 513                  | Improve clarity: Do the authors mean that the artificial RM and the stabilized natural RM were prepared at approximately the same time, or one after the other?                                                                                                                                                                                                                                                                                        | They were prepared approximately at the same time. The manuscript has been changed accordingly.                                                                                                                                                                                                                                                                                                                                                                     |
| 521                  | Replace "calibrated" with "characterized."                                                                                                                                                                                                                                                                                                                                                                                                             | The manuscript has been changed accordingly.                                                                                                                                                                                                                                                                                                                                                                                                                        |
| 328                  | " and is included in the reference values given above." The meaning is unclear.  The authors should be more precise: do the TA values in Table 3 really include the TA contribution from NaCl impurities, meaning the bias has not been corrected, or do they mean it has been considered, meaning the values in Table 3 have been corrected for this bias? See also the comment on line 236.                                                          | The values are computed from equation 5, which indicates that $TA_{background}$ , coming from the NaCl matrix, is a contribution to the TA value of the reference material, not a bias.  Meaning the TA reference value is computed from the sum of the amount contents of Na 2 CO 3 and NaHCO 3 (gravimetric information) and of $TA_{background}$ (potentiometric determination).  This has been made clearer in the manuscript. |
| Table 3, 1.340       | Table 4: The authors should add the units more precisely, as not all quantities are given in μmol/kg.                                                                                                                                                                                                                                                                                                                                                  | The manuscript has been revised accordingly.                                                                                                                                                                                                                                                                                                                                                                                                                        |
| 336 & 466-
471    | Table 4: It was mentioned that within-unit homogeneity was neglected; nevertheless, corresponding values are shown. Moreover, the results suggest that within-unit variability is even larger than between-bottle homogeneity, which seems unlikely. This supports the recommendation that measurement repeatability should be assessed in relation to the evaluation of homogeneity, stability and target uncertainty (also see comment on line 271). | The uncertainty of within-bottle homogeneity wasn't calculated but results of the standard deviations from the homogeneity study. Two subsections have been added to more clearly separate homogeneity & stability tests results from the uncertainty quantification.  The discussion about measurement repeatability has been added in section 3.3.4.                                                                                                              |
| 346-350,
Figure 2 | "Its stability has been studied" I recommend adding a figure to illustrate the stability results.                                                                                                                                                                                                                                                                                                                                                      | The mentioned stability study (Artificial solution – batch 2) has been added as Figure 1.                                                                                                                                                                                                                                                                                                                                                                           |

| _         | T                                               | ,                                                                                               |
|-----------|-------------------------------------------------|-------------------------------------------------------------------------------------------------|
| 581 & 595 | Results should not be excluded solely           | Yes, it was mentioned in the                                                                    |
|           | for statistical reasons, especially             | discussion of the ILC:                                                                          |
|           | when only a small number of                     | "Laboratory 1 later reported that                                                               |
|           | participants is involved. Have                  | the acid injection system was not                                                               |
|           | potential causes of                             | functioning properly during the                                                                 |
|           | deviation related to the measurement            | ILC. A leak at the microvalve,                                                                  |
|           | itself been investigated?                       | leading to inconsistent acid                                                                    |
|           |                                                 | delivery volumes, could explain                                                                 |
|           |                                                 | the observed bias." It is now also                                                              |
|           |                                                 | mentioned in this results section.                                                              |
| Table 5   | Table 6: Using Eq. 16, the values for $S_L$     | Thank you for checking, the value is                                                            |
|           | and $s_r$ given in Table 6, $n = 3$ , $p = 4$ , | indeed wrong. After checking the                                                                |
|           | and                                             | calculation, we also found the value                                                            |
|           | $(\mu) = 1.08$ , I calculate $(\Delta) = 1.40$  | of $(\Delta) = 1.40 \mu\text{mol/kg}$ . The                                                     |
|           | $\mu$ mol/kg. The authors should verify the     | manuscript was changed                                                                          |
|           | values, or clarify which numbers were           | accordingly.                                                                                    |
|           | used in their calculation.                      |                                                                                                 |
| Table 6   | Tables 6 and 7 are nearly identical. I          | The manuscript was changed                                                                      |
|           | recommend combining them.                       | accordingly.                                                                                    |
| 388-392 & | Since natural seawater consists of              | As the minor ions in natural                                                                    |
| 403       | around 90 % NaCl, it is unlikely that           | seawater are mostly bivalent (like                                                              |
|           | the                                             | Ca 2+ , Mg 2+ and SO 4 2- ) whereas Na + |
|           | difference in composition between               | and Cl - are monovalent ions, it make                                                |
|           | natural and artificial seawater could           | sense that the practical salinity (which is defined by a conductivity                           |
|           | account for a tenfold discrepancy               | ratio) of a solution composed only of                                                           |
|           | between the expected and observed               | NaCl would be less than one of a                                                                |
|           | differences in                                  | natural seawater, although it may not                                                           |
|           | practical and absolute salinity. In any         | explain the entire discrepancy. The                                                             |
|           | case, it is unclear why this matters. The       | sentence was rephrased as "The                                                                  |
|           | authors should clarify the relationship         | composition of the artificial                                                                   |
|           | between TA and salinity to make the             | seawater being composed in high                                                                 |
|           | relevance of this discrepancy for the           | majority of NaCl, may explain the higher discrepancy between practical                   |
|           | study apparent.                                 | and absolute salinity observed".                                                                |
|           |                                                 | The following sentences have also                                                               |
|           |                                                 | been added: "For natural seawater,                                                              |
|           |                                                 | knowledge of salinity is required to                                                            |
|           |                                                 | determine the TA value through                                                                  |
|           |                                                 | NLLS regression, which accounts                                                                 |
|           |                                                 | for the competing acid–base                                                                     |
|           |                                                 | equilibria present in seawater. In the                                                          |
|           |                                                 | artificial solution, the addition of NaCl to the solution background                            |
|           |                                                 | helps maintain an ionic strength                                                                |
|           |                                                 | similar to that of natural seawater                                                             |
|           |                                                 | aiming to mimic any potential                                                                   |
|           |                                                 | dilution effect caused by the addition                                                          |
|           |                                                 | of HCl; although this effect is most                                                            |
|           |                                                 | likely negligible (Okamura et al.,                                                              |
|           |                                                 | 2014)."                                                                                         |

| 432 | I doubt the validity of this method for    | We thank the reviewer for this                        |
|-----|--------------------------------------------|-------------------------------------------------------|
|     | the reasons already mentioned in the       | thoughtful comment. The                               |
|     | comment on line 236 et seq. It would       | determination of $TA_{background}$ was                |
|     | be more appropriate to quantify the        | based on several independent                          |
|     | impurities affecting TA using              | *                                                     |
|     | independent measurement methods.           | solutions and repeated                                |
|     | One cannot use the same instrument         | measurements. The fact that the                       |
|     |                                            | solution with zero NaCl content                       |
|     | or procedure intended to be                | showed no detectable bias from the                    |
|     | calibrated with the RM to determine        | measurement method provides                           |
|     | the bias of that RM — this constitutes     | confidence that, although not the                     |
|     | circular reasoning. It becomes             | most rigorous approach, this method                   |
|     | impossible to distinguish whether the      | offers a reasonable first estimation                  |
|     | bias originates from the RM itself         | of $TA_{background}$ . In addition, the               |
|     | (e.g.,                                     | consistency observed between                          |
|     | NaCl impurities) or from the               | measurements at salinities 35                         |
|     | instrument or measurement procedure.       | (artificial solution) and 38 (natural                 |
|     | For instance, if the bias depends on       | seawater) suggests that any                           |
|     | the ionic strength of the RM,              | dependence on ionic strength is                       |
|     | demonstrating that the $\Delta TA$ at zero | unlikely.                                             |
|     | NaCl is zero does not resolve the          | Furthermore, the uncertainties                        |
|     | issue.                                     |                                                       |
|     | 13330                                      | associated with both the gravimetric                  |
|     |                                            | preparation of the solutions and the                  |
|     |                                            | potentiometric measurements were                      |
|     |                                            | explicitly taken into account in the                  |
|     |                                            | determination of $TA_{background}$ ,                  |
|     |                                            | which should limit traceability                       |
|     |                                            | issues.                                               |
|     |                                            | However, we agree that a better                       |
|     |                                            | method could be found, and that the                   |
|     |                                            | determination of TA background will |
|     |                                            | necessitate further rigorous                          |
|     |                                            | investigation this is no more clearly                 |
|     |                                            | presented in discussion part 3.3.2.                   |
|     |                                            | presented in discussion part 3.3.2.                   |
| 449 | Why only "linear": There could be          | The term "linear" has been removed.                   |
| 112 | other types of dependencies.               | The term initial has been femoved.                    |
| 451 | "However, it does not allow for the        | The sentence has been revised                         |
|     | accuracy verification of TA values         | accordingly: "However, the                            |
|     | obtained using the nonlinear least-        | different chemical composition                        |
|     | squares regression method" This            | compared to natural seawater                          |
|     | statement is not incorrect, but it is      | prevents from using the nonlinear                     |
|     | misleading, as it implies that the         | least-squares regression method,                      |
|     | evaluation method itself is the cause of   | which is yet widely applied to                        |
|     | the problem. In fact, if an RM with an     | natural seawater samples to correct                   |
|     | 1                                          | -                                                     |
|     | assigned value is available, the           | the value considering the acid-base                   |
|     | evaluation method is not critical —        | system in the solution."                              |
|     | any method-related bias can be             |                                                       |

|         | compensated by the known value of the RM. The real issue lies in the different chemical composition of natural versus artificial seawater. This difference necessitates using an evaluation method (NLLS) that differs from the one used to assign the value to the artificial seawater RM (Gran's method).                                                                                                                                                                                                                        |                                                                                                                                                                                                                                                                                                                                                                                                                                                            |
|---------|------------------------------------------------------------------------------------------------------------------------------------------------------------------------------------------------------------------------------------------------------------------------------------------------------------------------------------------------------------------------------------------------------------------------------------------------------------------------------------------------------------------------------------|------------------------------------------------------------------------------------------------------------------------------------------------------------------------------------------------------------------------------------------------------------------------------------------------------------------------------------------------------------------------------------------------------------------------------------------------------------|
| 441     | If a natural seawater RM is needed anyway to measure natural seawater, what is the benefit of using the artificial seawater RM?                                                                                                                                                                                                                                                                                                                                                                                                    | Having a reference material such as the developed artificial solution, with a potential for a SI traceable reference value provided alongside a comprehensive uncertainty budget (in the opposite of the natural solution) as several advantages, that are describe in the paragraph above (1.440-450) (e.g. wide range of TA values, quantification of acid titrant amount content)                                                                       |
| 458-462 | "Having a natural seawater reference material that is easy to collect during open- ocean oceanographic cruises" I find it difficult to see how this proposal could be implemented in practice, or what its benefit would be. Which institution would characterize such an in situ RM prepared by the operator during a cruise?  And if that were feasible, why would the user rely on any other RM? If the operator is capable of characterizing an RM, they could directly apply the same method to measure their samples. | Some oceanographic laboratories already produce home made standards, which has for interest that it can be produced in large volumes. (e.g. EuroGO-SHIP project). The artificial material could serve has a reference material for validating their measurement method before attributing a reference value. This secondary material could also be sent to reference laboratories (e.g. NMIs) for characterization. This has been added to the manuscript. |
| 466-470 | " method's limited precision": Where has this been discussed? As mentioned above, this evaluation should indeed be addressed (following the guidance in ISO 33405.                                                                                                                                                                                                                                                                                                                                                                 | This has been acknowledge in regard of ISO 33405 and is now discusses in sect. 3.3.4.                                                                                                                                                                                                                                                                                                                                                                      |

| is not stated. I assume they were intended to demonstrate that the carbon content did not change over time. Consequently, any instability of the RM must result from sources other than carbon, such as silicates. The authors should not leave it to the reader to infer the reasons for including specific results in the investigation.  491  " lack stability": A good observation that appropriately addresses the scope of the paper.  496  " indicating potential secondary processes influencing alkalinity." Such as? It is indeed a peculiar finding that the TA results do not reflect the increase in silicate. Identifying secondary processes in natural seawater may be difficult because of its complex composition. However, the composition of the artificial seawater is known—except perhaps for the NaCl impurities—so, an evaluation of the discrepancy should at least be feasible for the artificial RM.  The manuscript was changed accordingly.  The manuscript was changed accordingly.  Secondary processes might be biological activity, pollution, or other ion exchange processes with the glass. This has been added to the text.  It is difficult to evaluate the discrepancy on the artificial RM as no nutrients analyses were performed at the beginning of the experiment. | 483 | The purpose of the DIC measurements                                                                                                                                                                                                                                                                                                                                                                                                                                                                                                                                                                                                                                                                                                                                                                                                                                                                                                                                                                                                                                                                                                                                                                                                                                                                                                                                                                                                                                                                                                                                                                                                                                                                                                                                                                                                                                                                                                                                                                                                                                                                                            | The manuscript was changed        |
|-----------------------------------------------------------------------------------------------------------------------------------------------------------------------------------------------------------------------------------------------------------------------------------------------------------------------------------------------------------------------------------------------------------------------------------------------------------------------------------------------------------------------------------------------------------------------------------------------------------------------------------------------------------------------------------------------------------------------------------------------------------------------------------------------------------------------------------------------------------------------------------------------------------------------------------------------------------------------------------------------------------------------------------------------------------------------------------------------------------------------------------------------------------------------------------------------------------------------------------------------------------------------------------------------------------------|-----|--------------------------------------------------------------------------------------------------------------------------------------------------------------------------------------------------------------------------------------------------------------------------------------------------------------------------------------------------------------------------------------------------------------------------------------------------------------------------------------------------------------------------------------------------------------------------------------------------------------------------------------------------------------------------------------------------------------------------------------------------------------------------------------------------------------------------------------------------------------------------------------------------------------------------------------------------------------------------------------------------------------------------------------------------------------------------------------------------------------------------------------------------------------------------------------------------------------------------------------------------------------------------------------------------------------------------------------------------------------------------------------------------------------------------------------------------------------------------------------------------------------------------------------------------------------------------------------------------------------------------------------------------------------------------------------------------------------------------------------------------------------------------------------------------------------------------------------------------------------------------------------------------------------------------------------------------------------------------------------------------------------------------------------------------------------------------------------------------------------------------------|-----------------------------------|
| intended to demonstrate that the carbon content did not change over time. Consequently, any instability of the RM must result from sources other than carbon, such as silicates. The authors should not leave it to the reader to infer the reasons for including specific results in the investigation.  491  " lack stability": A good observation that appropriately addresses the scope of the paper.  496  " indicating potential secondary processes influencing alkalinity." Such as? It is indeed a peculiar finding that the TA results do not reflect the increase in silicate. Identifying secondary processes in natural seawater may be difficult because of its complex composition. However, the composition of the artificial seawater is known—except perhaps for the NaCl impurities—so, an evaluation of the discrepancy should at least be feasible for the artificial RM.                                                                                                                                                                                                                                                                                                                                                                                                                  | 103 |                                                                                                                                                                                                                                                                                                                                                                                                                                                                                                                                                                                                                                                                                                                                                                                                                                                                                                                                                                                                                                                                                                                                                                                                                                                                                                                                                                                                                                                                                                                                                                                                                                                                                                                                                                                                                                                                                                                                                                                                                                                                                                                                |                                   |
| carbon content did not change over time. Consequently, any instability of the RM must result from sources other than carbon, such as silicates. The authors should not leave it to the reader to infer the reasons for including specific results in the investigation.  491                                                                                                                                                                                                                                                                                                                                                                                                                                                                                                                                                                                                                                                                                                                                                                                                                                                                                                                                                                                                                                    |     | •                                                                                                                                                                                                                                                                                                                                                                                                                                                                                                                                                                                                                                                                                                                                                                                                                                                                                                                                                                                                                                                                                                                                                                                                                                                                                                                                                                                                                                                                                                                                                                                                                                                                                                                                                                                                                                                                                                                                                                                                                                                                                                                              | accordingly.                      |
| time. Consequently, any instability of the RM must result from sources other than carbon, such as silicates. The authors should not leave it to the reader to infer the reasons for including specific results in the investigation.  491                                                                                                                                                                                                                                                                                                                                                                                                                                                                                                                                                                                                                                                                                                                                                                                                                                                                                                                                                                                                                                                                       |     |                                                                                                                                                                                                                                                                                                                                                                                                                                                                                                                                                                                                                                                                                                                                                                                                                                                                                                                                                                                                                                                                                                                                                                                                                                                                                                                                                                                                                                                                                                                                                                                                                                                                                                                                                                                                                                                                                                                                                                                                                                                                                                                                |                                   |
| the RM must result from sources other than carbon, such as silicates. The authors should not leave it to the reader to infer the reasons for including specific results in the investigation.  491 " lack stability": A good observation that appropriately addresses the scope of the paper.  496 " indicating potential secondary processes influencing alkalinity." Such as? It is indeed a peculiar finding that the TA results do not reflect the increase in silicate. Identifying secondary processes in natural seawater may be difficult because of its complex composition. However, the composition of the artificial seawater is known—except perhaps for the NaCl impurities—so, an evaluation of the discrepancy should at least be feasible for the artificial RM.                                                                                                                                                                                                                                                                                                                                                                                                                                                                                                                               |     |                                                                                                                                                                                                                                                                                                                                                                                                                                                                                                                                                                                                                                                                                                                                                                                                                                                                                                                                                                                                                                                                                                                                                                                                                                                                                                                                                                                                                                                                                                                                                                                                                                                                                                                                                                                                                                                                                                                                                                                                                                                                                                                                |                                   |
| than carbon, such as silicates. The authors should not leave it to the reader to infer the reasons for including specific results in the investigation.  491 " lack stability": A good observation that appropriately addresses the scope of the paper.  496 " indicating potential secondary processes influencing alkalinity." Such as? It is indeed a peculiar finding that the TA results do not reflect the increase in silicate. Identifying secondary processes in natural seawater may be difficult because of its complex composition. However, the composition of the artificial seawater is known—except perhaps for the NaCl impurities—so, an evaluation of the discrepancy should at least be feasible for the artificial RM.                                                                                                                                                                                                                                                                                                                                                                                                                                                                                                                                                                     |     |                                                                                                                                                                                                                                                                                                                                                                                                                                                                                                                                                                                                                                                                                                                                                                                                                                                                                                                                                                                                                                                                                                                                                                                                                                                                                                                                                                                                                                                                                                                                                                                                                                                                                                                                                                                                                                                                                                                                                                                                                                                                                                                                |                                   |
| silicates. The authors should not leave it to the reader to infer the reasons for including specific results in the investigation.  491                                                                                                                                                                                                                                                                                                                                                                                                                                                                                                                                                                                                                                                                                                                                                                                                                                                                                                                                                                                                                                                                                                                                                                         |     |                                                                                                                                                                                                                                                                                                                                                                                                                                                                                                                                                                                                                                                                                                                                                                                                                                                                                                                                                                                                                                                                                                                                                                                                                                                                                                                                                                                                                                                                                                                                                                                                                                                                                                                                                                                                                                                                                                                                                                                                                                                                                                                                |                                   |
| to the reader to infer the reasons for including specific results in the investigation.  491 " lack stability": A good observation that appropriately addresses the scope of the paper.  496 " indicating potential secondary processes influencing alkalinity." Such as? It is indeed a peculiar finding that the TA results do not reflect the increase in silicate. Identifying secondary processes in natural seawater may be difficult because of its complex composition. However, the composition of the artificial seawater is known—except perhaps for the NaCl impurities—so, an evaluation of the discrepancy should at least be feasible for the artificial RM.                                                                                                                                                                                                                                                                                                                                                                                                                                                                                                                                                                                                                                     |     | , and the second |                                   |
| including specific results in the investigation.  491  "… lack stability": A good observation that appropriately addresses the scope of the paper.  496  "… indicating potential secondary processes influencing alkalinity." Such as? It is indeed a peculiar finding that the TA results do not reflect the increase in silicate. Identifying secondary processes in natural seawater may be difficult because of its complex composition. However, the composition of the artificial seawater is known—except perhaps for the NaCl impurities—so, an evaluation of the discrepancy should at least be feasible for the artificial RM.                                                                                                                                                                                                                                                                                                                                                                                                                                                                                                                                                                                                                                                                        |     |                                                                                                                                                                                                                                                                                                                                                                                                                                                                                                                                                                                                                                                                                                                                                                                                                                                                                                                                                                                                                                                                                                                                                                                                                                                                                                                                                                                                                                                                                                                                                                                                                                                                                                                                                                                                                                                                                                                                                                                                                                                                                                                                |                                   |
| investigation.  " lack stability": A good observation that appropriately addresses the scope of the paper.  496  " indicating potential secondary processes influencing alkalinity." Such as? It is indeed a peculiar finding that the TA results do not reflect the increase in silicate. Identifying secondary processes in natural seawater may be difficult because of its complex composition. However, the composition of the artificial seawater is known—except perhaps for the NaCl impurities—so, an evaluation of the discrepancy should at least be feasible for the artificial RM.                                                                                                                                                                                                                                                                                                                                                                                                                                                                                                                                                                                                                                                                                                                 |     |                                                                                                                                                                                                                                                                                                                                                                                                                                                                                                                                                                                                                                                                                                                                                                                                                                                                                                                                                                                                                                                                                                                                                                                                                                                                                                                                                                                                                                                                                                                                                                                                                                                                                                                                                                                                                                                                                                                                                                                                                                                                                                                                |                                   |
|  <li>" lack stability": A good observation that appropriately addresses the scope of the paper.</li> <li>" indicating potential secondary processes influencing alkalinity." Such as? It is indeed a peculiar finding that the TA results do not reflect the increase in silicate. Identifying secondary processes in natural seawater may be difficult because of its complex composition. However, the composition of the artificial seawater is known—except perhaps for the NaCl impurities—so, an evaluation of the discrepancy should at least be feasible for the artificial RM.</li> <li>The manuscript was changed accordingly.</li> <li>Secondary processes might be biological activity, pollution, or other ion exchange processes with text.</li> <li>It is difficult to evaluate the discrepancy on the artificial RM as no nutrients analyses were performed at the beginning of the experiment.</li>                                                                                                                                                                                                                                                                                                                                                                                   |     | - 1                                                                                                                                                                                                                                                                                                                                                                                                                                                                                                                                                                                                                                                                                                                                                                                                                                                                                                                                                                                                                                                                                                                                                                                                                                                                                                                                                                                                                                                                                                                                                                                                                                                                                                                                                                                                                                                                                                                                                                                                                                                                                                                            |                                   |
| observation that appropriately addresses the scope of the paper.  496  " indicating potential secondary processes influencing alkalinity." Such as? It other ion exchange processes with is indeed a peculiar finding that the TA results do not reflect the increase in silicate. Identifying secondary processes in natural seawater may be difficult because of its complex composition. However, the composition of the artificial seawater is known—except perhaps for the NaCl impurities—so, an evaluation of the discrepancy should at least be feasible for the artificial RM.                                                                                                                                                                                                                                                                                                                                                                                                                                                                                                                                                                                                                                                                                                                         |     |

---

## Author Comment (AC2)

**Response to reviewers – RC2**

OS manuscript egusphere-2025-3588

Title: Metrological concepts applied to Total Alkalinity measurements in seawater: reference materials, inter-laboratory comparison and uncertainty budget

Auhors: Gaëlle Capitaine, Samir Alliouane, Thierry Cariou, Jonathan Fin, Paola Fisicaro, Thibaut Wagener

**Dear reviewer,**

We would like to greatly thank you for taking the time to review our manuscript, and giving us the opportunity to improve it thanks to your valuable comments and suggestions. We have carefully considered all your comments. You will find below how we addressed them in the revised manuscript.

The line numbers correspond to the reviewed manuscript with changes marked.

| Line                   | Comment                                                                                                                                                                                                                                                                                                                                                                                                                                                                                                                                                                                                                                                                                                                                                                                                                                                                                                              | Response                                                                                                                                                                                                                                                                                                                                                                                                                                                                                                                                                                                                                                                                                                                                                                                                                                                                                                                         |
|------------------------|----------------------------------------------------------------------------------------------------------------------------------------------------------------------------------------------------------------------------------------------------------------------------------------------------------------------------------------------------------------------------------------------------------------------------------------------------------------------------------------------------------------------------------------------------------------------------------------------------------------------------------------------------------------------------------------------------------------------------------------------------------------------------------------------------------------------------------------------------------------------------------------------------------------------|----------------------------------------------------------------------------------------------------------------------------------------------------------------------------------------------------------------------------------------------------------------------------------------------------------------------------------------------------------------------------------------------------------------------------------------------------------------------------------------------------------------------------------------------------------------------------------------------------------------------------------------------------------------------------------------------------------------------------------------------------------------------------------------------------------------------------------------------------------------------------------------------------------------------------------|
| Line General comment 1 | The authors do not provide uncertainty estimates for the natural seawater reference material which does not have a well-characterized reference value traceable to the SI. Although information about the trueness of the reference value is required to estimate its uncertainty, it seems that this shouldn't prevent the authors from developing a partial uncertainty budget for the open-cell titration method on natural seawater samples (similar to the uncertainty budget they presented in Table 7). In section 4.4, the authors discuss additional contributions that would need to be considered in an uncertainty budget for the titration of natural seawater samples. Why was this not done? Presenting an uncertainty budget for natural seawater samples would be informative to understanding the likely overall uncertainty and its most important contributions in measurements of real samples. | The uncertainty budget presented in Table 6 corresponds to the titration measurement method and the Gran's data treatment, which is also part of the determination of AT on NSW, it thus contributes to establish a partial uncertainty budget for natural samples. Estimating the uncertainty of the additional data treatment for natural seawater (NLLS regression) requires several investigation. A few of them are reported below, and presented in Sect 5.3.2: (1) The uncertainty of practical salinity measurements (2) Possible discrepancies between total fluoride and total sulphate amount contents computed from salinity and the actual composition of natural seawaters worldwide (3) The uncertainty of dissociation constants of fluoride and sulphate ions. We agree this is highly needed and that a complete uncertainty budget for TA measurement results on natural seawater is required. However, these |
|                        |                                                                                                                                                                                                                                                                                                                                                                                                                                                                                                                                                                                                                                                                                                                                                                                                                                                                                                               | uncertainty budget for TA measurement results on natural seawater is required. However, these                                                                                                                                                                                                                                                                                                                                                                                                                                                                                                                                                                                                                                                                                                                                                                                                                                    |
|                        |                                                                                                                                                                                                                                                                                                                                                                                                                                                                                                                                                                                                                                                                                                                                                                                                                                                                                                                      | investigation are complex and represent a consequent amount of work, which will require, we believe, its own paper.                                                                                                                                                                                                                                                                                                                                                                                                                                                                                                                                                                                                                                                                                                                                                                                                              |
|                        |                                                                                                                                                                                                                                                                                                                                                                                                                                                                                                                                                                                                                                                                                                                                                                                                                                                                                                                      | This is presented as a perspective in the conclusion section.                                                                                                                                                                                                                                                                                                                                                                                                                                                                                                                                                                                                                                                                                                                                                                                                                                                                    |

| General              |                                                                                                                                                                                                                                                                                                                                                                                                                                                                                                                                                                                                                               | The level of confidence has been                                                                                                                                                                 |
|----------------------|-------------------------------------------------------------------------------------------------------------------------------------------------------------------------------------------------------------------------------------------------------------------------------------------------------------------------------------------------------------------------------------------------------------------------------------------------------------------------------------------------------------------------------------------------------------------------------------------------------------------------------|--------------------------------------------------------------------------------------------------------------------------------------------------------------------------------------------------|
| comment 2            | The authors should include information about the degrees of freedom when presenting estimates of expanded uncertainties, so the level of confidence associated with the chosen coverage factor can be determined.                                                                                                                                                                                                                                                                                                                                                                                                             | specified throughout the manuscript, being 95%.                                                                                                                                                  |
| General comment 3    | The authors should carefully reconsider the organization of their tables. Some tables are cluttered with too much information and should be split into separate tables (or moved to the supplementary information), while others contain redundant information and should be consolidated. Some suggestions are offered in the detailed comments. The formatting of the table also makes it hard to read, as the table is cut off at the end of the page. Although the formatting will be edited in the final manuscript, for the benefit of the reviewer, the authors should ensure that each table fits onto a single page. | Tables have been reviewed and table 6 is now the combination of two former tables (6&7). It has been ensured that each table fits into a single page.                                            |
| General
comment 4 | Many equations lack proper introduction and explanation. The authors should also carefully check the manuscript and make sure proper subscripts and superscripts are used for the various equation terms.                                                                                                                                                                                                                                                                                                                                                                                                                     | The manuscript has been reviewed according to this comment, and more specific comments below.                                                                                                    |
| 114-119              | The description of the process is unclear. It sounds like a total of 35 liters of seawater was collected from various depths and filled into two containers. From those two containers, 25 liters of seawater were drawn and filled into a single container to produce a single batch of seawater. Was the seawater homogenized before filling into the two containers and/or again after filling into the 25 liter container?                                                                                                                                                                                                | The seawater was homogenized first after collection and a second time after HgCl2 addition (i.e. right before bottling). This is detailed in the preparation section.                            |
| 126                  | "Artificial seawater" is a misnomer because the background medium is sodium chloride only without the other major seawater salts.                                                                                                                                                                                                                                                                                                                                                                                                                                                                                             | This has been specified in the text: "It should be noted that even if called "artificial seawater", the solution presented is made in a simple NaCl matrix, without other common seawater salts" |
| Table 1              | Are the percent purity values listed in
the table from the manufacturer or from
the assay results at NMIJ and SMU? If
they are assay values, they should be                                                                                                                                                                                                                                                                                                                                                                                                                                                          | They are assay values, they are now listed in a separated column. The assays can be considered as purity assessments. For NaHCO3 the                                                             |

|                     | listed in a separate column or a separate table rather than in parentheses after the manufacturer name.  What are the likely impurities in the salts, and were the impurities assessed?                                                                                                                                                                                                                                                                                                 | assay is made in base amount content, which can be higher to 100% due to decomposition of the salt.                                                                                                                                                                                                                                                                     |
|---------------------|-----------------------------------------------------------------------------------------------------------------------------------------------------------------------------------------------------------------------------------------------------------------------------------------------------------------------------------------------------------------------------------------------------------------------------------------------------------------------------------------|-------------------------------------------------------------------------------------------------------------------------------------------------------------------------------------------------------------------------------------------------------------------------------------------------------------------------------------------------------------------------|
| Table 1 & 3         | Two batches of artificial RM were produced, yet Table 1 only lists amount contents and molalities for one batch. Which batch do these values refer to? Also consider combining Table 1 with Table 3. The background alkalinity of the NaCl should also be explicitly listed in the consolidated table.                                                                                                                                                                                  | The composition given in Table 1 indicates the targeted composition. This has been specified in the text. We believe the targeted composition should remain in the method section while the table 3 contains results, especially the reference value of artificial solutions. The background Alkalinity has been added in this table.                                   |
| 129, table 1        | The use of pHT may not be appropriate in a NaCl medium as it does not contain sulfate. The temperature and dissociation constants used to calculate pH should also be given. The text indicates that the pH was estimated roughly based on a Bjerrum plot. The authors should provide a more precisely calculated value (especially if listing the pH in Table 1) or exclude the information about pH altogether, as it isn't strictly necessary to report for this reference material. | As in Wolf-Gadrow et al (2007), the stoichiometric pK values (pK*) typical for seawater were used here for the simple system, at 25°C. We agree the information about pH is not strictly needed for the material, we thus removed the detailed information from table 1 and from the text, only mentioning that we chose to approach a pH close to the one of seawater. |
| 167                 | "Material described in Appendix B" – Is this referring to the HCl standardized at SMU? Please state explicitly to avoid confusion and reference Appendix B for the details.                                                                                                                                                                                                                                                                                                             | It refers to the entire materials and devices presented in Appendix B.                                                                                                                                                                                                                                                                                                  |
| 422-425             | indicates the possibility of background alkalinity in the other salts such as NaHCO3 and Na2CO3. Was this assessed?                                                                                                                                                                                                                                                                                                                                                                     | As both NaHCO3 and Na2CO3 have been characterized by coulometry at NMIJ in term of base amount content, no additional background alkalinity should exist in these salts.  This has been added in the discussion section 3.3.2.                                                                                                                                          |
| 422-425             | What would be the intercept in Fig. 2 if the linear regression was not forced to zero? Might the choice to force the regression to zero discard information about the background alkalinity from the NaHCO3 and Na2CO3?                                                                                                                                                                                                                                                                 | The intercept would be 1.96 µmol.kg -1.  As both NaHCO3 and Na2CO3 have been characterized by coulometry at NMIJ in term of base amount content, no additional background alkalinity should exist in these salts.  This has been added in the discussion section 3.3.2.                                                                                                 |
| 202-204,
Table 3 | "Means of standard deviation" – If pooling standard deviations with uniform sample sizes, it should be calculated as the square root of the mean of the variances.                                                                                                                                                                                                                                                                                                                      | The manuscript and the values have been corrected accordingly.                                                                                                                                                                                                                                                                                                          |

| 216                    | Change phrasing to "ratio of the slope to the standard deviation of the slope." Also consider changing the notation so that the Student's t-value is not confused with t for time. α and 0 should be in subscripts. This comment also applies to                                                                                                                                                                                                                                                                                                                                                                                                                                                                                                                                                                                                                                                                                                                                                                                                                                                         | The manuscript has been corrected accordingly. The notation of the time in equations 12 and 13 has been slightly modified to avoid confusion.                                    |
|------------------------|----------------------------------------------------------------------------------------------------------------------------------------------------------------------------------------------------------------------------------------------------------------------------------------------------------------------------------------------------------------------------------------------------------------------------------------------------------------------------------------------------------------------------------------------------------------------------------------------------------------------------------------------------------------------------------------------------------------------------------------------------------------------------------------------------------------------------------------------------------------------------------------------------------------------------------------------------------------------------------------------------------------------------------------------------------------------------------------------------------|----------------------------------------------------------------------------------------------------------------------------------------------------------------------------------|
|                        | Table 4.                                                                                                                                                                                                                                                                                                                                                                                                                                                                                                                                                                                                                                                                                                                                                                                                                                                                                                                                                                                                                                                                                                 |                                                                                                                                                                                  |
| Equ 8, 1.248           | Aren't the salts added as stock solutions? In this case, Eq. 8 should have mstock instead of msalt. mtotal should be the sum of the stock solutions plus additional water rather than the sum of the salts and water.                                                                                                                                                                                                                                                                                                                                                                                                                                                                                                                                                                                                                                                                                                                                                                                                                                                                                    | The manuscript has been corrected accordingly.                                                                                                                                   |
| 280                    | "To maximize the uncertainty of the                                                                                                                                                                                                                                                                                                                                                                                                                                                                                                                                                                                                                                                                                                                                                                                                                                                                                                                                                                                                                                                                      | Yes, this has been more clearly                                                                                                                                                  |
|                        | slope" suggests that the goal was to have a larger uncertainty. I think what was meant was that the first approach with the larger uncertainty estimate was selected as the more conservative estimate of the uncertainty of the slope.                                                                                                                                                                                                                                                                                                                                                                                                                                                                                                                                                                                                                                                                                                                                                                                                                                                                  | specified in the text.                                                                                                                                                           |
| Equation 11, 1.288-302 | The equation for the homogeneity uncertainty does not make sense to me. It appears to be a standard deviation of the mean, but if so, it should be s / However, this would not make sense either as the between-bottle variability was estimated differently for the different batches—some batches using the standard deviation of single measurements from different bottles and another batch using the standard deviation of the bottle means from repeatability measurements. It also does not make sense why the within-bottle homogeneity was neglected in the overall homogeneity uncertainty, as the within-bottle homogeneity was explicitly estimated and listed in Table 4. The observed between bottle variance should be a sum of the within bottle variance and the homogeneity variance. If the between bottle variance is calculated as the standard deviation of the means from repeatability measurements in different bottles, then $s_{obs,bet-bil} = \sqrt{u_{hom}^2 + \frac{s_{repeatability}^2}{n}}$ where n is the number of repeatability measurements within a single bottle. | The equation 11 has been reviewed as $u_{hom} = \sqrt{(M_{between} - M_{within})/n0}$ following ISO 33405. Details on the computation of this value have been added to the text. |

|                   | As the other reviewer noted, the within and between bottle homogeneity components can be evaluated with ANOVA according to ISO Guide 35. It would be beneficial for many readers who do not have access to the ISO documents to derive these equations at a high level.                                                                                                                                              |                                                                                                                                                                                                                                                   |
|-------------------|----------------------------------------------------------------------------------------------------------------------------------------------------------------------------------------------------------------------------------------------------------------------------------------------------------------------------------------------------------------------------------------------------------------------|---------------------------------------------------------------------------------------------------------------------------------------------------------------------------------------------------------------------------------------------------|
| Equation 12       | The equations for the stability                                                                                                                                                                                                                                                                                                                                                                                      | The added term in equation 13                                                                                                                                                                                                                     |
| and 13, 1.305-311 | uncertainty require more explanation for the reader. Two equations are used. In the case of no significant trend, the stability uncertainty only has one contribution from the uncertainty of the slope b1, while for cases with significant trends, the stability uncertainty has an additional rectangular distribution component.  Also, what value is used for time t? And as noted before, this notation can be | corresponds to the estimated degradation of the material, this has been specified in the text. A subscript has also been added to the term $t(t_m)$ so that it cannot be confused with the Student's value. The value used for $t_m$ is 3 months. |
|                   | confused with the Student's t value.                                                                                                                                                                                                                                                                                                                                                                                 |                                                                                                                                                                                                                                                   |
| 530               | The phrasing in this sentence is confusing. The median of the set of means (from repeatability measurements made by each participant) was calculated for two different materials—natural seawater and the artificial RM (Batch 1).                                                                                                                                                                                   | The manuscript has been corrected accordingly.                                                                                                                                                                                                    |
| 545               | I recommend replacing "samples" with "materials" to be clear that it was two different materials being analyzed and not two bottles.                                                                                                                                                                                                                                                                                 | The manuscript has been corrected accordingly.                                                                                                                                                                                                    |
| 551               | How is the mean $(X_l - Y_l)$ calculated? Is it the mean difference from the 5 participants? Please clarify in the text. Why does the mean have a different subscript $l$ instead of $i$ , and what does it indicate?                                                                                                                                                                                                | Yes, the text has been clarified and the typo error has been corrected to make notations consistent.                                                                                                                                              |
| 16                | There seems to be missing text that should precede this equation. The text following the equation states that $s_r$ is the intra-laboratory standard deviation divided by the square root of the mean number of replicates — this should be explicitly written in the equation.                                                                                                                                      | The manuscript has been corrected accordingly.                                                                                                                                                                                                    |
| 628-639           | Both of these equations need proper introduction for the reader. They are based on the equations in ISO 21748. As some readers may not have access to the ISO documents, it would be helpful                                                                                                                                                                                                                         | Precisions from the ISO and for the calculation of $s_L$ and $s_r$ have been added. $u(\mu)$ has been replaced by $u_{RM}$ to be consistent with equation 6.                                                                                      |

|              | to provide an explanation of these equations and how the inter and intralaboratory standard deviations are calculated.  The term $u(\mu)$ needs further explanation than simply "standard deviation of the certified reference value." It is the uncertainty of the reference value which includes contributions from the characterization of the salts, the homogeneity, and stability ( Equation 6 ). The notation should be revised so that it is consistent with Equation 6 .                                                                                                                                                                                                                                                                                                                                                                                                                                                                                                                                                                                                                                                                                                 |                                                                                                                                                                                                                                                                                   |
|--------------|-------------------------------------------------------------------------------------------------------------------------------------------------------------------------------------------------------------------------------------------------------------------------------------------------------------------------------------------------------------------------------------------------------------------------------------------------------------------------------------------------------------------------------------------------------------------------------------------------------------------------------------------------------------------------------------------------------------------------------------------------------------------------------------------------------------------------------------------------------------------------------------------------------------------------------------------------------------------------------------------------------------------------------------------------------------------------------------------------------------------------------------------------------------------------------------------------|-----------------------------------------------------------------------------------------------------------------------------------------------------------------------------------------------------------------------------------------------------------------------------------|
| Table 3, 336 | The organization of this table is very confusing. Some entries are standard uncertainties with units of µmol kg -1 , while others are not. The caption indicates all numerical values have units of µmol kg -1 , but this is not true for parameters such as the slope, slope standard deviation, and Student t values. The stability uncertainty (from Equation 12 and 13) are not listed in this table. At the very least, the authors should clearly indicate in the table which parameters are standard uncertainties and include the appropriate units for each parameter. A better approach, I think, is to limit the table to only one type of information (e.g., the standard uncertainties associated with homogeneity, stability over time, and stability to transport). Additional details on the stability evaluation (e.g., the slope, t -tests, etc.) can be described in a separate table. The authors could also consider combining the information on the homogeneity and stability uncertainties from Table 4 and Table 5, although the natural seawater reference material does not have a certified value and an associated uncertainty | The units have been corrected in the table, as well as indication of standard deviations values.  Table 3 gives the results of the stability and homogeneity tests while table 4 gives the uncertainty values.  To make the distinction clearer, two sub-sections have been made. |
| Table 4      | Although Equation 11 will need to be revised, I will point out that the values listed for the homogeneity uncertainty do not agree with Equation 11 if using the between bottle standard deviations in Table 4 as s and N = 3. The authors should check their calculations in the tables.                                                                                                                                                                                                                                                                                                                                                                                                                                                                                                                                                                                                                                                                                                                                                                                                                                                                                                       | The between-bottle standard deviation was noted as $s^2$ in the former equation 11. Using the between bottle standard deviations in Table 4 as $s^2$ and $N = 3$ allows to get the correct $u_{hom}$ values reported. Although the equation 11 has indeed now been reviewed.      |

| ·                     | Ta                                                                                                                                                                                                                                                                                                                                                                                                                                                                                                                                                                                                                                |                                                                                                                                                                                                                                                                                                                                                                                                                                                                                                                                   |
|-----------------------|-----------------------------------------------------------------------------------------------------------------------------------------------------------------------------------------------------------------------------------------------------------------------------------------------------------------------------------------------------------------------------------------------------------------------------------------------------------------------------------------------------------------------------------------------------------------------------------------------------------------------------------|-----------------------------------------------------------------------------------------------------------------------------------------------------------------------------------------------------------------------------------------------------------------------------------------------------------------------------------------------------------------------------------------------------------------------------------------------------------------------------------------------------------------------------------|
| 585                   | $S_R$ is the reproducibility standard deviation. This term should be introduced and defined much earlier in Section 2.4.1, where it is used in Equation 16.                                                                                                                                                                                                                                                                                                                                                                                                                                                                       | Only $s_r$ , the intra-laboratory standard deviation, is used in equation 15 and 16, where it is defined.                                                                                                                                                                                                                                                                                                                                                                                                                         |
| 585                   | Replace "precision" with "reproducibility standard deviation" to be clear what quantity is being reported. The term "precision" should be reserved for qualitative descriptions. It may also be informative to report the reproducibility standard deviation excluding Laboratory 1, as their measurements were discovered to have a systematic error due to malfunctioning of the titrant delivery on their measurement system.                                                                                                                                                                                                  | Precision is defined in the VIM as the "closeness of agreement between [] measured quantity values obtained by replicate measurements on the same or similar objects under specified conditions", it is "expressed numerically by measures of imprecision, such as standard deviation". As indicated in the manuscript, the precision of the method is given by the computation of $s_L$ and $s_r$ , being, respectively, inter and intra laboratory variation. Laboratory 1 is already excluded from the calculation (line 561). |
| Table 5               | The numeric values should not be left in E+00 notation form The units also need to be specified for the standard uncertainties.                                                                                                                                                                                                                                                                                                                                                                                                                                                                                                   | The manuscript has been changed accordingly.                                                                                                                                                                                                                                                                                                                                                                                                                                                                                      |
| Section 5.3           | This section deserves more discussion of the results rather than just a description of the data contained in Table 6 and Table 7. Consider splitting this section into two—one discussing the top-down uncertainty estimates and the other discussing the bottom-up estimates.                                                                                                                                                                                                                                                                                                                                      | With the restructuration of the article suggested by reviewer 1, the discussion about the uncertainty estimate results now comes right after the result section, it has been splitted in accordance with your comment.                                                                                                                                                                                                                                                                                                            |
| Table 6,
1.690-692 | The formatting of this table needs much reworking to improve readability. The last column lists the individual standard uncertainties for the sub-sources of uncertainty and a combined standard uncertainty for the input parameter all in the same column. These should be separate columns. Other information is also needed such as the sensitivity coefficients used in the uncertainty propagation, and the degrees of freedom for each uncertainty contribution.  Consider including a more condensed table of the uncertainty budget in the main manuscript and a more detailed version in the supplementary information. | Table 6 has been reviewed accordingly. Tables have been added in Appendix D with details on the uncertainty propagation, integrating sensitivity coefficients.                                                                                                                                                                                                                                                                                                                                                                    |

|              | This section discussed leaching of silicate from the borosilicate glass bottles as a potential cause for instability in some batches of reference material. It would be beneficial for other reference material producers and for future investigations to provide more details on the specifications of the borosilicate glass used (such as the manufacturer and coefficient of linear expansion of the glass), as well as any cleaning procedures performed before bottling. Were the bottles cleaned in any way?  Line 140 states that Schott borosilicate bottles were used for Batch 2 of the artificial reference material. What about the other batches? | Other batches were bottled in Pyrex bottles, which may explain the difference observe in the amount of silicate release (although all bottles were borosilicate 3.3, i.e. same thermal expansion coefficient). This information has been added in the manuscript (Sect 3.3.4 and 3.1.1). The cleaning treatment has been specified. |
|--------------|------------------------------------------------------------------------------------------------------------------------------------------------------------------------------------------------------------------------------------------------------------------------------------------------------------------------------------------------------------------------------------------------------------------------------------------------------------------------------------------------------------------------------------------------------------------------------------------------------------------------------------------------------------------|-------------------------------------------------------------------------------------------------------------------------------------------------------------------------------------------------------------------------------------------------------------------------------------------------------------------------------------|
| 740          | This figure is rather confusing and not very informative. Although it highlights some major sources uncertainty such as the measured potential and the volume of acid delivered, it doesn't include all the sources of uncertainty in Table 7 and their magnitudes. A bar graph providing a visual summary of Table 7 may be a better choice.                                                                                                                                                                                                                                                                                                      | It doesn't include all the sources of uncertainties as the remaining sources are highly negligible and would not appear either on this diagram nor on a bar graph. This precision has been added to the text.                                                                                                                       |
| Equation A.2 | The total hydrogen ion concentration [H + ] T on the total pH scale includes free hydrogen ions and bisulfate ions only (Dickson, 1993). [H + ] + [HSO 4 - ] + [HF] is the total hydrogen ion concentration on the seawater pH scale. The notation should be revised in this equation.                                                                                                                                                                                                                                                                                                             | The equation is written as follows: $[H^+] + [HSO_4^-] + [HF] \approx [H^+]_T$ i.e. as approximately equal to the total hydrogen scale, as indeed, this doesn't include fluoride ions, which can be neglected here.                                                                                                                 |

---

## Referee Report (RR1)

Review of OS manuscript egusphere-2025-3588

Title: Metrological concepts applied to Total Alkalinity measurements in seawater: reference materials, inter-laboratory comparison and uncertainty budget

General comments

The revision has addressed most of the reviewer's concerns. A few more minor comments are mentioned below. The reviewer supports publication after they have been addressed. There is no need for another review.

Specific technical comments

(The line numbers indicate those given by the authors. They do not indicate the position exactly, but have nevertheless been kept by the reviewer.)

| line | Reviewer's comment | Author's comment | Re-review |
|---|---|---|---|
| general | The reviewer recommends using an LLM-based AI to improve the language. In parts, the paper is difficult to read due to linguistic weaknesses, which the reviewer did not further correct. | The language has been improved in parts were it was difficult to read but without using an AI based tool. | Technically, the topic is presented properly now. I leave it to the editors to assess language. |
| 21 | Remove "potentially". Either a result is traceable to a metrological reference or it isn't. There is no in-between status. | | New comment |
| 55 | The reviewer is not convinced that simply having no uncertainty budget is the sole reason why TA measurements using the conventional method are not traceable (to whatever reference). The method relies on several measured quantities the traceability of which may not be fully established or may even be inconsistent. For example, the total | | New comment. |

| | | | |
|---|---|---|---|
| | hydrogen ion concentration is quantified through pH/potential measurements using glass electrodes. However, what is the metrological refence of those results? Primary pH buffers, the values of which include or do not include the Bates-Guggenheim convention? In fact, the pH of those buffers is defined in terms of activity, while Dickson's guide assumes the potentials are a measure of H+ ion concentration. Moreover, how are liquid junction potentials of the glass electrode considered, which also affect the measured potentials significantly? Those are difficult questions to be answered in assessing traceability of the TA measurement procedure. That said, the reviewer does not intend to question the overall paper on the basis of these traceability concerns. However, it would be expected that this point is acknowledged in the introduction and the traceability section as an open issue. In fact, it even supports the value of the proposed artificial RM. | | |
| 172 | The measurement result at zero NaCl mol/kg sol is shown … | The manuscript has been changed accordingly. | Correct: The measurement results at zero NaCl mol kg-1 sol is shown in Fig. 1 and supports this reasonable assumption. Or: The measurement results at zero NaCl mol kg-1 sol is are shown in Fig. 1 and support this reasonable assumption. |
| 175 | Even with goodwill, Fig. 2 does not support the assumption of a linear relationship passing through | The fact that both the gravimetric and potentiometric approaches yield | "The measurements presented in Fig. 1, which |

| | | | |
|---|---|---|---|
| | the origin. It rather shows a square root like behavior, which is difficult to explain. Alternatively, the ΔTA values at 1, 2 and 3 mol/kg NaCl solution content indeed suggest that there is a linear relationship - which one can expect in dependence of NaCl content – but with an offset at zero NaCl content. Which raises the question, why the measured ΔTA value is zero at zero NaCl content? I suspect, that the reason for this discrepancy can be found in the different metrological references involved in the gravimetric and measured TA values. See also comments related to lines 236 and 522.

Anyhow, the linear extrapolation might be used as a rough estimate for the background alkalinity. However, the authors must comment on the difficulties I have mentioned. | ΔTA=0 μmol kg−1 at zero NaCl content supports the internal consistency of the measurements. We acknowledge, however, that the data presented in Figure 2 do not perfectly support a linear relationship passing through the origin, and we agree that this aspect warrants improvement. This limitation has been clarified and further discussed in the revised manuscript (Sect. 3.1.2 and 3.3.2) | can question the linear behaviour… " I would suggest to write "Linearity of the measurement results is a rough assumption that is further discussed in section 3.3.2." |
| 285 | "This study highlighted that the determination of the homogeneity is highly dependent on the variability of the measurement method." Should rather be "This study highlighted that the robustness of the determination of the homogeneity is highly dependent on the variability of the measurement method. | | New comment |
| 288 | "It was chosen to neglect the within-bottle homogeneity." Again, the authors claim compliance with ISO Guide 35 (ISO 33405, respectively); however, their homogeneity analysis | A one-way ANOVA has been performed on the results obtained from the homogeneity testing. The ANOVA results were then used to calculate the between-bottle homogeneity uncertainty based on | Add reason: "Uncertainty resulting from within-bottle inhomogeneity can usually |

| | | | |
|---|---|---|---|
| | appears superficial to some extent. A one-way ANOVA must be applied to account for both within-unit and between-unit homogeneity. One might decide to disregard within-unit uncertainty for the reasons mentioned by the authors. In that case, only between-bottle homogeneity should be calculated according to the (corrected) Eq. 11. Otherwise, a proper one-way ANOVA analysis is expected for the homogeneity values given in Table 4. The authors must also evaluate the repeatability standard deviation of the homogeneity with respect to the target uncertainty (see Section 7.5.1 of ISO 33405). | the corrected equation 11 (see comment below). The corresponding values of *uhom* have been corrected accordingly in the manuscript. ISO 33405:2024 (Section 7.5.1 of the ISO), states that the repeatability standard deviation of the homogeneity study procedure should be less than one third of the target standard uncertainty of the TA measurement result for the procedure to be considered suitable. In our case, the criterion was slightly exceeded, but the results can nevertheless be regarded as a preliminary estimate of the material's homogeneity. This has been added in the discussion section 3.4.4. | be neglected for liquid refence materials." A quantitative number should be added to the discussion in 3.4.4 to support the statement that ", the (1/3) criterion was slightly exceeded", i.e. how do the measurement repeatabilities of the three batches, which can be expressed by M-within, compare to the target uncertainty. |
| 426 | Again, remove "potential". The issue of the TA-background has been appropriately discussed. It is indeed an issue that must be addressed. But it is not so significant that SI traceability of the assigned TA value must be stated as "potential". It is rather an uncertainty of the uncertainty. | | New comment |
| 450 | Replace "precision" by "repeatability". | | New comment |
| 458 | "Having a natural seawater reference material that is easy to collect during open-ocean oceanographic cruises …" I find it difficult to see how this proposal could be implemented in practice, or what its benefit would be. Which institution would characterize such an *in situ* RM prepared by the operator during a cruise? And if that were feasible, why would the user rely on any other RM? If the | Some oceanographic laboratories already produce home made standards, which has for interest that it can be produced in large volumes. (e.g. EuroGO-SHIP project). The artificial material could serve has a reference material for validating their measurement method before attributing a reference value. This secondary material could also be sent to reference laboratories (e.g. NMIs) for characterization. This has been added to the manuscript. | I see the necessity, but I am not yet convinced that the proposed concept of a kind of "practical traceability" using two RMs having different traceabilities has been developed in sufficient metrological depth. However, the issue is not fundamental in the context |

| | operator is capable of characterizing an RM, they could directly apply the same method to measure their samples. | | of the paper (even though it is very fundamental in general). Thus, I consider it resolved. |
|---|---|---|---|
| 586 | "… lack stability": A good observation that appropriately addresses the scope of the paper. | The manuscript was changed accordingly. | ?
Why? This comment didn't request a change. I was just appreciating the observation. |

---

## Author Response (AR2)

OS manuscript egusphere-2025-3588

Title: Metrological concepts applied to Total Alkalinity measurements in seawater: reference materials, inter-laboratory comparison and uncertainty budget

Auhors: Gaëlle Capitaine, Samir Alliouane, Thierry Cariou, Jonathan Fin, Paola Fisicaro, Thibaut Wagener

**Response to the Editor**

Dear Editor Mario Hoppema,

We would like to greatly thank you for taking the time to review our manuscript, and giving us the opportunity to improve it thanks to your valuable comments and suggestions. We have carefully considered all your editorial comments; they have been integrated in the reviewed manuscript.

**Response to reviewer – RC1**

Dear reviewer,

We would like to greatly thank you for taking the time to review our manuscript this second time, and giving us the opportunity to improve it thanks to your valuable comments and suggestions. We have carefully considered all your comments. You will find below how we addressed them in the revised manuscript.

The line numbers correspond to the reviewed manuscript with changes marked.

| Line | Comment | Response |
|---|---|---|
| General comment | The reviewer recommended using an LLM-based AI to improve the language. Technically, the topic is presented properly now. I leave it to the editors to assess language. | This comment is considered addressed. |
| 21 | Remove "potentially". Either a result is traceable to a metrological reference or it isn't. There is no in-between status. | The manuscript has been changed accordingly. |
| 56-57, 731 | The reviewer is not convinced that simply having no uncertainty budget is the sole reason why TA measurements using the conventional method are not traceable (to whatever reference). The method relies on several measured quantities the traceability of which may not be fully established or may even be inconsistent. For example, the total hydrogen ion concentration is quantified through pH/potential measurements using glass electrodes. However, what is the metrological refence of those results? Primary pH buffers, the values of which include or do not include the Bates- | The introduction has been changed as follows: "However, the RMs distributed aren't fully traceable **partly** due to the fact that they aren't given with a rigorously assessed uncertainty. **Other traceability issues coming from the measurement process should also be carefully investigated**". This has also been specified in the traceability section. |

| | | |
|---|---|---|
| | Guggenheim convention? In fact, the pH of those buffers is defined in terms of activity, while Dickson's guide assumes the potentials are a measure of H+ ion concentration. Moreover, how are liquid junction potentials of the glass electrode considered, which also affect the measured potentials significantly? Those are difficult questions to be answered in assessing traceability of the TA measurement procedure. That said, the reviewer does not intend to question the overall paper on the basis of these traceability concerns. However, it would be expected that this point is acknowledged in the introduction and the traceability section as an open issue. In fact, it even supports the value of the proposed artificial RM. | |
| 172 | Correct: The measurement result at zero NaCl mol kg- 1 sol is shown in Fig. 1 and support**s** this reasonable assumption.
Or: The measurement results at zero NaCl mol kg- 1 sol  are shown in Fig. 1
and support this reasonable assumption. | The manuscript has been corrected accordingly. |
| 176 | "The measurements presented in Fig. 1, which can question the linear behaviour… " I would suggest to write "Linearity of the measurement results is a rough assumption that is further discussed in section 3.3.2." | The manuscript has been changed accordingly. |
| 287 | "This study highlighted that the determination of the homogeneity is highly dependent on the variability of the measurement method." Should rather be "This study highlighted that the robustness of the determination of the homogeneity is highly dependent on the variability of the measurement method. | The manuscript has been changed accordingly. |
| 289, 464 | Add reason: "Uncertainty resulting from within-bottle inhomogeneity can usually be neglected for liquid refence materials."
A quantitative number should be added to the discussion in 3.4.4 to support the statement that ", the (1/3) criterion was slightly exceeded", i.e. how do the measurement repeatabilities of the three batches, which can be expressed by M-within, compare to the target uncertainty. | The reason given has been added. The M_within value has also been added in the discussion section, being of 1.5µmol/kg, while the targeted uncertainty is 2µmol/kg. |

| | | |
|---|---|---|
| 436 | Again, remove "potential". The issue of the TA- background has been appropriately discussed. It is indeed an issue that must be addressed. But it is not so significant that SI traceability of the assigned TA value must be stated as "potential". It is rather an uncertainty of the uncertainty. | The manuscript has been changed accordingly. |
| 460 | Replace "precision" by "repeatability". | The manuscript has been changed accordingly. |
| Section 6 | I see the necessity, but I am not yet convinced that the proposed concept of a kind of "practical traceability" using two RMs having different traceabilities has been developed in sufficient metrological depth. However, the issue is not fundamental in the context of the paper (even though it is very fundamental in general). Thus, I consider it resolved. | This comment is considered addressed. |
| 486 | ? Why? This comment didn't request a change. I was just appreciating the observation. | "was less stable" was changed to "lack stability" in the previous round of revision. This didn't change the meaning of the sentence. |

**Response to reviewer – RC2**

Dear reviewer,

We would like to greatly thank you for taking the time to review our manuscript this second time, and giving us the opportunity to improve it thanks to your valuable comments and suggestions. We have carefully considered all your comments. You will find below how we addressed them in the revised manuscript.

The line numbers correspond to the reviewed manuscript with changes marked.

| Line | Comment | Response |
|---|---|---|
| 166-167 | Line 165: Instead of stating "using the materials and devices presented in Appendix B" and leaving the reader to search for this information in a supplementary section and losing their train of thought, it would be clearer to state, for example, "using the HCl standardized at SMU and a the Metrohm titration system as described in Appendix B." | The manuscript has been changed accordingly. |
| 815 | Line 802: Metrohm is misspelled. | The manuscript has been changed accordingly. |

| 577 | Line 567: "The precision of the method, $sR$..." The authors did not address my previous comment here. They introduce a new notation $sR$ that is not used in the other equations. (Equations 15 and 16 use sr for intra-laboratory standard deviation, which is different from sR.) The authors define sR as "precision...given by computation of sL and sr," but they don't describe exactly how sR is computed. Based on Table 5, it appears that sR is computed by summing sL and sr in quadrature (i.e., $sR = sqrt(sL^2+sr^2) = 1.99$ umol/kg as the authors report in the text). | The notation "$sR$" has been removed to avoid confusion. |
|---|---|---|
| Throughout the manuscript | Scientific notation: In various places in the manuscript, the authors have still left numeric values in E+XX notation rather than in scientific notation. | All values has been changed for scientific notation. |
| 340-347 | Fig 2 is a new addition in the revised manuscript. I would recommend including the results of Batch 1 on this graph so the stability of the two batches can be compared. | Figure 2 has been updated accordingly. |
| Throughout the manuscript | Expanded uncertainties and coverage factors: Although it is acceptable to report expanded uncertainties with a coverage factor of 2, it may not always represent a 95% level of confidence, depending on the effective degrees of freedom which was why I recommended the authors report this information. It can be included in the supplementary section. It may also be acceptable to state that the coverage factor of 2 corresponds approximately to a 95% level of confidence if this assumption is true for the authors' estimates. | It has been specified that the coverage factor of 2 corresponds approximately to a 95% level of confidence. |